# FAST ESCAPE, SLOW CONVERGENCE: LEARNING DYNAMICS OF PHASE RETRIEVAL UNDER POWER-LAW DATA

**Guillaume Braun**
RIKEN AIP

**Bruno Loureiro**
École Normale Supérieure, PSL & CNRS

**Ha Quang Minh**
RIKEN AIP

**Masaaki Imaizumi**
University of Tokyo & RIKEN AIP & Kyoto University

## ABSTRACT

Scaling laws describe how learning performance improves with data, compute, or training time, and have become a central theme in modern deep learning. We study this phenomenon in a canonical nonlinear model: phase retrieval with anisotropic Gaussian inputs whose covariance spectrum follows a power law. Unlike the isotropic case, where dynamics collapse to a two-dimensional system, anisotropy yields a qualitatively new regime in which an infinite hierarchy of coupled equations governs the evolution of the summary statistics. We develop a tractable reduction that reveals a three-phase trajectory: (i) fast escape from low alignment, (ii) slow convergence of the summary statistics, and (iii) spectral-tail learning in low-variance directions. From this decomposition, we derive explicit scaling laws for the mean-squared error, showing how spectral decay dictates convergence times and error curves. Experiments confirm the predicted phases and exponents. These results provide the first rigorous characterization of scaling laws in nonlinear regression with anisotropic data, highlighting how anisotropy reshapes learning dynamics.

## 1 INTRODUCTION

Scaling laws quantify how the performance of a learning algorithm varies with resources such as training time, dataset size, or model capacity. Empirically, losses often follow simple power laws across wide ranges of data and computation, enabling forecasting from a handful of measurements (Hestness et al., 2017; Kaplan et al., 2020; Hoffmann et al., 2022). These regularities naturally raise the fundamental question: *when and how do such laws emerge from first principles?*

Despite their central role in modern deep learning practice, neural scaling laws remain theoretically poorly understood. A notable exception is provided by linear models, where the scaling of the generalization error has been thoroughly analysed within the classical kernel literature, encompassing both ridge regression (Caponnetto & De Vito, 2007; Rudi & Rosasco, 2017) and stochastic gradient descent (Yao et al., 2007; Ying & Pontil, 2008; Carratino et al., 2018; Pillaud-Vivien et al., 2018; Kunstner & Bach, 2025). Recent developments in this direction, driven by the empirical observations of cross-overs and bottlenecks in the context of neural networks, demonstrate that analogous phenomena are already present in linear settings (Cui et al., 2021; Defilippis et al., 2024; Bahri et al., 2024; Maloney et al., 2022; Atanasov et al., 2024; Paquette et al., 2024; Bordelon et al., 2024; Lin et al., 2024). By contrast, *nonlinear* settings, ubiquitous in practice (e.g., functional data, learned feature maps, or embeddings with heavy spectral tails), remain far less understood.

We address this gap in the canonical nonlinear regression problem of phase retrieval:

$$y = \langle x, w^\star \rangle^2 + \xi, \qquad x \sim \mathcal{N}(0, Q),$$

where $w^\star \in \mathbb{R}^d$ is the target vector, and $Q$ has eigenvalues $(\lambda_i)_{i=1}^d$ obeying a power law $\lambda_i \propto i^{-a}$ with $a > 1$ and noise $\xi$. This model captures two core difficulties: (i) a nonconvex landscape, and (ii) strong anisotropy that induces highly unbalanced learning across directions.

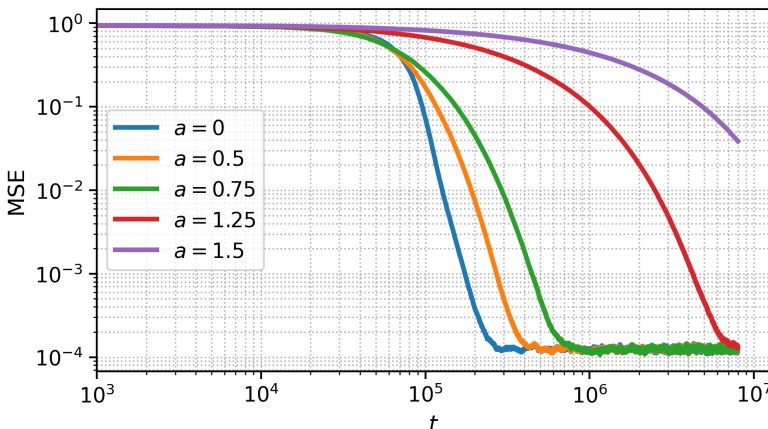

Figure 1: Evolution of the MSE during training with online SGD for different spectral exponents $a$ (log-log scale). For $a > 1$, convergence is markedly slower than the exponential decay seen in the isotropic case, reflecting the difficulty of learning directions associated with small eigenvalues.

Figure 1 illustrates this phenomenon: under the same initialization, noise level, stepsize, and optimization algorithm (SGD), the mean-square error (MSE) behaves differently depending on $a$. Intuitively, when $a$ is larger, directions associated with small eigenvalues $\lambda_i$ are harder to learn, so the MSE decays more slowly. By contrast, in isotropic designs, all directions progress at comparable rates, causing the error to drop sharply. These observations motivate our central question:

*How does the input spectrum govern finite-time convergence in nonlinear regression, and can we predict the learning curve from the spectral decay?*

Concretely, we seek to (i) analyze the mechanisms behind the plateau–drop structure in anisotropic phase retrieval, and (ii) derive scaling laws that quantify MSE decay as functions of the spectral parameter $a$ and time $t$.

### 1.1 CONTRIBUTIONS.

- **New phenomena in the anisotropic case.** We show that anisotropy challenges several aspects of the intuition developed in isotropic settings. In the isotropic setting, the dynamics collapse to a low-dimensional ODE, and the main challenge is escaping mediocrity, i.e. the regime where the correlation with the signal $w^*$ is vanishing, before convergence accelerates. Under anisotropy, by contrast, the dynamics form an infinite hierarchy of coupled equations. This structural change flips the qualitative behavior, yielding an *escape-convergence trade-off*: escaping from mediocrity can be faster, but convergence to a low MSE is slowed by the difficulty of learning directions associated with small eigenvalues, as illustrated by Figure 1 (a large $a$ is associated with a slow decay of the MSE) and Figure 3 (a large $a$ leads quickly to constant order correlation with the signal). Numerical experiments confirm this phase-level contrast between isotropic and anisotropic regimes.

- **Analytical framework.** We first obtain a closed-form representation of the dynamics via *Duhamel's formula*. We then introduce a phase decomposition of the trajectory, isolating regimes where different approximations become valid. This allows us to analyze the qualitative behavior of the ODE hierarchy phase by phase: (i) *fast escape from mediocrity*, (ii) *macroscopic convergence of summary statistics*, and (iii) *spectral-tail learning of small-eigenvalue directions*. The combination of Duhamel representation and phase-specific approximations provides a systematic way to make infinite-dimensional dynamics tractable.

- **Scaling laws.** As a byproduct of this analysis, we derive explicit scaling laws for anisotropic phase retrieval. These formulas quantify how the eigenvalue decay governs the MSE. Numerical results corroborate the predicted exponents across different spectral profiles.

## 1.2 OTHER RELATED WORK

**Theory of scaling laws.** The study of risk scaling in problems with power-law structure is a classical theme in the kernel literature, where it falls under the framework of *source and capacity conditions*. It has been extensively investigated for kernel ridge regression (Caponnetto & De Vito, 2007; Cui et al., 2021), random features regression (Rudi & Rosasco, 2017; Defilippis et al., 2024), and also for (S)GD (Yao et al., 2007; Ying & Pontil, 2008; Carratino et al., 2018; Pillaud-Vivien et al., 2018). This line of work has recently gained renewed relevance in the context of neural scaling laws, with linear models emerging as theoretical testbeds to explain the plateaux and crossovers observed in practice (Bahri et al., 2024; Maloney et al., 2022; Atanasov et al., 2024; Bordelon et al., 2024; Worschech & Rosenow, 2024; Paquette et al., 2024; Lin et al., 2024). More recently, Wortsman & Loureiro (2025) have investigated kernel ridge regression under anisotropic power-law inputs, showing that in some regimes the scaling of the covariates transfer to the features. Sums of orthogonal single-index models have been analysed in the *feature-learning* regime, albeit still under isotropic input distributions (Ren et al., 2025; Ben Arous et al., 2025; Defilippis et al., 2025). In short, while the linear setting with anisotropic spectra (e.g. power-law decays) is by now well understood, the nonlinear regime has remained largely isotropic. Our work addresses this gap by deriving compute–error scaling laws in a nonlinear model with anisotropic Gaussian inputs.

**Phase retrieval and quadratic neural networks.** Phase retrieval (PR) is a classical inverse problem motivated by imaging: reconstruct a signal from intensity-only measurements; see Dong et al. (2023) for a recent tutorial. A rich algorithmic literature includes spectral initializations and non-convex Wirtinger-flow refinements (Ma et al., 2021; Candès et al., 2015; Tan & Vershynin, 2019; 2023; Davis et al., 2020). PR can be viewed as learning a single neuron with a quadratic activation, connecting it to the broader theory of quadratic networks and their training dynamics (Sarao Mannelli et al., 2020; Arnaboldi et al., 2023a; Martin et al., 2024; Erba et al., 2025). In this context, Ben Arous et al. (2025) have studied scaling laws for one-pass SGD and Defilippis et al. (2025) for full-batch ERM, in a quadratic net model where the target coefficients follow a power-law. Most theoretical analyses of PR assume isotropic sub-Gaussian measurements; the impact of anisotropic covariances on the learning curve has received less attention. Our analysis isolates precisely this aspect and quantifies how the input spectrum shapes the three-phase trajectory and the resulting scaling laws.

**Non-convex optimization and feature learning.** A complementary line of work studies gradient-based training in multi-index and shallow networks, characterizing feature learning, convergence phases, and computational–statistical trade-offs, predominantly under isotropic designs (Saad & Solla, 1995a;b; Goldt et al., 2019; Veiga et al., 2022; Arnaboldi et al., 2023b; 2024; Collins-Woodfin et al., 2024; Ben Arous et al., 2022; Abbe et al., 2022; 2023; Bietti et al., 2025; Dandi et al., 2024; Bruna & Hsu, 2025). Results with anisotropic inputs are more limited: some works consider spiked covariances and rely on preconditioned methods (Ba et al., 2023), while others treat a broader but weaker anisotropic regime that does not include power-law spectra (Goldt et al., 2020; Braun et al., 2025). By contrast, we analyze the unpreconditioned gradient flow in a strongly anisotropic regime and show that anisotropy destroys the finite-dimensional closure of the dynamics, leading to an infinite hierarchy whose Duhamel–Volterra reduction yields explicit scaling laws.

## 1.3 NOTATIONS

We use $\|\cdot\|$ and $\langle \cdot, \cdot \rangle$ to denote the Euclidean norm and scalar product, respectively. When applied to a matrix, $\|\cdot\|$ refers to the operator norm. Any positive definite matrix $Q$ induces a scalar product defined by $\langle x, y \rangle_Q = x^\top Q y$. The Frobenius norm of a matrix $A$ is denoted by $\|A\|_F$. The $d \times d$ identity matrix is represented by $I_d$. The $(d-1)$-dimensional unit sphere is denoted by $\mathbb{S}^{d-1}$. We use the notation $a_n \lesssim b_n$ (or $a_n \gtrsim b_n$) for sequences $(a_n)_{n \geq 1}$ and $(b_n)_{n \geq 1}$ if there exists a constant $C > 0$ such that $a_n \leq C b_n$ (or $a_n \geq C b_n$) for all $n$. If the inequalities hold only for sufficiently large $n$, we write $a_n = O(b_n)$ (or $a_n = \Omega(b_n)$). We denote by $C_b^\infty(\mathbb{R}_{\geq 0}; \ell^2)$ the Banach space of bounded and infinitely differentiable functions from $\mathbb{R}_{\geq 0}$ to $\ell^2 := \{(x_n)_{n \geq 0} : \sum x_n^2 < \infty\}$, the Hilbert space of square summable sequences. We write $*$ for convolution, and for a function $f$ we denote by $\hat{f}$ its Laplace transform.

## 2 PROBLEM SETUP

**Data distribution.** By orthogonal invariance of the Gaussian distribution, we may assume without loss of generality that the covariance matrix is diagonal: $Q = \mathrm{diag}(\lambda_1, \dots, \lambda_d) \in \mathbb{R}^{d \times d}$. We assume a *power-law* spectrum,

$$\lambda_i = \frac{i^{-a}}{\sum_{j=1}^d j^{-a}}, \qquad a > 1,$$

so that $\mathrm{tr}(Q) = 1$. This assumption reflects the slow, heavy-tailed eigenvalue decay observed in many empirical covariance spectra (e.g., images, text embeddings, kernel features). From a theoretical perspective, the exponent $a$ provides a simple parametrization of anisotropy: it controls the balance between a few dominant directions and a long tail of weak ones. It is widely used as a canonical model for scaling laws in learning dynamics. We consider the phase retrieval setting

$$y = \langle x, w^\star \rangle^2 + \xi, \qquad x \sim \mathcal{N}(0, Q),$$

where the target weights are generated by $w^\star = \frac{u}{\|Q^{1/2}u\|}, u \sim \mathcal{N}(0, I_d)$. This construction ensures that $\|Q^{1/2}w^\star\| = 1$. For numerical experiments, we sometimes adopt the alternative normalization $\|w^\star\|^2 = d$, so that all curves start from the same baseline when comparing different decay exponents $a$; both normalizations are of similar order. The noise term $\xi \sim \mathcal{N}(0, \sigma^2)$ is independent of $x$. Since our analysis focuses on the *population dynamics*, label noise only shifts the loss by a constant and can be set to zero without loss of generality.

**Model.** We consider estimators of the form $\langle x, w \rangle^2$ for $w \in \mathbb{R}^d$ and the population loss used to train our model

$$\mathcal{L}(w) = \mathbb{E}_x\Big[\big((x^\top w)^2 - (x^\top w^\star)^2\big)^2\Big].$$

However, under anisotropy, this loss can be misleading: a vector $w$ that aligns with $w^\star$ only along the directions corresponding to the largest eigenvalues may still achieve a small loss. To evaluate how well $w$ actually recovers the signal direction, a more natural metric is the MSE defined as

$$\mathrm{MSE}(w, w^\star) = \frac{1}{d}\min\{\|w - w^\star\|^2,\ \|w + w^\star\|^2\}.$$

**Optimization.** The learner maintains a weight vector $w(t) \in \mathbb{R}^d$, initialized uniformly at random on the unit sphere $\mathbb{S}^{d-1}$. Its evolution follows the gradient flow dynamics

$$\dot{w}(t) = -\nabla_w \mathcal{L}(w(t))$$

which can be viewed as the continuous-time counterpart of gradient descent.

## 3 LOSS GEOMETRY AND EVOLUTION OF SUMMARY STATISTICS

In this section, we describe the main characteristics of the loss landscape and establish ODEs to describe the evolution of the key summary statistics. The proofs are in the appendix, Section B. For convenience, we introduce the following notations: $s := \|w\|_Q^2 = w^\top Q w$, $s_\star := \|w^\star\|_Q^2$, and $u := \langle w, w^\star \rangle_Q = w^\top Q w^\star$.

### 3.1 LOSS SIMPLIFICATION

Although the loss is defined on a $d$-dimensional parameter space, it depends only on two summary statistics: $\|w\|_Q^2$ and $\langle w, w^\star \rangle_Q^2$. This dimensional reduction makes the geometry of the loss landscape transparent, as captured in the following proposition.

**Proposition 1.** *Let $x \sim \mathcal{N}(0, Q) \in \mathbb{R}^d$, where $Q \in \mathbb{R}^{d \times d}$ is a symmetric positive definite diagonal matrix. Let $w, w^\star \in \mathbb{R}^d$. The population loss can be rewritten as*

$$\mathcal{L}(w) = 3\|w\|_Q^4 + 3\|w^\star\|_Q^4 - 4\langle w, w^\star \rangle_Q^2 - 2\|w\|_Q^2 \cdot \|w^\star\|_Q^2 = 3s^2 + 3s_\star^2 - 4u^2 - 2s_\star s$$

Next, we compute the population gradient and characterize the critical points.

## 3.2 Gradient, Hessian, and critical points

**Proposition 2.** *The population gradient and Hessian of the loss $\mathcal{L}$ are given by*

$$\nabla\mathcal{L}(w) = 12sQw - 4s_\star Qw - 8uQw^\star, \tag{3.1}$$

$$\nabla^2\mathcal{L}(w) = 24(Qw)(Qw)^\top + (12s - 4s_\star)Q - 8(Qw^\star)(Qw^\star)^\top. \tag{3.2}$$

*The set of critical points is $\{0,\ w^\star,\ -w^\star\} \cup \{w : (u,s) = (0, s_\star/3)\}$. Among them, $0$ is a strict local maximum, $\pm w^\star$ are strict global minima, and every $w$ with $(u,s) = (0, s_\star/3)$ is a saddle point.*

**Remark 1.** *The loss landscape contains no spurious local minima: the only minima are the global optima $\pm w^\star$. At the same time, the origin is a strict local maximum, and the remaining critical points are saddles. Nevertheless, the dynamics exhibit a long plateau phase. At initialization, one typically has $u(0) \approx d^{-1/2} \approx 0$. In this low-correlation regime, the gradient is small, so the iterates take a long time to escape. Even after leaving this regime, convergence remains slower than in the isotropic case.*

**Remark 2.** *The anisotropic gradient flow can be viewed as a $Q$-preconditioned version of the isotropic flow. Under the reparametrization $z = Q^{1/2}w$, define the isotropic loss $\mathcal{L}_I(z) = \mathcal{L}(Q^{-1/2}z)$. By the chain rule, $\dot{z}(t) = -Q\nabla\mathcal{L}_I(z(t))$. Thus, while the critical points coincide with those in the isotropic case, the dynamics differ substantially due to the preconditioning by $Q$.*

## 3.3 Infinite-dimensional structure of gradient flow

As shown in Proposition 1, the loss depends only on the summary statistics $u(t)$ and $s(t)$. Controlling these two quantities already yields a faithful description of the loss trajectory.

A crucial distinction, however, arises between the isotropic and anisotropic settings. In the isotropic case ($Q = I$), the dynamics of the loss can be expressed entirely in terms of two scalars: the signal overlap $u(t)$ and the energy $s(t)$. This leads to a closed, two-dimensional ODE system.

In contrast, as soon as $Q \neq I$, the situation changes qualitatively. Proposition 3 shows that the evolution of $u(t)$ and $s(t)$ necessarily involves higher-order weighted overlaps,

$$s^{(k)}(t) := \|w(t)\|_{Q^k}^2, \qquad s_\star^{(k)} := \|w^\star\|_{Q^k}^2, \qquad u^{(k)}(t) := \langle w(t), w^\star\rangle_{Q^k},$$

with the conventions $s = s^{(1)}(t)$, $u = u^{(1)}(t)$, and $s_\star = s_\star^{(1)} = 1$.

The resulting system is an *infinite hierarchy of coupled ODEs*. Unlike the isotropic case, no finite-dimensional closure exists: the time derivative of order-$k$ statistics depends on order-$(k{+}1)$ statistics, and so on. Understanding the anisotropic dynamics thus requires working with this infinite-dimensional structure.

**Proposition 3.** *The gradient flow dynamics satisfy*

$$\dot{w}_i(t) = 4\lambda_i\big(s_\star - 3s(t)\big)w_i(t) + 8\lambda_i u(t)w_i^\star, \qquad i = 1,\ldots,d, \tag{3.3}$$

$$\dot{s}^{(k)}(t) = 8\,(s_\star - 3s(t))\,s^{(k+1)}(t) + 16\,u(t)\,u^{(k+1)}(t), \tag{3.4}$$

$$\dot{u}^{(k)}(t) = 4\,(s_\star - 3s(t))\,u^{(k+1)}(t) + 8\,u(t)\,s_\star^{(k+1)}. \tag{3.5}$$

*In particular, setting $k = 1$ recovers the dynamics of $s(t)$ and $u(t)$.*

## 3.4 Three-phase structure of the dynamics

The system of ODEs from Proposition 3 involves higher-order moments, preventing a closed-form analysis as in the isotropic case. To guide our theory, we first examine the empirical evolution of the key observables $u(t)$ and $s(t)$; see Figure 2. The trajectories display a *three-phase structure*:

(i) **Phase I: Escape from mediocrity, $s(t) \approx 0$ ($t \leq T_1$).** At initialization, $u(0) \approx 1/\sqrt{d}$, there is almost no correlation with the signal. During this "warm-up" stage, the correlation $u(t)$ escapes exponentially from zero and reaches a small but fixed constant $u(T_1) = \delta > 0$. The overall MSE, however, remains essentially unchanged, since only a few easy directions—those aligned with large eigenvalues—have been learned so far.

(ii) **Phase II: Convergence** $u(t), s(t) \to 1$ ($T_1 \le t < T_2$)**.** Here, the signal alignment strengthens and the summary statistics $u(t), s(t)$ approach their limiting values. Two distinct episodes appear:

- **(IIa) Transition** ($T_1 \le t \le T_1'$)**.** The energy $s(t)$ crosses the critical threshold $1/3$ and stays above it.
- **(IIb) Asymptotic convergence** ($T_1' < t < T_2$)**.** Both $u(t)$ and $s(t)$ approach 1, but convergence is slower than in the isotropic case (see Section F), reflecting the difficulty of learning directions corresponding to small eigenvalues of $Q$.

(iii) **Phase III: Spectral-tail learning** $u(t), s(t) \approx 1$ ($t > T_2$)**.** After $u(t)$ and $s(t)$ have essentially stabilized, the MSE continues to decrease as the flow progressively learns directions associated with the small eigenvalues of $Q$.

This empirical decomposition provides a roadmap for the analysis: by isolating each phase, the infinite-dimensional system becomes amenable to tractable approximations.

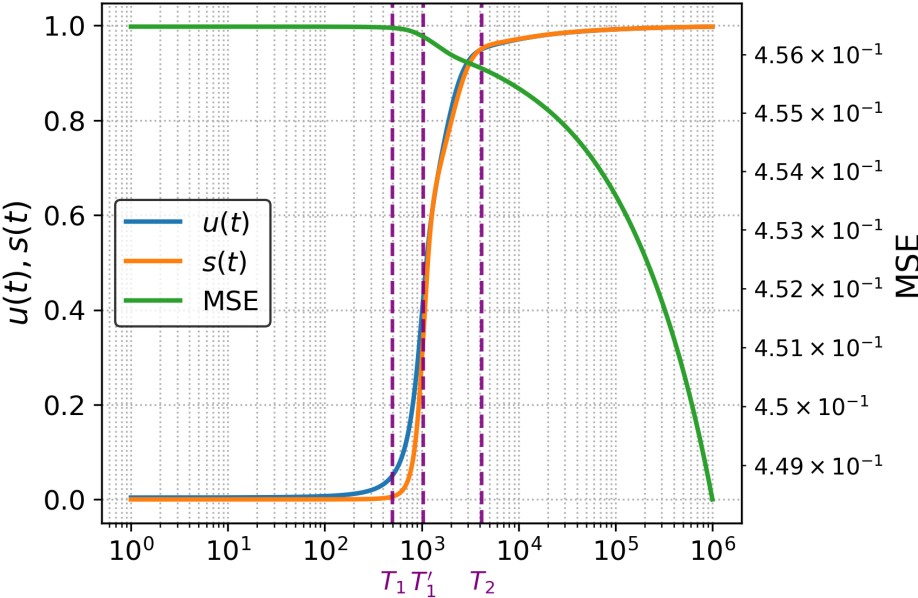

Figure 2: Evolution of the MSE, $u(t)$ and $s(t)$ under population gradient descent (log-log scale). Parameters: $a = 2$, $d = 10^4$, $\eta = 10^{-2}$, $T = 10^6$, $\varepsilon = 0.05$.

## 4 MAIN RESULTS: THREE-PHASE GRADIENT FLOW DYNAMICS

The experiments in Section 3.4 revealed a characteristic trajectory with three successive phases: (i) escape from mediocrity, (ii) convergence of the summary statistics, and (iii) spectral-tail learning. We now show that this qualitative picture admits a rigorous derivation from the gradient flow equations. The following theorems make precise the stopping times, plateau behavior, and tail-driven decay observed empirically.

### 4.1 PHASE I–II: ESCAPE AND APPROXIMATE CONVERGENCE OF THE SUMMARY STATISTICS

The next theorem formalizes escape and approximate convergence, under initialization conditions whose full statement is deferred to Appendix C (see Assumption A1).

**Theorem 1** (Phases I and II)**.** *For $d$ sufficiently large, let $\varepsilon \gtrsim d^{-(a-1)/2}$. Under mild conditions on initialization (e.g. $u(0) \asymp d^{-1/2}$), there exist stopping times $T_1 = O(\log d)$, $T_1' = T_1 + O(1)$, and $T_2 = T_1' + O(\varepsilon^{-2a/(a-1)} \log(1/\varepsilon))$ such that:*

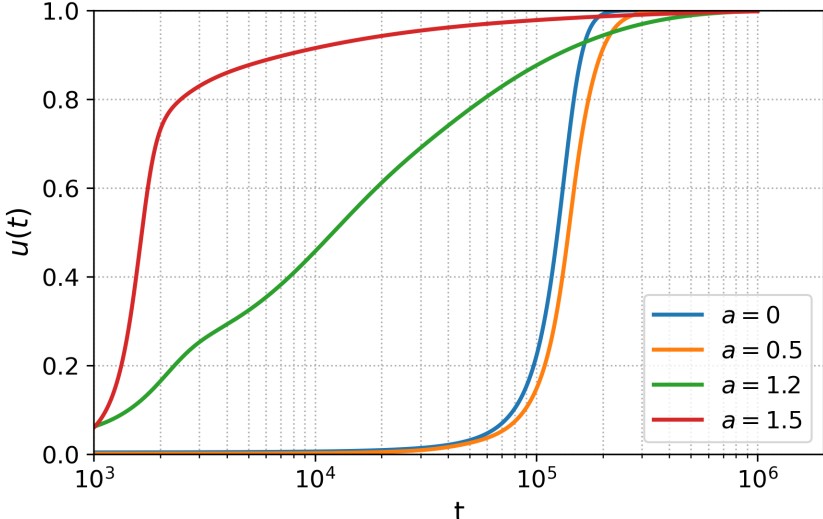

Figure 3: Evolution of the correlation $u(t)$ for different exponent $a$ ($d = 1000$, $\eta = 10^{-3}$).

(i) *There exists a constant $\delta > 0$ such that $|u(T_1)|$, $s(T_1)$, $|u^{(2)}(T_1)| \geq \delta$, and the sign of $u^{(2)}(T_1)$ agrees with that of $u(T_1)$.*

(ii) *There exists a constant $s_0 > 0$ such that $s(t) > 1/3 + s_0$ for all $t \geq T_1'$.*

(iii) *At time $T_2$, both statistics are close to their limits: $|u(T_2)|$, $s(T_2) \in [1 - \varepsilon, 1]$.*

**Remark 3.** *Several features of the theorem are worth noting.*

- ***Initialization and mediocrity.*** *The sign of $u(t)$ is fixed at initialization; w.l.o.g. one may assume it positive by replacing $w_\star$ with $-w_\star$. Under random isotropic initialization, $|u(0)|$ is typically of order $d^{-1/2}$ (as in the isotropic case), hence the escape mediocrity time satisfies $T_1 \asymp \log(1/|u(0)|) = \Theta(\log d)$. In particular, anisotropy does not remove this logarithmic barrier. A table summarizing the different stopping times can be found in Section A.*

- ***Anisotropic vs. isotropic.*** *In isotropic models, $u(0)$ directly controls early growth and convergence is relatively fast; in the anisotropic setting, the Volterra reduction shows that post-plateau escape is controlled by a linear functional of the initialization (via residue/resolvent calculus), and convergence after escape is tail–controlled and slower.*

- ***Accuracy vs. time.*** *For $\varepsilon \gtrsim d^{-(a-1)/2}$ stabilization occurs within $O(\varepsilon^{-2a/(a-1)} \log(1/\varepsilon))$, while demanding $\varepsilon \ll d^{-(a-1)/2}$ leads to even higher time costs (see Phase III analysis).*

**Remark 4.** *The analysis of Phase I reveals that the $u(t)$ grows exponentially at a rate depending on $a$: the convergence is faster when $a$ is large (see Proposition 9 in the appendix, and Figure 3 for a numerical illustration).*

### 4.2   SPECTRAL-TAIL LEARNING AND PHASE III

While Phases I–II describe the evolution of summary statistics $(u, s)$, these quantities do not capture how the estimator $w(t)$ approaches the ground truth entrywise. The mean squared error

$$\text{MSE}(t) = \tfrac{1}{d} \sum_{i=1}^{d} (w_i(t) - w_i^\star)^2$$

is the natural measure of recovery: it is the quantity plotted in Fig. 1, and its log–log decay rates yield the scaling laws highlighted in the introduction. Moreover, the MSE aggregates contributions from all eigen-directions, making it the right observable to expose the effect of the spectral tail.

Up to time $T_2$, while the summary statistics are still converging, the MSE shows almost no decrease and stays essentially at its initial level. The following proposition captures this plateau behavior.

**Proposition 4.** *Let $\sigma_\star^2 = \frac{1}{d} \sum_{i=1}^d (w_i^\star)^2$. Under the assumptions of Theorem 1 we have*

$$\left| \text{MSE}(T_2) - \sigma_\star^2 \right| \lesssim \left( \frac{\varepsilon^{-a}}{d} \right)^{1/3} + \left( \frac{\log d}{d} \right)^{1/3}.$$

After $T_2$, progress is governed by coordinates aligned with the smallest eigenvalues. Let $e_i(t) = w_i(t) - w_i^\star$ be the coordinate error and define $\pi_i = e_i(T_2)^2 / \sum_j e_j(T_2)^2$. The weighted average $\widehat{S}_d(\tau) = \sum_i \pi_i e^{-16\lambda_i \tau}$ describes the spectral decay of the error, while the uniform benchmark $S_d(\tau) = d^{-1} \sum_i e^{-16\lambda_i \tau}$ admits explicit asymptotics. We also set $s_\star^{(2)} = \sum_i \lambda_i^2 (w_i^\star)^2$, $H_{d,a} = \sum_{j=1}^d j^{-a}$, $\beta_d = 16/H_{d,a}$, and $x_d = (\beta_d \tau)^{1/a}$.

**Theorem 2** (Phase III: spectral-tail learning)**.** *Under the assumptions of Theorem 1, the MSE satisfies for every $\tau \geq 0$,*

$$\text{MSE}(T_2 + \tau) = \big(1 + O(\varepsilon)\big) \, \text{MSE}(T_2) \, \widehat{S}_d(\tau) + O\Big(\varepsilon^2 \tau^2 \tfrac{1}{d} s_\star^{(2)}\Big). \tag{4.1}$$

*In particular, $\widehat{S}_d(\tau) = (1 + o(1)) S_d(\tau)$, and $S_d(\tau)$ admits the asymptotics*

$$S_d(\tau) = \begin{cases} 1 - \frac{16}{d} \tau + O(\frac{\tau^2}{d}), & \beta_d \tau \ll 1, \\ 1 - \Gamma(1 - \frac{1}{a}) \frac{x_d}{d} + o(\frac{x_d}{d}), & 1 \ll x_d \ll d, \\ \leq \exp(-\beta_d \tau \, d^{-a}), & x_d \gtrsim d. \end{cases}$$

Thus, the plateau persists until $T_2$, after which the MSE decays at a rate controlled by the spectral tail. The exponent $a$ determines the slope of this decay on log–log scales, accounting for the scaling laws observed in Fig. 1.

## 5    PROOF OUTLINE

We briefly sketch the main ingredients of our analysis, deferring complete proofs to the appendix: Section C (Phase I), Section D (Phase II), and Section E (Phase III).

### 5.1    PHASE I: ESCAPE FROM MEDIOCRITY

In the anisotropic setting, the summary statistics $(u, s)$ do not form a closed system; instead, they are coupled to an infinite hierarchy of correlations. Our strategy is to lift the dynamics to an infinite-dimensional space, solve the system there, and then project back to obtain a closed Volterra representation for $u(t)$, which we can analyze directly.

**Step 1: Infinite-dimensional formulation.**    The core difficulty is the infinite hierarchy of coupled correlations. To tame it, we collect them into the vector $U(t) = (u^{(1)}(t), u^{(2)}(t), \dots)^\top \in \mathcal{H} := C_b^\infty(\mathbb{R}_{\geq 0}; \ell^2)$. Let $B$ be the right-shift operator on sequences, defined by $(Bx)_k := x_{k+1}$ and let $S := (B s_\star^\infty) e_1^\top$ be a rank one operator with $s_\star^\infty = (s_\star^{(1)}, s_\star^{(2)}, \dots)^\top \in \ell^2$ and $e_1 = (1, 0, 0, \dots)^\top$. Then the correlation dynamics (3.5) collapse into the compact operator form

$$\dot{U}(t) = \Big(4\big(1 - 3s(t)\big)B + 8S\Big) U(t). \tag{5.1}$$

This is the key structural observation: the entire infinite system is generated by the shift operator $B$ plus a rank-one perturbation $S$. Applying Duhamel's formula yields the following representation.

**Lemma 1.** *The unique solution $U \in C_b^\infty(\mathbb{R}_{\geq 0}; \ell^2)$ of (5.1) satisfies, for all $t \geq 0$,*

$$U(t) = e^{4B\,\Theta(t)} U_0 + 8 \int_0^t e^{4B\,(\Theta(t) - \Theta(\tau))} (B s_\star^\infty) u^{(1)}(\tau) \, d\tau, \tag{5.2}$$

*where $u^{(1)}(\tau) := \langle e_1, U(\tau) \rangle = u(t)$, and $\Theta(t) := \int_0^t (1 - 3s(\tau)) \, d\tau$.*

**Step 2: Reduction to a Volterra equation.** Let $\delta > 0$ be arbitrarily small, and let $T_1$ denote the stopping time such that $s(t) \leq \delta$ for all $t \leq T_1$. On this time interval we may approximate $\Theta(t) \approx t$. Projecting the representation (5.2) onto $e_1$ then yields the Volterra equation

$$u(t) = a_0(t) + 8 \int_0^t K(t - \tau) u(\tau) d\tau, \qquad (5.3)$$

where

$$a_0(t) = \sum_i w_i(0) w_i^\star \lambda_i e^{4\lambda_i t}, \qquad K(t) = \sum_i (w_i^\star)^2 \lambda_i^2 e^{4\lambda_i t}.$$

Applying the Laplace transform gives $\hat{u}(p) = \frac{\hat{a}_0(p)}{1 - 8\hat{K}(p)}$. The growth of $u(t)$ is therefore governed by the rightmost pole of $\hat{u}(p)$ (see Section C and Chapter VII of Gripenberg et al. (1990)). Computing $\hat{K}(p)$ and solving the equation $1 = 8\hat{K}(p)$ yields the following.

**Lemma 2** (Exponential growth rate). *The correlation $u(t)$ grows at rate $e^{\rho_{true} t}$, where $\rho_{true} > 4\lambda_1$ is the unique positive solution of*

$$1 - 8\widehat{K}(\rho) = 0.$$

**Step 3: Control of the higher-order correlations.** A similar analysis shows that the influence of all higher-order terms is dominated by the leading mode $u(t)$. Without loss of generality, we assume that $u(t)$ grows positively (the case of negative growth is analogous). The key point is that at time $T_1$, the second correlation $u^{(2)}(T_1)$ has the same sign as $u(T_1)$, a fact that will be critical in the analysis of Phase IIa.

**Proposition 5** (Control of higher-order correlations). *For all $k \geq 1$ and all $t \in [0, T_1]$, we have*

$$-\lambda_1^k \sqrt{\tfrac{\log d}{d}} e^{4\lambda_1 t} + 8 s_\star^{(k)} \int_0^t u(\tau) d\tau \leq u^{(k)}(t) \leq \lambda_1^k \sqrt{\tfrac{\log d}{d}} e^{4\lambda_1 t} + \lambda_1^{k-1} (u(t) - u(0)). \tag{5.4}$$

*In particular, for $d$ sufficiently large there exists a constant $c_1 > 0$ such that $u^{(2)}(T_1) \geq c_1$.*

## 5.2 Phase II: Convergence of summary statistics

Since the proof techniques differ, we separate the analysis of Phase IIa and Phase IIb.

### 5.2.1 Analysis of Phase IIa

The first step is to show that $s(t)$, which at time $T_1$ is still small ($s(T_1) = \delta_2'$), must *increase up to the critical threshold* $1/3$. Indeed, from

$$\dot{s}(t) = 8(1 - 3s(t)) s^{(2)}(t) + 16 u(t) u^{(2)}(t),$$

both terms on the right-hand side are positive whenever $s(t) \leq 1/3$, so $s(t)$ is driven upward and crosses $1/3$ in finite time.

The second step establishes *stability beyond the threshold*: once $s(t)$ has passed $1/3$, it cannot fall back below. Close to the boundary, the positive contribution $16 u(t) u^{(2)}(t)$ dominates the negative drift from the first term. A careful comparison shows that $s(t)$ remains uniformly above $1/3 + \delta$ for some constant $\delta > 0$. The detailed proof of these two points is given in Section D.1.

### 5.2.2 Analysis of Phase IIb

We now study the error $\Delta(t) := 1 - u(t) \geq 0$ for $t \geq T_1'$ through the Volterra equation

$$\Delta(t) = b_\Theta(t) + \int_{T_1'}^t K_\Theta(t, \tau) \Delta(\tau) d\tau,$$

where the source term $b_\Theta$ is detailed in Appendix D.2. Since after $T_1'$ we have $\Theta(t) - \Theta(\tau) \leq -s_0(t - \tau)$, the kernel $K_\Theta$ is decreasing, so the integral contribution diminishes over time.

To control the positive part of $b_\Theta$, we split the spectrum at a cutoff $\lambda_c$: directions with $\lambda_i < \lambda_c$ form the *tail*, which is small but slow to learn, while $\lambda_i \geq \lambda_c$ form the *head*, easier but requiring more training time when $\lambda_c$ is small. Balancing these two effects yields the following result.

**Proposition 6.** *Let $\varepsilon \gtrsim d^{-\frac{a-1}{2}}$, and set*

$$T_2(\varepsilon) \;=\; T_1' \;+\; \frac{C}{4s_0}\,\varepsilon^{-\frac{2a}{a-1}}\,\log\frac{1}{\varepsilon},$$

*with $C > 0$ sufficiently large. Then $\Delta(T_2) \leq \varepsilon$.*

**Remark 5.** *The condition $\varepsilon \gtrsim d^{-\frac{a-1}{2}}$ prevents $\varepsilon$ from being too small. Otherwise the required time becomes much larger; for instance, $\varepsilon = d^{-1}$ already forces $T \gtrsim d^a$. Note that the exponent $\frac{2a}{a-1}$ decreases as $a$ increases, so $T_2(\varepsilon)$ is shorter for faster spectral decay. Intuitively, for larger $a$, the lighter tail means fewer small eigenvalues to learn, so the error contracts more rapidly. In contrast, for $a$ close to $1$ the heavy tail creates many slow directions, delaying convergence. This trend is confirmed numerically in Section F.*

### 5.3 Phase III: Spectral-tail learning and the scaling law

After $T_2$, both $u(t)$ and $s(t)$ are close to one, so the coordinate dynamics reduce to

$$\dot{w}_i(t) \;\approx\; 8\lambda_i\big(w_i^\star - w_i(t)\big),$$

whose solution shows exponential relaxation of each $w_i$ toward $w_i^\star$. The resulting MSE is a weighted spectral average $\widehat{S}_d(\tau)$; under the heuristic $\pi_i \approx d^{-1}$, this reduces to the benchmark $S_d(\tau)$, whose asymptotics follow from classical sum–integral comparisons and reveal distinct decay regimes. The approximation error is controlled by a Duhamel (variation-of-constants) representation, together with Phase IIb bounds on $1 - u(t)$ and $1 - s(t)$. A Grönwall argument shows that these corrections remain small, so the full trajectory closely tracks the idealized one. Altogether, this yields the scaling law of Theorem 2, see Section F.1 for an illustration of the approximation.

## 6 Discussion

**Summary.** We analyzed the gradient flow dynamics of anisotropic phase retrieval. Unlike the isotropic case, which reduces to a two-dimensional ODE, the anisotropic setting induces an infinite hierarchy of coupled equations. Our main contribution is to render this structure tractable via a Duhamel–Volterra reduction, revealing a three-phase trajectory: (i) escape from mediocrity, (ii) convergence of summary statistics, and (iii) spectral-tail learning driven by small eigenvalues. This decomposition leads to explicit scaling laws, validated empirically.

**Limitations.** Our analysis assumes Gaussian inputs with a power-law spectrum, simplifying the theory, but this may not be essential for the three-phase phenomenon. In addition, our quantitative results apply only in the gradient flow limit; extending them to discrete-time SGD remains an open problem.

**Future directions.** Several extensions are natural. On the theoretical side, deriving scaling laws for discrete-time SGD, analyzing finite-sample effects, and relaxing distributional assumptions would strengthen the link to practice. On the modeling side, extending beyond quadratic nonlinearities to more general single- and multi-index models could reveal new scaling regimes. More broadly, our findings raise the question of whether analogous phase decompositions and scaling laws govern the training dynamics of wider neural networks, potentially offering a bridge between nonlinear regression models and deep learning theory.

### Acknowledgements

We would like to thank Elisabetta Cornacchia, Florent Krzakala, Gabriele Sicuro, Simone Maria Giancola and Urte Adomaityte for insightful discussions. BL was supported by the French government, managed by the National Research Agency (ANR), under the France 2030 program with the project references "ANR-23-IACL-0008" (PR[AI]RIE-PSAI) and "ANR-25-CE23-5660" (MAPLE), as well as the Choose France - CNRS AI Rising Talents program. MI was supported by JSPS KAKENHI (24K02904), JST CREST (JPMJCR21D2), JST FOREST (JPMJFR216I), and JST BOOST (JPMJBY24A9).

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

The appendix is organized as follows. Section A contains a summary of the different stopping times. Section B contains the proofs of the results stated in Section 3. Section C provides the detailed analysis of Phase I, followed by the analysis of Phase II in Section D and Phase III in Section E. Finally, Section F presents additional numerical experiments.

## A  SUMMARY OF THE DIFFERENT STOPPING TIMES

The definition and properties of the stopping times are summarized in Table 1.

| Phase | Definition | Scaling | Dependence on initialization |
|---|---|---|---|
| I | $|u(t)|$ reaches a fixed $\delta$ at $T_1$ | $T_1 \asymp \log(1/|u(0)|)$ | *Strong*: logarithmic in the initial alignment $|u(0)| \asymp d^{-1/2}$. |
| IIa | $s(t)$ crosses $1/3$ and stabilizes above $1/3 + s_0$ | $T_1' - T_1 = O(1)$ | Only depends on the choice of $\delta$ |
| IIb | $|u(t)|, s(t) \in [1-\varepsilon, 1]$ | $T_2 - T_1' = O\left(\varepsilon^{-\frac{2a}{a-1}} \log \frac{1}{\varepsilon}\right)$ | Only depends on the approximationg error $\varepsilon$, independent of $d$. |

Table 1: Summary of the different stopping times.

## B  PROOF OF SECTION 3

This section provides the proofs of the results stated in Section 3.

**Proposition 1.** *Let $x \sim \mathcal{N}(0, Q) \in \mathbb{R}^d$, where $Q \in \mathbb{R}^{d \times d}$ is a symmetric positive definite diagonal matrix. Let $w, w^\star \in \mathbb{R}^d$. The population loss can be rewritten as*

$$\mathcal{L}(w) = 3\|w\|_Q^4 + 3\|w^\star\|_Q^4 - 4\langle w, w^\star\rangle_Q^2 - 2\|w\|_Q^2 \cdot \|w^\star\|_Q^2 = 3s^2 + 3s_\star^2 - 4u^2 - 2s_\star s$$

*Proof.* Expanding the loss gives

$$\mathcal{L}(w) = \mathbb{E}\big[(x^\top w)^4\big] + \mathbb{E}\big[(x^\top w^\star)^4\big] - 2\mathbb{E}\big[(x^\top w)^2(x^\top w^\star)^2\big].$$

**Step 1. Fourth moment of a Gaussian linear form.**   Since $x \sim \mathcal{N}(0, Q)$, the scalar $x^\top w$ is Gaussian with variance $w^\top Q w = \|w\|_Q^2$. Its fourth moment is

$$\mathbb{E}\big[(x^\top w)^4\big] = 3\,(w^\top Q w)^2 = 3\|w\|_Q^4.$$

By the same reasoning,

$$\mathbb{E}\big[(x^\top w^\star)^4\big] = 3\|w^\star\|_Q^4.$$

**Step 2. Mixed fourth moment.**   Expanding the product gives

$$(x^\top w)^2(x^\top w^\star)^2 = \sum_{i,j,k,\ell} x_i x_j x_k x_\ell \, w_i w_j w_k^\star w_\ell^\star.$$

For Gaussian $x$, Wick's formula expresses the fourth moment as a sum over pairings:

$$\mathbb{E}[x_i x_j x_k x_\ell] = \mathbb{E}[x_i x_j]\mathbb{E}[x_k x_\ell] + \mathbb{E}[x_i x_k]\mathbb{E}[x_j x_\ell] + \mathbb{E}[x_i x_\ell]\mathbb{E}[x_j x_k].$$

Since $Q = \mathrm{diag}(\lambda_1, \ldots, \lambda_d)$, $\mathbb{E}[x_i x_j] = \lambda_i \delta_{ij}$.

Evaluating the three pairings:

- Pairing $(i,j)(k,\ell)$ contributes

$$\sum_{i,k} \lambda_i \lambda_k \, w_i^2 (w_k^\star)^2 = \|w\|_Q^2 \cdot \|w^\star\|_Q^2.$$

- Pairing $(i, k)(j, \ell)$ contributes

$$\sum_{i,j} \lambda_i \lambda_j \, w_i w_j w_i^\star w_j^\star = \Big( \sum_i \lambda_i w_i w_i^\star \Big)^2 = \langle w, w^\star \rangle_Q^2.$$

- Pairing $(i, \ell)(j, k)$ contributes the same quantity

$$\langle w, w^\star \rangle_Q^2.$$

Summing up,

$$\mathbb{E}\big[(x^\top w)^2 (x^\top w^\star)^2\big] = \|w\|_Q^2 \cdot \|w^\star\|_Q^2 + 2\langle w, w^\star \rangle_Q^2.$$

**Step 3. Final expression.**   Putting everything together,

$$\mathcal{L}(w) = 3\|w\|_Q^4 + 3\|w^\star\|_Q^4 - 2\Big( \|w\|_Q^2 \cdot \|w^\star\|_Q^2 + 2\langle w, w^\star \rangle_Q^2 \Big)$$
$$= 3\|w\|_Q^4 + 3\|w^\star\|_Q^4 - 2\|w\|_Q^2 \cdot \|w^\star\|_Q^2 - 4\langle w, w^\star \rangle_Q^2.$$

$\square$

**Proposition 2.** *The population gradient and Hessian of the loss $\mathcal{L}$ are given by*

$$\nabla \mathcal{L}(w) = 12sQw - 4s_\star Qw - 8uQw^\star, \tag{3.1}$$
$$\nabla^2 \mathcal{L}(w) = 24(Qw)(Qw)^\top + (12s - 4s_\star)Q - 8(Qw^\star)(Qw^\star)^\top. \tag{3.2}$$

*The set of critical points is $\{0, \ w^\star, \ -w^\star\} \cup \{w : (u, s) = (0, s_\star/3)\}$. Among them, $0$ is a strict local maximum, $\pm w^\star$ are strict global minima, and every $w$ with $(u, s) = (0, s_\star/3)$ is a saddle point.*

*Proof.* The proof proceeds in three steps: computing the gradient, identifying the critical points, and classifying them via the Hessian.

**Gradient.**   To compute the gradient, we use the closed-form expression $\mathcal{L}(w) = 3s^2 + 3(s_\star)^2 - 2ss_\star - 4u^2$, apply the chain rule and use $\nabla s = 2Qw$ and $\nabla u = Qw^\star$ to get

$$\nabla \mathcal{L}(w) = 6s\nabla s - 2s_\star \nabla s - 8u\nabla u = (12s - 4s_\star)Qw - 8uQw^\star,$$

which is (3.1).

**Critical points.**   Setting $\nabla \mathcal{L}(w) = 0$ and using $Q \succ 0$ leads to

$$(12s - 4s_\star)w - 8uw^\star = 0.$$

– *Case 1:* If $12s - 4s_\star \neq 0$, then $w$ must lie in $\operatorname{span} w^\star$. Write $w = \alpha w^\star$; then $s = \alpha^2 s_\star$ and $u = \alpha s_\star$. Substituting gives

$$(12\alpha^2 - 4)\alpha w^\star = 8\alpha w^\star \quad \Longleftrightarrow \quad 12\alpha^3 - 12\alpha = 0 \quad \Longleftrightarrow \quad \alpha \in \{0, \pm 1\}.$$

Hence, we obtain the critical points $0$ and $\pm w^\star$.

– *Case 2:* If $12s - 4s_\star = 0$, then necessarily $s = s_\star/3$. The gradient condition reduces to $-8uw^\star = 0$, i.e. $u = 0$. Thus every $w$ with $(u, s) = (0, s_\star/3)$ is a critical point.

**Hessian.**   Differentiating (3.1), and recalling that $\nabla s = 2Qw$ and $\nabla u = Qw^\star$, we obtain

$$\nabla^2 \mathcal{L}(w) = 24\,(Qw)(Qw)^\top + (12s - 4s_\star)Q - 8\,(Qw^\star)(Qw^\star)^\top,$$

which is (3.2).

**Characterization of the critical points.** At $w = 0$ (so $s = 0$, $u = 0$),

$$\nabla^2 \mathcal{L}(0) = -4s_\star Q - 8\,(Qw^\star)(Qw^\star)^\top,$$

which is strictly negative definite since $Q \succ 0$ and the rank-one term is negative semidefinite. Thus $w = 0$ is a strict local maximum since $\mathcal{L}(0) > \mathcal{L}(\pm w^\star) = 0$.

At $w = \pm w^\star$ (so $s = s_\star$, $u = \pm s_\star$),

$$\nabla^2 \mathcal{L}(\pm w^\star) = 24\,(Qw^\star)(Qw^\star)^\top + (12s_\star - 4s_\star)Q - 8\,(Qw^\star)(Qw^\star)^\top = 16\,(Qw^\star)(Qw^\star)^\top + 8s_\star Q \succ 0,$$

hence both $\pm w^\star$ are strict local (and, by the loss formula, global) minima.

Finally, for any $w$ with $(u, s) = (0, s_\star/3)$, the Hessian reduces to

$$\nabla^2 \mathcal{L}(w) = 24(Qw)(Qw)^\top - 8(Qw^\star)(Qw^\star)^\top.$$

Since $Q \succ 0$, $w^\star \neq 0$, and $w^\top Q w^\star = 0$, $s = w^\top Q w = \frac{1}{3}s_\star = \frac{1}{3}(w^\star)^\top Q w^\star$, we have

$$w^\top \nabla^2 \mathcal{L}(w)\, w = w^\top [24Qww^\top Q - 8Qw^\star(w^\star)^\top Q]w = 24(w^\top Q w)^2 = 8[(w^\star)^\top Q w^\star]^2 > 0,$$
$$(w^\star)^\top \nabla^2 \mathcal{L}(w)\, w^\star = (w^\star)^\top [24Qww^\top Q - 8Qw^\star(w^\star)^\top Q](w^\star) = -8[(w^\star)^\top Q w^\star]^2 < 0.$$

It follows that $\nabla^2 \mathcal{L}(w)$ must necessarily have both positive and negative directions. Thus, this critical point is a saddle. $\qquad\square$

**Proposition 3.** *The gradient flow dynamics satisfy*

$$\dot{w}_i(t) = 4\lambda_i\big(s_\star - 3s(t)\big)w_i(t) + 8\lambda_i u(t)w_i^\star, \qquad i = 1, \ldots, d, \tag{3.3}$$

$$\dot{s}^{(k)}(t) = 8\,(s_\star - 3s(t))\, s^{(k+1)}(t) + 16\, u(t)\, u^{(k+1)}(t), \tag{3.4}$$

$$\dot{u}^{(k)}(t) = 4\,(s_\star - 3s(t))\, u^{(k+1)}(t) + 8\, u(t)\, s_\star^{(k+1)}. \tag{3.5}$$

*In particular, setting $k = 1$ recovers the dynamics of $s(t)$ and $u(t)$.*

*Proof.* We first derive the component-wise ODE, then deduce the dynamics for the summary statistics $s(t)$ and $u(t)$. Recall from the previous computation that the population gradient flow $\dot{w}(t) = -\nabla \mathcal{L}(w(t))$ satisfies

$$\dot{w}(t) = 4(s_\star - 3s)Qw + 8u\, Qw^\star. \tag{B.1}$$

**Component-wise dynamics.** With $Q = \mathrm{diag}(\lambda_1, \ldots, \lambda_d)$, taking the $i$-th coordinate in (B.1) yields

$$\dot{w}_i(t) = 4(s_\star - 3s)\lambda_i\, w_i(t) + 8u\, \lambda_i\, w_i^\star.$$

**Dynamics for $s(t)$.** Differentiate $s(t) = w(t)^\top Q w(t)$ along the flow:

$$\dot{s}(t) = 2\,\dot{w}(t)^\top Q w(t).$$

Using (B.1) and defining

$$s^{(2)}(t) := w(t)^\top Q^2 w(t), \qquad u^{(2)}(t) := w(t)^\top Q^2 w^\star,$$

we obtain

$$\dot{s}(t) = 8(s_\star - 3s)\, s^{(2)}(t) + 16u\, u^{(2)}(t).$$

**Dynamics for $u(t)$.** Differentiate $u(t) = w(t)^\top Q w^\star$:

$$\dot{u}(t) = \dot{w}(t)^\top Q w^\star.$$

Using (B.1) and letting $s_\star^{(2)} := w^{\star\top} Q^2 w^\star$, we get

$$\dot{u}(t) = 4(s_\star - 3s)\, u^{(2)}(t) + 8u\, s_\star^{(2)}.$$

These identities establish the stated ODEs for $s(t)$ and $u(t)$.

**Higher-order moments.** The same computation applies to $s^{(k)}(t) = w(t)^\top Q^k w(t)$ and $u^{(k)}(t) = w(t)^\top Q^k w^\star$, leading to

$$\dot{s}^{(k)}(t) = 8(s_\star - 3s(t))\, s^{(k+1)}(t) + 16u(t)\, u^{(k+1)}(t),$$
$$\dot{u}^{(k)}(t) = 4(s_\star - 3s(t))\, u^{(k+1)}(t) + 8u(t)\, s_\star^{(k+1)}.$$

$\square$

## C    ANALYSIS OF PHASE I: ESCAPING MEDIOCRITY

In this section, we provide the detailed proofs for the analysis of Phase I outlined in Section 5.1. We begin by formalizing the infinite-dimensional ODE in a suitable Banach space (Section C.1). We then derive a closed-form solution (Section C.2), show that $u(t)$ satisfies a Volterra integral equation by projection (Section C.3), and analyze its asymptotic behavior. Finally, we study higher-order correlation terms (Section C.4), showing that their dynamics are primarily driven by $u(t)$.

### C.1    STEP 0: PRELIMINARIES ON BANACH-VALUED ODES

Before entering the main steps of the proof, we formalize the Banach space on which the ODE (C.9) is defined. Throughout the following, $\mathcal{L}(w)$ refers to the loss function in Proposition 1. Recall that we define $U(t) = (u^{(1)}(t), u^{(2)}(t), \dots)^\top$, where $u^{(k)}(t) = w(t)^\top Q^k w^\star$. We show that

$$U \in \mathcal{H} := C_b^\infty(\mathbb{R}_{\geq 0}; \ell^2),$$

where $\ell^2 = \{(a_k)_{k \geq 1} : \sum_{k=1}^\infty a_k^2 < \infty\}$ is the Hilbert space of square-summable sequences. Thus $U$ is a bounded, smooth $\ell^2$-valued function on $\mathbb{R}_{\geq 0}$.

**Lemma 3** (Coercivity of the loss). *The loss function $\mathcal{L}$ in Proposition 1 is coercive, that is* $\lim_{\|w\| \to \infty} \mathcal{L}(w) = \infty$.

*Proof.* By the Cauchy-Schwarz inequality

$$\mathcal{L}(w) = 3\|w\|_Q^4 + 3\|w^\star\|_Q^4 - 4\langle w, w^\star \rangle_Q^2 - 2\|w\|_Q^2\|w^\star\|_Q^2$$
$$\geq 3\|w\|_Q^4 + 3\|w^\star\|_Q^4 - 6\|w\|_Q^2\|w^\star\|_Q^2 = 3(\|w\|_Q^2 - \|w^\star\|_Q^2)^2,$$

from which it clearly follows that $\lim_{\|w\| \to \infty} \mathcal{L}(w) = \infty$. $\square$

**Lemma 4** (Local Lipschitz continuity of the gradient). *Consider the function $Z : \mathbb{R}^d \mapsto \mathbb{R}^d$ defined by*

$$Z(w) = 3\|w\|_Q^2 Qw - \|w^*\|_Q^2 Qw - 2(w^\top Qw^*)Qw^*. \tag{C.1}$$

*For all $w_1, w_2 \in \mathbb{R}^d$,*

$$\|Z(w_1) - Z(w_2)\| \leq 3\|Q\|^2\|w_1 - w_2\|\Big[(\|w_1\| + \|w_2\|)\|w_1\| + \|w_2\|^2 + \|w^*\|^2\Big]. \tag{C.2}$$

*In particular, for $\|w_1\| \leq M$, $\|w_2\| \leq M$, with $M > 0$,*

$$\|Z(w_1) - Z(w_2)\| \leq 3(3M^2 + \|w^*\|^2)\|Q\|^2\|w_1 - w_2\|. \tag{C.3}$$

*Proof.*

$$\|Z(w_1) - Z(w_2)\|$$
$$\leq 3\|\, \|w_1\|_Q^2 Qw_1 - \|w_2\|_Q^2 Qw_2\| + \|w^*\|_Q^2\|Qw_1 - Qw_2\| + 2\|(w_1 - w_2)^\top Qw^* Qw^*\|$$
$$\leq 3\|\, \|w_1\|_Q^2 - \|w_2\|_Q^2\|\, \|Qw_1\| + 3\|w_2\|_Q^2\|Qw_1 - Qw_2\|$$
$$\quad + \|w^*\|_Q^2\|Q\|\, \|w_1 - w_2\| + 2\|w_1 - w_2\|\, \|Qw^*\|^2$$
$$\leq 3\|w_1 - w_2\|_Q[\|w_1\|_Q + \|w_2\|_Q]\, \|Qw_1\| + 3\|w_2\|_Q^2\|Q\|\, \|w_1 - w_2\|$$
$$\quad + \|w^*\|_Q^2\|Q\|\, \|w_1 - w_2\| + 2\|w_1 - w_2\|\, \|Qw^*\|^2$$
$$= \|w_1 - w_2\|[3(\|w_1\| + \|w_2\|)\|Q\|^2\|w_1\| + 3\|w_2\|^2\|Q\|^2 + \|w^*\|_Q^2\|Q\| + 2\|Qw^*\|^2]$$
$$\leq 3\|Q\|^2\|w_1 - w_2\|[(\|w_1\| + \|w_2\|)\|w_1\| + \|w_2\|^2 + \|w^*\|^2].$$

$\square$

We recall the classical Picard-Lindelöf Theorem for the existence and uniqueness of the solution of the initial value problem

$$\dot{x} = f(t, x), \quad x(t_0) = x_0, \tag{C.4}$$

where $f \in C(U; \mathbb{R}^d)$, $U \subset \mathbb{R}^{d+1}$ is an open subset, and $(t_0, x_0) \in U$, see e.g. Teschl (2012) (Theorem 2.2, Lemma 2.3).

**Theorem 3** (**Picard-Lindelöf**). *Let $f \in C^k(U; \mathbb{R}^d)$, $k \geq 0$, where $U \subset \mathbb{R}^{d+1}$ is an open subset and $(t_0, x_0) \in U$. Assume that $f$ is locally Lipschitz continuous in the second argument, uniformly with respect to the first argument, i.e. $\forall V \subset U$, $V$ compact, $\exists L = L(V) > 0$ such that*

$$||f(t, x_1) - f(t, x_2)|| \leq L||x_1 - x_2||, \ \forall (t, x_1), (t, x_2) \in V. \tag{C.5}$$

*Then there exists $\epsilon > 0$ such that the initial value problem (C.4) possesses a unique solution $x^*(t) \in C^{k+1}(I; \mathbb{R}^d)$, where $I = [t_0 - \epsilon, t_0 + \epsilon]$.*

**Lemma 5** (Well-posedness of the gradient flow). *The gradient flow on $\mathbb{R} \times \mathbb{R}^d$*

$$\begin{cases} \dot{w}(t) = -\nabla\mathcal{L}(w(t)), \\ \quad w(0) = w_0 \in \mathbb{R}^d \end{cases} \tag{C.6}$$

*has a unique solution $w \in C_b^\infty(\mathbb{R}_{\geq 0}; \mathbb{R}^d)$ $\forall w_0 \in \mathbb{R}^d$.*

*Proof.* By Lemma 4, the gradient $\nabla_w \mathcal{L}(w)$ is locally Lipschitz on $\mathbb{R}^d$, thus by Picard-Lindelöf Theorem there exists a unique local solution $w \in C^\infty(I; \mathbb{R}^d)$ on $I = [-\epsilon, \epsilon]$ for some $\epsilon > 0$, since $\mathcal{L} \in C^\infty(\mathbb{R}^{d+1}; \mathbb{R}^d)$. The interval $I$ can in fact be extended to all of $\mathbb{R}$. By differentiating $\mathcal{L}(w(t))$ we have, by the chain rule,

$$\frac{d\mathcal{L}(w(t))}{dt} = (\dot{w}(t))^\top \nabla\mathcal{L}(w(t)) = -||\nabla\mathcal{L}(w(t))||^2 \leq 0.$$

Thus $\mathcal{L}(w(t))$ is decreasing along the flow $w(t)$ and therefore $w(t)$ always lies in the level set $L_{w_0}(\mathcal{L}) = \{w \in \mathbb{R}^d : \mathcal{L}(w) \leq \mathcal{L}(w_0)\}$. Since $\mathcal{L}(w)$ is coercive by Lemma 3, $L_{w_0}(\mathcal{L})$ must necessarily be bounded. Thus Picard-Lindelöf Theorem can be repeatedly applied to extend $I$ to all of $\mathbb{R}_{\geq 0}$. Since $\mathcal{L}(w)$ is bounded below by zero, the limit $\lim_{t \to \infty} \mathcal{L}(w(t))$ necessarily exists. Let $B_{\mathbb{R}^d}(0, M)$ be the smallest ball in $\mathbb{R}^d$, centered at the origin, such that $L_{w_0}(\mathcal{L}) \subset B_{\mathbb{R}^d}(0, M)$, then $||w(t)|| \leq M$ $\forall t \geq 0$. Thus $w(t) \in C_b^\infty(\mathbb{R}_{\geq 0}; \mathbb{R}^d)$. $\qquad\square$

**Lemma 6.** *Let $w$ be the unique global solution of the gradient flow (C.6). Define $U : \mathbb{R}_{\geq 0} \to \mathbb{R}^\infty$ by $U(t) = (u^{(1)}(t), u^{(2)}(t), \dots)^\top$, where $u^{(k)}(t) = w(t)^\top Q^k w^\star$. Then $U \in C_b^\infty(\mathbb{R}_{\geq 0}; \ell^2)$.*

*Proof.* By the assumption that $\text{tr}(Q) = 1$ and $\lambda_1 \geq \cdots \geq \lambda_d > 0$, we have $||Q|| = \lambda_1 < 1$. For each fixed $t$, we have

$$\sum_{k=1}^\infty |u^{(k)}(t)|^2 = \sum_{k=1}^\infty (w(t)^\top Q^k w^\star)^2 \leq ||w(t)||^2 ||w^\star||^2 \sum_{k=1}^\infty ||Q^k||^2$$

$$\leq ||w(t)||^2 ||w^\star||^2 \sum_{k=1}^\infty ||Q||^{2k} = ||w(t)||^2 ||w^\star||^2 \frac{||Q||^2}{1 - ||Q||^2} < \infty.$$

Since $w$ is bounded, for each fixed $t$, we thus have $U(t) \in \ell^2$, and

$$||U||_\infty = \sup_{t \geq 0} ||U(t)|| \leq ||w||_\infty ||w^\star|| \frac{||Q||}{\sqrt{1 - ||Q||^2}}.$$

As $w$ is bounded and smooth, so is $U$, hence $U \in C_b^\infty(\mathbb{R}_{\geq 0}; \ell^2)$. $\qquad\square$

## C.2 STEP 1: INFINITE-DIMENSIONAL ODE FOR THE MOMENT SYSTEM

Contrary to the isotropic setting where the dynamics only depend on two ODEs involving $u(t)$ and $s(t)$, in the anisotropic setting, higher order correlations $u^{(k)}$ and $s^{(k)}$ appear. We first derive ODEs for higher-order correlation terms and then solve the infinite-dimensional ODE system.

Differentiating $u^{(k)}(t) = w(t)^\top Q^k w_\star$ along (3.1) gives, for $k \geq 1$,

$$\dot{u}^{(k)}(t) = 4\big(s_\star - 3s(t)\big)u^{(k+1)}(t) + 8\,u(t)s_\star^{(k+1)}. \tag{C.7}$$

Using the normalization $s_\star = 1$, we obtain

$$\dot{u}^{(k)}(t) = 4\big(1 - 3s(t)\big)u^{(k+1)}(t) + 8\,u(t)s_\star^{(k+1)}. \tag{C.8}$$

**Infinite-dimensional ODE**. Let $U(t) = (u^{(1)}(t), u^{(2)}(t), \dots)^\top \in \mathbb{R}^\infty$. By Lemma 6, we have $U \in \mathcal{H} = C_b^\infty(\mathbb{R}_{\geq 0}; \ell^2)$. Consider $B : \ell^2 \to \ell^2$, the right-shift operator on sequences in $\ell^2$, defined by $(Bx)_k := x_{k+1}$ for $x = (x_k)_{k\in\mathbb{N}} \in \ell^2$. Let $S := (Bs_\star^\infty)\, e_1^\top : \ell^2 \to \ell^2$ be a rank one operator with $s_\star^\infty = (s_\star^{(1)}, s_\star^{(2)}, \dots)^\top \in \ell^2$, $Bs_\star^\infty = (s_\star^{(2)}, s_\star^{(3)}, \dots)^\top \in \ell^2$ and $e_1 = (1, 0, 0, \dots)^\top$ is the first basis vector in the canonical orthonormal basis for $\ell^2$. Then (C.8) admits the equivalent formulation

$$\dot{U}(t) = \Big(4\big(1 - 3s(t)\big)B + 8S\Big) U(t). \tag{C.9}$$

This is an infinite-dimensional ODE, with $U$ belonging to the Banach space $C_b^\infty(\mathbb{R}_{\geq 0}; \ell^2)$ of smooth, bounded functions with values in the Hilbert space $\ell^2$. In the following, we seek to solve (C.9).

We recall that on a Banach space $\mathcal{B}$, with $\mathcal{L}(\mathcal{B})$ denoting the Banach space of bounded linear operators on $\mathcal{B}$, the exponential operator $e^A$ is well-defined $\forall A \in \mathcal{L}(\mathcal{B})$, with

$$e^A = \sum_{k=0}^\infty \frac{A^k}{k!} : \mathcal{L}(\mathcal{B}) \to \mathcal{L}(\mathcal{B}), \tag{C.10}$$

$$||e^A|| \leq \sum_{k=0}^\infty \frac{||A||^k}{k!} = e^{||A||} < \infty. \tag{C.11}$$

**Lemma 7** (Shift exponential). *For any $\theta \in \mathbb{R}$ and $x \in \ell^2$,*

$$\big(e^{\theta B} x\big)_k = \sum_{m=0}^\infty \frac{\theta^m}{m!}\, x_{k+m}. \tag{C.12}$$

*Proof.* Because $B$ is bounded on $\ell^2$ with $\|B\| \leq 1$, the exponential series for $e^{\theta B}$ converges in operator norm for every $\theta \in \mathbb{R}$, and (C.12) follows from $B^m x$ having coordinates $(B^m x)_k = x_{k+m}$. $\qquad\square$

**Proposition 7.** *Define the function $\Theta : \mathbb{R}_{\geq 0} \to \mathbb{R}$ by*

$$\Theta(t) := \int_0^t (1 - 3s(\tau))\, d\tau. \tag{C.13}$$

*The initial value problem on $\ell^2$*

$$\dot{U}(t) = \Big(4\big(1 - 3s(t)\big)B + 8S\Big) U(t), \; U(0) = U_0 \in \ell^2, \tag{C.14}$$

*has a unique solution $U \in C_b^\infty(\mathbb{R}_{\geq 0}; \ell^2)$, satisfying, $\forall t \geq 0$,*

$$U(t) = e^{4B\,\Theta(t)}\, U_0 + 8 \int_0^t e^{4B\,(\Theta(t)-\Theta(\tau))}\, S\, U(\tau)\, d\tau. \tag{5.2}$$

*Equivalently, since $S = (Bs_\star^\infty)\, e_1^\top$, (5.2) can be written as*

$$U(t) = e^{4B\,\Theta(t)}\, U_0 + 8 \int_0^t e^{4B\,(\Theta(t)-\Theta(\tau))}\, (Bs_\star^\infty)\, u^{(1)}(\tau)\, d\tau,$$

*where $u^{(1)}(\tau) := \langle e_1, U(\tau) \rangle$.*

To prove Proposition 7, we apply the following result on ODEs in Banach space.

**Theorem 4** (Pazy (2012), Theorem 5.1). *Let $\mathcal{B}$ be a Banach space. Assume that $A(t) : \mathcal{B} \to \mathcal{B}$ is a bounded linear operator $\forall t \in [0, T]$, $0 \le T < \infty$, and that the map $t \to A(t)$ is continuous in the uniform operator topology, with $\max_{t \in [0,T]} ||A(t)|| < \infty$. Then $\forall u_0 \in \mathcal{B}$, the initial value problem*

$$\begin{cases} \frac{du}{dt} = A(t)u(t), & 0 \le t_0 \le t \le T < \infty, \\ u(t_0) = u_0 \in \mathcal{B}. \end{cases} \tag{C.15}$$

*has a unique solution $u \in C^1([t_0, T]; \mathcal{B})$.*

**Proof of Proposition 7.** Let $a : \mathbb{R}_{\ge 0} \to \mathbb{R}$ be defined by $a(t) = 1 - 3s(t) = 1 - 3w(t)^\top Q w(t)$. Since $w \in C_b^\infty(\mathbb{R}_{\ge 0}; \mathbb{R}^d)$ by Lemma 5, we have $s \in C_b^\infty(\mathbb{R}_{\ge 0}; \mathbb{R})$, $a \in C_b^\infty(\mathbb{R}_{\ge 0}; \mathbb{R})$, with

$$|a(t)| \le 1 + 3||Q|| \, ||w(t)||^2, \quad ||a||_\infty \le 1 + 3||Q|| \, ||w||_\infty^2,$$
$$|a(t_1) - a(t_2)| = 3|(w(t_1)^\top Q w(t_1) - w(t_2)^\top Q w(t_2)|$$
$$\le 3||Q|| \, ||w(t_1) - w(t_2)|| \, [||w(t_1)|| + ||w(t_2)||].$$

Let $A : \mathbb{R}_{\ge 0} \to \mathcal{L}(\ell^2)$ be defined by $A(t) = 4a(t)B + 8S$, then $A(t) : \ell^2 \to \ell^2$ is a bounded linear operator $\forall t \ge 0$, with

$$||A(t)|| \le 4||B|| \, (1 + 3||Q|| \, ||w(t)||^2) + 8||S||$$
$$||A||_\infty \le 4||B|| \, (1 + 3||Q|| \, ||w||_\infty^2) + 8||S||,$$
$$||A(t_1) - A(t_2)|| \le 12||B|| \, ||Q|| \, ||w(t_1) - w(t_2)|| \, [||w(t_1)|| + ||w(t_2)||]$$
$$\le 24||B|| \, ||Q|| \, ||w||_\infty ||w(t_1) - w(t_2)||.$$

Thus $A(t)$ is continuous in the uniform operator topology and $\sup_{t \ge 0} ||A(t)|| < \infty$. By Theorem 4, the initial value problem (C.14) has a unique solution $U \in C^1([0, T]; \ell^2)$, $\forall 0 < T < \infty$. Since $s(t)$ is smooth, it follows immediately that $U \in C^\infty([0, T]; \ell^2)$, $\forall 0 < T < \infty$. Consider the following function $U : \mathbb{R}_{\ge 0} \to \ell^2$ satisfying

$$U(t) = e^{4B\,\Theta(t)} U_0 + 8 \int_0^t e^{4B\,(\Theta(t) - \Theta(\tau))} S\,U(\tau)\,d\tau$$

$$= e^{4B\,\Theta(t)} U_0 + 8e^{4B\,\Theta(t)} \int_0^t e^{-4B\,\Theta(\tau)} S\,U(\tau)\,d\tau,$$

where we recall that $\Theta(t) = \int_0^\top a(\tau)\,d\tau = \int_0^\top (1 - 3s(\tau))\,d\tau$, with $\dot{\Theta}(t) = a(t)$. Differentiating $U(t)$ on both sides gives, via the product rule,

$$\dot{U}(t) = 4\,a(t)B\,e^{4B\,\Theta(t)} U_0 + 32\,a(t)\,B\,e^{4B\,\Theta(t)} \int_0^t e^{-4B\,\Theta(\tau)} S\,U(\tau)\,d\tau + 8\,S\,U(t)$$

$$= 4\,a(t)B\,U(t) + 8\,S\,U(t),$$

which is precisely the differential equation in (C.14). Since $a \in C_b^\infty(\mathbb{R}_{\ge 0}; \mathbb{R})$, it follows that $U \in C^\infty(\mathbb{R}_{\ge 0}; \ell^2)$. Therefore, the unicity of solution implies that $U$ must be the unique solution of (C.14) and we must have $U \in C_b^\infty(\mathbb{R}_{\ge 0}; \ell^2)$. $\qquad\square$

## C.3 STEP 2: VOLTERRA REDUCTION

Taking the first coordinate of (5.2) yields a Volterra equation for $u(t) = u^{(1)}(t)$.

**Proposition 8** (Volterra equation). *Define*

$$a_\Theta(t) := \left(e^{4B\,\Theta(t)} U(0)\right)_1 = \sum_{m \ge 0} \frac{(4\Theta(t))^m}{m!}\, u^{(1+m)}(0), \tag{C.16}$$

$$K_\Theta(t) := \left(e^{4B\,\Theta(t)} B s_\star^\infty\right)_1 = \sum_{m \ge 0} \frac{(4\Theta(t))^m}{m!}\, s_\star^{(2+m)}. \tag{C.17}$$

*Then $u$ satisfies*

$$u(t) \;=\; a_\Theta(t) \;+\; 8 \int_0^t K_\Theta(t-\tau)\, u(\tau)\, d\tau. \tag{C.18}$$

*Moreover,*

$$a_\Theta(t) = \sum_i w_i(0) w_i^\star \lambda_i\, e^{4\lambda_i \Theta(t)}, \qquad K_\Theta(t) = \sum_i (w_i^\star)^2 \lambda_i^2\, e^{4\lambda_i \Theta(t)}. \tag{C.19}$$

*Proof.* Equation (C.18) is the first coordinate of (5.2). The spectral forms follow by expanding $u^{(1+m)}(0) = \sum_i \lambda_i^{1+m} w_i(0) w_i^\star$ and $s_\star^{(2+m)} = \sum_i (w_i^\star)^2 \lambda_i^{2+m}$ and summing the exponential series. □

To analyze the growth of $u(t)$, we employ the Laplace transform together with residue calculus; see (Gripenberg et al., 1990, Chapters 2, 3, and 7) for a detailed exposition. We first illustrate the method in the ideal case $\Theta(t) = t$, and then explain how the analysis extends to the approximated case.

### C.3.1 IDEAL CASE: $\Theta(t) = t$

Fix $b > 0$ (e.g. $b = 4$ if $\Theta(t) = t$) and consider the scalar Volterra equation of the second kind

$$\alpha_b(t) \;=\; a_\Theta(t) \;+\; 8 \int_0^t K_b(t-s)\, \alpha_b(s)\, ds, \qquad K_b(t) = \sum_{i \geq 1} (w_i^\star)^2 \lambda_i^2\, e^{b\lambda_i t}, \quad t \geq 0, \tag{C.20}$$

where $\lambda_1 = \max_i \lambda_i$, and the series defining $K_b$ (hence $\widehat{K}_b$) converges for $\Re s > b\lambda_1$. Define

$$D_b(s) \;:=\; 1 - 8\,\widehat{K}_b(s) \;=\; 1 - 8 \sum_{i \geq 1} \frac{(w_i^\star)^2 \lambda_i^2}{s - b\lambda_i}, \qquad \Re s > b\lambda_1.$$

**Lemma 8.** *Assume $(w_1^\star)^2 \lambda_1^2 > 0$. There is a unique real $\rho(b) > b\lambda_1$ such that*

$$D_b(\rho(b)) = 0 \quad \Longleftrightarrow \quad 1 \;=\; 8 \sum_{i=1}^d \frac{(w_i^\star)^2 \lambda_i^2}{\rho(b) - b\lambda_i}. \tag{C.21}$$

*Moreover, the map $b \mapsto \rho(b)$ is strictly increasing and continuous.*

*Proof.* For real $\rho > b\lambda_1$, define

$$F_b(\rho) := 8\,\widehat{K}_b(\rho) = 8 \sum_{i=1}^d \frac{(w_i^\star)^2 \lambda_i^2}{\rho - b\lambda_i}.$$

Then $F_b$ is continuous and strictly decreasing on $(b\lambda_1, \infty)$. Since $(w_1^\star)^2 \lambda_1^2 > 0$, we have

$$\lim_{\rho \downarrow b\lambda_1} F_b(\rho) = +\infty, \qquad \lim_{\rho \to \infty} F_b(\rho) = 0.$$

Hence there exists a unique $\rho(b) \in (b\lambda_1, \infty)$ such that $F_b(\rho(b)) = 1$, equivalently $D_b(\rho(b)) = 0$.

Now let $b_2 > b_1$ and fix $\rho > b_2\lambda_1$. Since $\rho - b_2\lambda_i < \rho - b_1\lambda_i$ for all $i$, we have $F_{b_2}(\rho) > F_{b_1}(\rho)$. Evaluating at $\rho = \rho(b_1)$ gives

$$1 = F_{b_1}(\rho(b_1)) < F_{b_2}(\rho(b_1)).$$

Because $F_{b_2}$ is strictly decreasing, this implies $\rho(b_2) > \rho(b_1)$.

Continuity follows because $(b, \rho) \mapsto F_b(\rho)$ is continuous on sets $\{\rho \geq b\lambda_1 + \varepsilon\}$ and each $F_b$ is strictly monotone, so the inverse graph $\rho(b)$ depends continuously on $b$. □

**Proposition 9.** *Fix $b > 0$ and let $\rho(b) > b\lambda_1$ be the unique real zero of*

$$D_b(s) = 1 - 8 \sum_{i=1}^{d} \frac{(w_i^\star)^2 \lambda_i^2}{s - b\lambda_i}.$$

*Set $\delta_b := \rho(b) - b\lambda_1 > 0$. Then, for all $t \geq 0$,*

$$\alpha_b(t) = \frac{c(b)}{\sqrt{d}} e^{\rho(b)t} \left(1 + o(1)\right), \tag{C.22}$$

*where the constant*

$$c(b) = \frac{\sqrt{d}\,\widehat{a}_\Theta(\rho(b))}{D_b'(\rho(b))} \quad \text{satisfies} \quad |c(b)| \lesssim \frac{(s_\star^{(2)})^{3/2}}{\delta_b}.$$

*In particular, if $\widehat{a}_\Theta(\rho(b)) > 0$, then $c(b) > 0$.*

*Proof.* The proof consists in solving the Volterra equation in the Laplace domain, and then applying Laplace inversion together with residue calculus to characterize the solution in the original domain.

**Step 1: Laplace inversion.** For $\operatorname{Re} s > b\lambda_1$ we have

$$\widehat{\alpha_b}(s) = \frac{\widehat{a}_\Theta(s)}{D_b(s)}, \qquad \widehat{a}_\Theta(s) = \sum_{i=1}^{d} \frac{\lambda_i\, w_i(0)\, w_i^\star}{s - b\lambda_i}.$$

By Bromwich inversion, for any $\sigma_R > \rho(b)$,

$$\alpha_b(t) = \frac{1}{2\pi i} \int_{\sigma_R - i\infty}^{\sigma_R + i\infty} e^{st} \frac{\widehat{a}_\Theta(s)}{D_b(s)}\, ds.$$

**Step 2: Contour shift and residue extraction.** Fix $\sigma_* \in (b\lambda_1, \rho(b))$ and $T > 0$. Consider the rectangle with vertices $\sigma_R \pm iT$ and $\sigma_* \pm iT$. The only singularity of $e^{st}\widehat{a}_\Theta(s)/D_b(s)$ in the strip $\{\sigma_* < \operatorname{Re} s < \sigma_R\}$ is the simple pole at $s = \rho(b)$ (zeros of $D_b$ are real and interlace the poles $b\lambda_i$). Introduce the notation

$$\delta_b := \rho(b) - b\lambda_1 > 0$$

for the gap between the top pole and the dominant zero.

On the horizontal edges $s = x \pm iT$ with $x \in [\sigma_*, \sigma_R]$, we obtain

$$|\widehat{a}_\Theta(s)| \leq \sum_i \frac{|\lambda_i w_i(0) w_i^\star|}{|x - b\lambda_i \pm iT|} \leq \frac{1}{T} \sum_i |\lambda_i w_i(0) w_i^\star| \leq \frac{1}{T} \|w(0)\|_2 \left(\sum_i (w_i^\star)^2 \lambda_i^2\right)^{1/2} = \frac{\sqrt{s_\star^{(2)}}}{T},$$

where we used Cauchy–Schwarz and $\|w(0)\|_2 = 1$. Moreover, since

$$D_b(x \pm iT) = 1 - 8 \sum_i \frac{(w_i^\star)^2 \lambda_i^2}{x - b\lambda_i \pm iT} \to 1 \qquad \text{uniformly in } x \in [\sigma_*, \sigma_R] \text{ as } T \to \infty,$$

we have $\inf_{x \in [\sigma_*, \sigma_R]} |D_b(x \pm iT)| \geq \frac{1}{2}$ for all $T$ large enough. Therefore the horizontal contributions are $O(T^{-1})$ and vanish as $T \to \infty$.

By the residue theorem, we obtain the exact identity (valid for all $t \geq 0$):

$$\alpha_b(t) = \frac{\widehat{a}_\Theta(\rho(b))}{D_b'(\rho(b))} e^{\rho(b)t} + \mathcal{R}_{\sigma_*}(t), \qquad \mathcal{R}_{\sigma_*}(t) := \frac{1}{2\pi i} \int_{\sigma_* - i\infty}^{\sigma_* + i\infty} e^{st} \frac{\widehat{a}_\Theta(s)}{D_b(s)}\, ds. \tag{C.23}$$

**Step 3: Coefficient bound.** Let $v_i := \lambda_i/(\rho(b) - b\lambda_i)$. Then

$$\widehat{a}_\Theta(\rho(b)) = \sum_i w_i(0) w_i^\star v_i.$$

Under a standard isotropic initialization (e.g. $w(0) \sim \mathrm{Unif}(\mathbb{S}^{d-1})$ or $w(0) \sim N(0, I_d/d)$), conditional on $w^\star$ this is centered with variance $d^{-1}\sum_i (w_i^\star)^2 v_i^2$, hence

$$|\widehat{a}_\Theta(\rho(b))| \;\lesssim\; \frac{1}{\sqrt{d}}\Big(\sum_i (w_i^\star)^2 v_i^2\Big)^{1/2} \;\leq\; \frac{1}{\sqrt{d}} \cdot \frac{1}{\delta_b}\Big(\sum_i (w_i^\star)^2 \lambda_i^2\Big)^{1/2} \;=\; \frac{\sqrt{s_\star^{(2)}}}{\delta_b \sqrt{d}}.$$

Next,

$$D_b'(\rho(b)) = 8 \sum_i \frac{(w_i^\star)^2 \lambda_i^2}{(\rho(b) - b\lambda_i)^2} \geq \frac{1}{8 \sum_i (w_i^\star)^2 \lambda_i^2} = \frac{1}{8 s_\star^{(2)}},$$

where we used $1 = 8 \sum_i (w_i^\star)^2 \lambda_i^2/(\rho(b) - b\lambda_i)$ and Cauchy–Schwarz. Hence

$$\left| \frac{\widehat{a}_\Theta(\rho(b))}{D_b'(\rho(b))} \right| \;\lesssim\; \frac{(s_\star^{(2)})^{3/2}}{\delta_b \sqrt{d}}. \tag{C.24}$$

**Step 4: Remainder bound on $\mathrm{Re}\, s = \sigma_*$.** Since $\sigma_* \in (b\lambda_1, \rho(b))$ and all zeros of $D_b$ are real, $D_b$ has no zeros on the line $\mathrm{Re}\, s = \sigma_*$, and $D_b(\sigma_* + iy) \to 1$ as $|y| \to \infty$. Thus

$$m_b(\sigma_*) := \inf_{y \in \mathbb{R}} |D_b(\sigma_* + iy)| \;>\; 0.$$

Moreover, for $s = \sigma_* + iy$,

$$|\widehat{a}_\Theta(\sigma_* + iy)| = \Big| \sum_i w_i(0) w_i^\star \frac{\lambda_i}{\sigma_* - b\lambda_i + iy} \Big| \;\lesssim\; \frac{1}{\sqrt{d}} \left( \sum_i (w_i^\star)^2 \frac{\lambda_i^2}{(\sigma_* - b\lambda_i)^2 + y^2} \right)^{1/2},$$

by the same isotropic-initialization scaling as in Step 3 (applied to the fixed direction depending on $y$). Using $\sigma_* - b\lambda_i \geq \sigma_* - b\lambda_1$ and the standard integral $\int_\mathbb{R} [(a^2 + y^2)^{-1/2}(A^2 + y^2)^{-1/2}]\, dy \leq \pi/\sqrt{aA}$, one obtains

$$|\mathcal{R}_{\sigma_*}(t)| \;\leq\; \frac{C(s_\star^{(2)}, b)}{\sqrt{d}\, m_b(\sigma_*)} e^{\sigma_* t}, \qquad C(s_\star^{(2)}, b) = O\Big( \frac{(s_\star^{(2)})^{3/2}}{\sigma_* - b\lambda_1} \Big).$$

With the midpoint choice $\sigma_* = \rho(b) - \delta_b/2$, this gives

$$\frac{|\mathcal{R}_{\sigma_*}(t)|}{|\widehat{a}_\Theta(\rho(b))/D_b'(\rho(b))|\, e^{\rho(b)t}} \;\lesssim\; \frac{1}{m_b(\sigma_*)} e^{-(\delta_b/2)\, t}. \tag{C.25}$$

**Step 5: Uniformity for $t = O(\log d)$.** From the relation,

$$1 = 8 \sum_{i=1}^d \frac{(w_i^\star)^2 \lambda_i^2}{\rho(b) - b\lambda_i} \;\geq\; \frac{8(w_1^\star)^2 \lambda_1^2}{\rho(b) - b\lambda_1} = \frac{8(w_1^\star)^2 \lambda_1^2}{\delta_b},$$

so $\delta_b \geq 8(w_1^\star)^2 \lambda_1^2$. If $(w_1^\star)^2 \lambda_1^2 \geq \kappa_\Theta > 0$, then $\delta_b \geq \delta_\Theta := 8\kappa_\Theta$, a constant independent of $d$. Taking $t = c \log d$ in (C.25) yields exponential decay in $d$. Combining with (C.24) gives the claim with $c(b) = \sqrt{d}\, \widehat{a}_\Theta(\rho(b))/D_b'(\rho(b))$ and $|c(b)| \lesssim (s_\star^{(2)})^{3/2}/\delta_b$. $\qquad\square$

### C.3.2 GENERAL CASE

The ideal case $\Theta(t) = t$ served as a benchmark where the growth rate and coefficient could be computed explicitly. In the general case, $\Theta(t)$ is only an approximation of $t$, and the kernel $K_\Theta$ becomes a perturbation of the ideal one. The key point is to show that the dominant pole of the Laplace transform is stable under this perturbation, so that the same exponential growth persists up to a small shift in the rate and coefficient.

**Proposition 10.** *Denote by $\rho(4) > 4\lambda_1$ the unique real zero of $F_4$ on $(4\lambda_1, \infty)$. Write $F(p) := 1 - 8\widehat{K_\Theta}(p)$ and let $\rho_{\text{true}}$ be the rightmost real zero of $F$. Then:*

> *i) $|\rho_{\text{true}} - \rho(4)| \leq C\,\delta$.*
>
> *ii) Let $c(\delta) = \sqrt{d}\,\widehat{a}_\Theta(\rho_{\text{true}})/F'(\rho_{\text{true}})$. We have $|c(\delta)| \lesssim (s_\star^{(2)})^{3/2}/(\rho_{\text{true}} - 4\lambda_1)$. and*

$$u(t) = \frac{c(\delta)}{\sqrt{d}}\, e^{\rho_{\text{true}} t}\, \big(1 + o(1)\big).$$

*Proof.* Let $\Delta K := K_\Theta - K$ be the difference between the ideal kernel and the true one.

**Step 1: Pointwise kernel perturbation.** Since $(1 - 3\delta)t \leq \Theta(t) \leq t$ and $e^x - 1 \leq x$ for $x \leq 0$,

$$\big|e^{4\lambda_i \Theta(t)} - e^{4\lambda_i t}\big| \leq 4\lambda_i\,(t - \Theta(t))\,e^{4\lambda_i t} \leq 12\delta\,\lambda_i\,t\,e^{4\lambda_i t}.$$

Multiplying by $(w_i^\star)^2 \lambda_i^2$ and summing,

$$|\Delta K(t)| \leq 12\lambda_1\,\delta\,t\,K(t) \qquad (t \geq 0). \tag{C.26}$$

**Step 2: Laplace control of the perturbation.** For $\Re p > 4\lambda_1$,

$$\widehat{\Delta K}(p) = \int_0^\infty e^{-pt}\,\Delta K(t)\,dt, \qquad \left|\widehat{\Delta K}(p)\right| \leq \int_0^\infty e^{-(\Re p)t}\,|\Delta K(t)|\,dt.$$

Using (C.26) and differentiating under the integral sign (dominated by $tK(t)e^{-(\Re p)t}$),

$$\left|\widehat{\Delta K}(p)\right| \leq 12\lambda_1\,\delta \int_0^\infty t\,e^{-pt}\,K(t)\,dt = -12\lambda_1\,\delta\,\widehat{K}'(p).$$

Hence, for $F(p) = 1 - 8\widehat{K_\Theta}(p) = 1 - 8(\widehat{K}(p) + \widehat{\Delta K}(p))$,

$$|F(p) - F_4(p)| = 8\left|\widehat{\Delta K}(p)\right| \leq 96\,\lambda_1\,\delta\,\big(-\widehat{K}'(p)\big), \qquad \Re p > 4\lambda_1. \tag{C.27}$$

**Step 3: Unicity of the dominant zero.** Let $p_0 = \rho(4)$; then $F_4'(p_0) = -8\widehat{K}'(p_0) =: c_0 > 0$ and $F_4$ is strictly increasing on $(4\lambda_1, \infty)$. By continuity, pick $r > 0$ such that on $I := [p_0 - r, p_0 + r]$

$$F_4'(p) \geq c_0/2, \qquad M_1 := \sup_{p \in I}(-\widehat{K}'(p)) < \infty. \tag{C.28}$$

From (C.27),

$$\sup_{p \in I} |F(p) - F_4(p)| \leq 96\,\lambda_1\,M_1\,\delta =: \varepsilon_\delta.$$

Choose $\delta_\Theta$ small such that $\varepsilon_\delta \leq (c_0/4)r$ for all $\delta \in (0, \delta_\Theta]$. Then

$$F(p_0+r) \geq F_4(p_0+r) - \varepsilon_\delta \geq (c_0/4)r > 0, \qquad F(p_0-r) \leq F_4(p_0-r) + \varepsilon_\delta \leq -(c_0/4)r < 0.$$

Because $F'(p) = -8\widehat{K_\Theta}'(p) = 8\int_0^\infty te^{-pt}K_\Theta(t)\,dt > 0$ on $(4\lambda_1, \infty)$, there is a unique zero $\rho_{\text{true}} \in (p_0 - r, p_0 + r)$ and it is the rightmost one. Moreover,

$$|\rho_{\text{true}} - \rho(4)| \leq \frac{\varepsilon_\delta}{c_0/2} \lesssim \delta,$$

proving i).

**Step 4: Contour decomposition and leading coefficient.** Bromwich inversion and a contour shift to $\Re p = \sigma_* \in (4\lambda_1, \rho_{\text{true}})$ give the exact identity

$$u(t) = \frac{\widehat{a}_\Theta(\rho_{\text{true}})}{F'(\rho_{\text{true}})}\, e^{\rho_{\text{true}} t} + \frac{1}{2\pi i} \int_{\sigma_* - i\infty}^{\sigma_* + i\infty} e^{pt}\,\frac{\widehat{a}_\Theta(p)}{F(p)}\,dp, \tag{C.29}$$

since the only singularity crossed is the simple pole at $p = \rho_{\text{true}}$. For the coefficient, set $v_i = \lambda_i/(\rho_{\text{true}} - 4\lambda_i)$. Then

$$\left| \widehat{a}_\Theta(\rho_{\text{true}}) \right| = \left| \sum_i w_i(0) w_i^\star v_i \right| \lesssim \frac{1}{\sqrt{d}} \left( \sum_i (w_i^\star)^2 v_i^2 \right)^{1/2} \leq \frac{1}{\sqrt{d}} \cdot \frac{\sqrt{s_\star^{(2)}}}{\rho_{\text{true}} - 4\lambda_1},$$

where the $\lesssim d^{-1/2}$ scaling follows under isotropic random initialization (e.g. $w(0) \sim \mathcal{N}(0, I_d/d)$ or $w(0) \sim \text{Unif}(\mathbb{S}^{d-1})$), and is encoded in Assumption A1. Further, $F'(\rho_{\text{true}}) = 8 \int_0^\infty t e^{-\rho_{\text{true}} t} K_\Theta(t)\, dt > 0$, and continuity from the ideal case implies $F'(\rho_{\text{true}}) \geq \frac{1}{16\, s_\star^{(2)}}$ for all small $\delta$. Therefore, under Assumption A1

$$\left| \frac{\widehat{a}_\Theta(\rho_{\text{true}})}{F'(\rho_{\text{true}})} \right| \lesssim \frac{(s_\star^{(2)})^{3/2}}{\sqrt{d}\,(\rho_{\text{true}} - 4\lambda_1)}. \tag{C.30}$$

**Step 5: Vertical-line remainder.** On $\Re p = \sigma_*$, monotonicity/convexity and $F'(\rho_{\text{true}}) > 0$ yield a uniform gap $\inf_y |F(\sigma_* + iy)| \gtrsim \rho_{\text{true}} - \sigma_*$. Also,

$$|\widehat{a}_\Theta(\sigma_* + iy)| \lesssim \frac{1}{\sqrt{d}} \left( \sum_i (w_i^\star)^2 \frac{\lambda_i^2}{(\sigma_* - 4\lambda_i)^2 + y^2} \right)^{1/2},$$

by the same isotropic-initialization scaling as above (applied at the fixed value $y$). Estimating the integral in (C.29) by Cauchy–Schwarz in $y$ and using $\int_{\mathbb{R}} \frac{dy}{(a^2 + y^2)} = \pi/a$ and

$$\int_{\mathbb{R}} \frac{dy}{\sqrt{a^2 + y^2}\sqrt{A^2 + y^2}} \leq \pi/\sqrt{aA},$$

we obtain

$$\underbrace{\left| \frac{1}{2\pi i} \int_{\sigma_* - i\infty}^{\sigma_* + i\infty} e^{pt} \frac{\widehat{a}_\Theta(p)}{F(p)}\, dp \right|}_{R} \lesssim \frac{1}{\sqrt{d}} \cdot \frac{e^{\sigma_* t}}{\rho_{\text{true}} - \sigma_*}.$$

Choose the midpoint $\sigma_* = \rho_{\text{true}} - \Delta/2$ with $\Delta := \rho_{\text{true}} - 4\lambda_1$. Then

$$R \lesssim \frac{1}{\sqrt{d}} e^{(\rho_{\text{true}} - \Delta/2)t}.$$

**Step 6: Conclusion.** From the ideal dispersion relation,

$$1 = 8 \sum_i \frac{(w_i^\star)^2 \lambda_i^2}{\rho(4) - 4\lambda_i} \geq \frac{8(w_1^\star)^2 \lambda_1^2}{\rho(4) - 4\lambda_1} \quad \Rightarrow \quad \rho(4) - 4\lambda_1 \geq 8\kappa_\Theta.$$

By i), for small $\delta$ the true gap obeys $\Delta = \rho_{\text{true}} - 4\lambda_1 \geq 4\kappa_\Theta =: \Delta_\Theta > 0$. Hence for $t = c \log d$,

$$\frac{R}{\left| \widehat{a}_\Theta(\rho_{\text{true}})/F'(\rho_{\text{true}}) \right| e^{\rho_{\text{true}} t}} \lesssim e^{-(\Delta_\Theta/2)t} = d^{-c\Delta_\Theta/2} \to 0.$$

Putting this with (C.30) yields the stated asymptotic with $c(\delta) = \sqrt{d}\, \widehat{a}_\Theta(\rho_{\text{true}})/F'(\rho_{\text{true}})$, continuous in $\delta$ and bounded away from 0 as $\delta \to 0$ when $\widehat{a}_\Theta(\rho(4)) > 0$. $\qquad\square$

This proposition motivates our assumption on initialization.

**Assumption A1** (Initialization). *We assume that $u(0) \asymp d^{-1/2}$, $s(0) \asymp d^{-1}$ and $\hat{a}_\Theta(\rho_{\text{true}}) \asymp d^{-1/2}$. Moreover, we assume that $w_1^\star$ is of constant order and non-zero.*

**Remark 6** (On the initialization assumption). *The assumption in A1 is natural under random initialization. Indeed, if $w(0) \sim \mathcal{N}(0, I_d/d)$ (or uniform on the sphere) independently of $w_\star$, then conditionally on $w_\star$ we have*

$$\widehat{a}_\Theta(\rho_{\text{true}}) = \sum_i w_i(0) w_i^\star v_i \sim \mathcal{N}\left(0, \tfrac{1}{d} \sum_i (w_i^\star)^2 v_i^2\right), \qquad v_i = \frac{\lambda_i}{\rho_{\text{true}} - 4\lambda_i}.$$

*This variance is of order $1/d$. Hence $\widehat{a}_\Theta(\rho_{\text{true}})$ is of order $d^{-1/2}$ with large probability, so that the prefactor in Proposition 10(ii) is indeed nontrivial.*

*The quantity $u(0)$ also fluctuates on the order $d^{-1/2}$ for Gaussian initialization, but establishing a deterministic relation between $u(0)$ and $\widehat{a}_\Theta(\rho_{\text{true}})$ is delicate, as the two depend differently on the spectrum and on $w_\star$. This explains why A1 is stated as a mild probabilistic assumption rather than a deterministic condition.*

## C.4  STEP 3: BOUNDING HIGHER-ORDER STATISTICS AND POSITIVITY OF $u^{(2)}$

Our goal in this subsection is to show that the second moment $u^{(2)}(T_1)$ is *strictly positive*.

For $k \geq 1$, define

$$(K_\Theta)_k(t) := \left(e^{4B\,\Theta(t)}s_\star\right)_k = \sum_{m \geq 0} \frac{(4\Theta(t))^m}{m!} s_\star^{(k+m)}, \qquad s_\star^{(\ell)} = \sum_i (w_i^\star)^2 \lambda_i^\ell, \quad \ell \geq 1.$$

**Lemma 9** (Kernel domination). *For all $k \geq 1$ and $t \in [0, T_1]$,*

$$(K_\Theta)_k(t) \leq \lambda_1^{k-1} K_\Theta(t), \qquad K_\Theta(t) := \sum_{m \geq 0} \frac{(4\Theta(t))^m}{m!} s_\star^{(1+m)}.$$

*Proof.* Since $\Theta(t) \geq 0$ for $t \leq T_1$, all coefficients $\frac{(4\Theta(t))^m}{m!}$ are nonnegative. For each $m \geq 0$,

$$s_\star^{(k+m)} = \sum_i (w_i^\star)^2 \lambda_i^{k+m} \leq \lambda_1^{k-1} \sum_i (w_i^\star)^2 \lambda_i^{1+m} = \lambda_1^{k-1} s_\star^{(1+m)}.$$

Multiplying termwise by the nonnegative coefficients and summing over $m$ yields the claim. $\qquad\square$

**Lemma 10** (Kernel positivity). *For all $k \geq 1$ and $t \in [0, T_1]$,*

$$(K_\Theta)_k(t) \geq s_\star^{(k)} \geq (w_1^\star)^2 \lambda_1^k.$$

*Proof.* When $\Theta(t) \geq 0$, all coefficients $\frac{(4\Theta(t))^m}{m!}$ in the definition of $(K_\Theta)_k(t)$ are nonnegative. In particular, the $m = 0$ term contributes $s_\star^{(k)}$, so $(K_\Theta)_k(t) \geq s_\star^{(k)}$. Finally, $s_\star^{(k)} = \sum_i (w_i^\star)^2 \lambda_i^k \geq (w_1^\star)^2 \lambda_1^k$. $\qquad\square$

**Proposition 11** (Control of higher-order correlations). *For all $k \geq 1$ and all $t \in [0, T_1]$, we have*

$$-\lambda_1^k \sqrt{\tfrac{\log d}{d}}\, e^{4\lambda_1 t} + 8\, s_\star^{(k)} \int_0^t u(\tau)\, d\tau \leq u^{(k)}(t) \leq \lambda_1^k \sqrt{\tfrac{\log d}{d}}\, e^{4\lambda_1 t} + \lambda_1^{k-1}\big(u(t) - u(0)\big). \tag{C.31}$$

*In particular, for $d$ sufficiently large there exists a constant $c_1 > 0$ such that $u^{(2)}(T_1) \geq c_1$.*

*Proof.* Work on $[0, T_1]$ where $\Theta \geq 0$ (so Lemma 10 applies), and recall Lemma 9. We have w.h.p.

$$|u^{(k+m)}(0)| \leq \lambda_1^{k+m} \sqrt{\frac{\log d}{d}}, \qquad \text{for all } k \geq 1,\ m \geq 0. \tag{C.32}$$

By Duhamel's formula, for each $k \geq 1$ and $t \in [0, T_1]$,

$$u^{(k)}(t) = \sum_{m \geq 0} \frac{(4\Theta(t))^m}{m!} u^{(k+m)}(0) + 8 \int_0^t (K_\Theta)_k(t - \tau)\, u(\tau)\, d\tau.$$

From (C.32) and $\Theta(t) \leq t$, we obtain

$$\left| \sum_{m \geq 0} \frac{(4\Theta(t))^m}{m!} u^{(k+m)}(0) \right| \leq \sqrt{\frac{\log d}{d}} \sum_{m \geq 0} \frac{(4\Theta(t)\lambda_1)^m}{m!} \lambda_1^k$$

$$= \lambda_1^k \sqrt{\frac{\log d}{d}}\, e^{4\lambda_1 \Theta(t)}$$

$$\leq \lambda_1^k \sqrt{\frac{\log d}{d}}\, e^{4\lambda_1 t}.$$

Furthermore, by Lemma 9, we have $(K_\Theta)_k \le \lambda_1^{k-1}(K_\Theta)$, so

$$\int_0^t (K_\Theta)_k(t-\tau)\,u(\tau)\,d\tau \;\le\; \lambda_1^{k-1}\big(u(t) - a_\Theta(t)\big).$$

Combining with the homogeneous upper bound yields the right-hand inequality in (C.31).

Lemma 10 gives $(K_\Theta)_k(t-\tau) \ge s_\star^{(k)}$ for $\tau \in [0,t]$, hence

$$\int_0^t (K_\Theta)_k(t-\tau)\,u(\tau)\,d\tau \;\ge\; s_\star^{(k)} \int_0^t u(\tau)\,d\tau.$$

Combining with the homogeneous lower bound gives the left-hand inequality in (C.31).

Finally, under the positive growth of $\alpha$ assumed in Step 3 (e.g. $\alpha(t) \ge \frac{C}{\sqrt{d}} e^{\rho t}$ for large $t$), together with a rate gap $\rho > 4\lambda_1$, the integral term $8 s_\star^{(2)} \int_0^{T_1} \alpha$ dominates the homogeneous remainder $\lambda_1^2 \sqrt{\frac{\log d}{d}}\, e^{4\lambda_1 T_1} = O(\sqrt{\log d})$ for $d$ large enough. Hence, there exists $c_1 > 0$ such that $u^{(2)}(T_1) \ge c_1$. $\qquad\square$

## D   ANALYSIS OF PHASE II

In Section D.1, we show that $s(t)$ crosses the threshold $1/3$ within $O(1)$ time and remains above it thereafter. Once $s(t) > 1/3$, the key quantity $\Theta(t)$ appearing in the Volterra equation begins to decrease and eventually becomes negative. This marks a qualitative shift in the dynamics of the system. We leverage this change in Section D.2 to derive a convergence rate for the summary statistics.

### D.1   PHASE IIA: CROSSING THE $s(t) = 1/3$ THRESHOLD AND IRREVERSIBLE GROWTH

Recall that at the end of Phase I, there exists an absolute constant $\delta > 0$ such that

$$u(T_1) \;>\; \delta, \qquad s(T_1) \;\ge\; \delta, \qquad u^{(2)}(T_1) \;>\; \delta.$$

At a high level, the behavior of $s(t)$ in this regime is governed by a simple mechanism. While $s(t) \le 1/3$, both terms in (D.1) are nonnegative, so $s(t)$ is pushed upward and necessarily crosses the threshold $1/3$ in finite time. Once $s(t)$ has crossed, the positive mixed term $16\,u(t)u^{(2)}(t)$ outweighs the negative contribution of the first term near the boundary, which prevents $s(t)$ from falling back. Thus $s(t)$ remains bounded away from $1/3$ uniformly after crossing. The next proposition makes this precise.

**Proposition 12** (Crossing and stability beyond the $1/3$-threshold). *Let $T_1$ be the stopping time from Theorem 1. There exist constants $\delta > 0$ and $C > 0$ such that:*

1. *There exists $T_1' \in [T_1,\, T_1 + C]$ with*

$$s(T_1') \;\ge\; \tfrac{1}{3} + \delta.$$

2. *For all $t \ge T_1'$,*

$$s(t) \;\ge\; \tfrac{1}{3} + \delta.$$

*Proof.* From $T_1$ onward we first prove that $u^{(2)}(t)$ and then $u(t)$ stay strictly positive. While $s(t) \le 1/3$, this makes both terms in $\dot{s}$ nonnegative, so $s$ reaches $1/3$ in finite time (Part A). Next, we work in a thin band $[1/3,\, 1/3 + \eta]$, show that for $\eta$ small enough the drift of $s$ is still uniformly positive, so $s$ reaches $1/3 + \eta$ in finite time, and that the vector field points inward at $s = 1/3 + \eta$, preventing any return (Part B).

Recall

$$\dot{s}(t) \;=\; 8\big(1 - 3s(t)\big)\,s^{(2)}(t) \;+\; 16\,u(t)\,u^{(2)}(t), \tag{D.1}$$

$$\dot{u}(t) \;=\; 4\big(1 - 3s(t)\big)\,u^{(2)}(t) \;+\; 8\,s_\star^{(2)}\,u(t), \qquad \dot{u}^{(2)}(t) \;=\; 4\big(1 - 3s(t)\big)\,u^{(3)}(t) \;+\; 8\,s_\star^{(3)}\,u(t). \tag{D.2}$$

**Part A: Pre-band positivity and finite-time reach of $s = 1/3$.** Fix $\tau := t - T_1 \geq 0$ and define $\Theta_{T_1}(\tau) := \Theta(T_1 + \tau) - \Theta(T_1)$. Writing $\alpha_i(t) := w_i(t)w_i^\star$, we have

$$\dot{\alpha}_i(t) = 4(1 - 3s(t))\lambda_i \alpha_i(t) + 8\lambda_i(w_i^\star)^2 u(t).$$

Multiplying by $e^{-4\lambda_i \Theta(t)}$ and integrating from $T_1$ to $T_1 + \tau$ yields

$$\alpha_i(T_1 + \tau) = e^{4\lambda_i \Theta_{T_1}(\tau)}\alpha_i(T_1) + 8\int_0^\tau e^{4\lambda_i(\Theta_{T_1}(\tau) - \Theta_{T_1}(s))}\lambda_i(w_i^\star)^2 u(T_1 + s)\,ds.$$

Multiplying by $\lambda_i^2$ and summing gives

$$u^{(2)}(T_1 + \tau) = \underbrace{\sum_i \lambda_i^2 e^{4\lambda_i \Theta_{T_1}(\tau)}\alpha_i(T_1)}_{=:\ \tilde{a}_0^{(2)}(\tau)} + 8\int_0^\tau K_\times(\tau - s)\,u(T_1 + s)\,ds, \tag{D.3}$$

with

$$K_\times(\zeta) := \sum_i (w_i^\star)^2 \lambda_i^3\, e^{4\lambda_i \Theta_{T_1}(\zeta)} \geq 0.$$

On the pre-band interval where $s \leq 1/3$, we have $\Theta_{T_1}(\tau) \geq 0$, so $e^{4\lambda_i \Theta_{T_1}(\tau)} \geq 1$. Thus

$$\tilde{a}_0^{(2)}(\tau) \geq \sum_i \lambda_i^2 \alpha_i(T_1) = u^{(2)}(T_1) > 0.$$

By (D.3) and $K_\times \geq 0$,

$$u^{(2)}(t) \geq u^{(2)}(T_1) > 0 \qquad \text{for all } T_1 \leq t \leq T_{1/3} := \inf\{t \geq T_1 : s(t) = 1/3\}. \tag{D.4}$$

While $s \leq 1/3$, from (D.2) and (D.4),

$$\dot{u}(t) \geq 8s_\star^{(2)}u(t), \qquad T_1 \leq t \leq T_{1/3},$$

so $u(t) \geq u(T_1)e^{8s_\star^{(2)}(t - T_1)} > 0$ there. Finally, (D.1) gives on $\{s \leq 1/3\}$

$$\dot{s}(t) \geq 16\,u(t)\,u^{(2)}(t) \geq 16\,u(T_1)\,u^{(2)}(T_1) =: \kappa_\Theta > 0.$$

Hence

$$T_{1/3} - T_1 \leq \frac{(\frac{1}{3}) - s(T_1)}{\kappa_\Theta} \leq \frac{1/3}{16\,u(T_1)u^{(2)}(T_1)}. \tag{D.5}$$

Thus $s$ reaches $1/3$ in finite time.

**Part B: Crossing the band $[1/3,\ 1/3 + \eta]$ and no return.** We record uniform bounds that will be used in-band. Since $Q \preceq \lambda_1 I$ and $\|w(t)\| \leq M$,

$$0 \leq s^{(2)}(t) \leq \lambda_1^2 \|w(t)\|^2 \leq S_2^{\max}, \qquad S_2^{\max} := \lambda_1^2 M^2, \tag{D.6}$$

and $0 \leq s(t) \leq S^{\max}$ with $S^{\max} := \lambda_1 M^2$. By Cauchy–Schwarz,

$$|u^{(2)}(t)| \leq \sqrt{s(t)}\sqrt{s_\star^{(3)}} \leq \sqrt{S^{\max} s_\star^{(3)}} =: C_2, \quad |u^{(3)}(t)| \leq \sqrt{s^{(2)}(t)}\sqrt{s_\star^{(4)}} \leq \sqrt{S_2^{\max} s_\star^{(4)}} =: C_3. \tag{D.7}$$

Fix $\eta \in (0, 1/6]$. On $\{1/3 \leq s \leq 1/3 + \eta\}$ we have $1 - 3s \in [-3\eta, 0]$. From (D.2) and (D.7),

$$\dot{u} \geq -12\eta\,C_2 + 8s_\star^{(2)}u, \qquad \dot{u}^{(2)} \geq -12\eta\,C_3 + 8s_\star^{(3)}u.$$

As in a linear comparison argument, if

$$\eta \leq \tfrac{2}{3}\tfrac{s_\star^{(2)}}{C_2}\,u(T_1), \qquad \eta \leq \tfrac{2}{3}\tfrac{s_\star^{(3)}}{C_3}\,u(T_1), \tag{D.8}$$

then throughout the band one has $u(t) \geq u(T_1)$ and $u^{(2)}(t) \geq u^{(2)}(T_1)$.

Now, from (D.1), (D.6), and these lower bounds,

$$\dot{s}(t) \ \geq \ -24\eta\, S_2^{\max} \ + \ 16\, u(T_1)\, u^{(2)}(T_1).$$

Choosing

$$\eta \ \leq \ \min\left\{ \ \frac{u(T_1)u^{(2)}(T_1)}{3S_2^{\max}}, \ \frac{2}{3}\frac{s_\star^{(2)}}{C_2}u(T_1), \ \frac{2}{3}\frac{s_\star^{(3)}}{C_3}u(T_1), \ \frac{1}{6} \ \right\}, \tag{D.9}$$

we get $\dot{s}(t) \geq 8\, u(T_1)u^{(2)}(T_1) > 0$ across the band. Thus $s$ overshoots to $1/3 + \eta$ in time at most $\eta/(8u(T_1)u^{(2)}(T_1))$, and at any last contact with $s = 1/3 + \eta$ the same inequality shows $\dot{s} > 0$, so the vector field points inward. Therefore, $s$ cannot return below $1/3 + \eta$.

Together with (D.5), this completes the proof: $s$ reaches $1/3$ in finite time, then crosses to $1/3 + \eta$ in finite time, and never falls back below. $\qquad\square$

## D.2 PHASE IIB: APPROXIMATE CONVERGENCE OF THE SUMMARY STATISTICS

After the entrance time $T_1'$, the error $\Delta(t) := 1 - u(t)$ satisfies

$$\Delta(t) \ = \ b_\Theta(t) \ + \ \int_{T_1'}^t K_\Theta(t,\tau)\,\Delta(\tau)\,d\tau, \qquad t \geq T_1', \tag{D.10}$$

with

$$K_\Theta(t,\tau) = \sum_{i=1}^d 8\,\lambda_i^2\,(w_i^\star)^2\, e^{\,4\lambda_i\left(\Theta(t)-\Theta(\tau)\right)}, \quad h_\Theta(t) := \sum_{i=1}^d \lambda_i w_i^\star w_i(T_1')\, e^{\,4\lambda_i\left(\Theta(t)-\Theta(T_1')\right)},$$

$$b_\Theta(t) \ := \ 1 - h_\Theta(t) \ - \ \int_{T_1'}^t K_\Theta(t,\tau)\,d\tau.$$

We will repeatedly use the following facts:

(i) $1 \geq \Delta(\tau) \geq 0$ and $K_\Theta(t,\tau) \geq 0$ for $t \geq \tau \geq T_1'$.

(ii) $\Theta'(t) = 1 - 3s(t) \leq -s_0 < 0$, hence $e^{\,4\lambda(\Theta(t)-\Theta(\tau))} \leq e^{-4s_0\lambda\,(t-\tau)}$ for $t \geq \tau \geq T_1'$.

The first fact results from Lemma 11.

For a cutoff $\lambda_c > 0$ let

$$\mathcal{I}_< := \{i : \lambda_i < \lambda_c\}, \qquad \mathcal{I}_\geq := \{i : \lambda_i \geq \lambda_c\}.$$

Define

$$T(\lambda_c) := \sum_{\lambda_i < \lambda_c} \lambda_i\,(w_i^\star)^2, \qquad s_\star^{(2)} := \sum_{i=1}^d \lambda_i^2\,(w_i^\star)^2, \qquad s(T_1') = \sum_{i=1}^d \lambda_i\,w_i(T_1')^2 \leq 1,$$

and the head alignment term

$$S_\geq(\lambda_c) := \sum_{\lambda_i \geq \lambda_c} \lambda_i\,\big|w_i^\star\,w_i(T_1')\big| \ \leq \ \Big( \sum_{\lambda_i \geq \lambda_c} \lambda_i\,(w_i^\star)^2 \Big)^{1/2} \Big( \sum_{\lambda_i \geq \lambda_c} \lambda_i\,w_i(T_1')^2 \Big)^{1/2} \ \leq \ \sqrt{s_\star^{(2)}\,s(T_1)}.$$

**Proposition 13.** *For all $t \geq T_1'$ and any cutoff $\lambda_c > 0$,*

$$\Delta(t) \ \leq \ T(\lambda_c) \ + \ \sqrt{T(\lambda_c)\,s(T_1')} \ + \ S_\geq(\lambda_c)\,e^{-\,4s_0\,\lambda_c\,(t-T_1')}. \tag{D.11}$$

*Proof.* From the Volterra equation (D.10) and the definition of $b_\Theta$,

$$\Delta(t) \ = \ (1 - h_\Theta(t)) + \int_{T_1'}^t K_\Theta(t,\tau)\,\big(\Delta(\tau) - 1\big)\,d\tau.$$

Since $0 \leq \Delta(\tau)$ and $K_\Theta \geq 0$, the integral is nonpositive. Hence

$$\Delta(t) \ \leq \ (1 - h_\Theta(t))_+.$$

Now split the spectrum into $\mathcal{I}_< = \{i : \lambda_i < \lambda_c\}$ and $\mathcal{I}_\geq = \{i : \lambda_i \geq \lambda_c\}$. Using $\Theta(t) \leq \Theta(T'_1)$ and $\Theta'(t) \leq -s_0$, we obtain

$$(1 - h_\Theta(t))_+ \leq \sum_{i \in \mathcal{I}_<} \lambda_i (w_i^\star)^2 + \sum_{i \in \mathcal{I}_<} \lambda_i |w_i^\star w_i(T'_1)| + \sum_{i \in \mathcal{I}_\geq} \lambda_i |w_i^\star w_i(T'_1)| \, e^{-4s_0 \lambda_c \, (t - T'_1)}.$$

The first sum equals $T(\lambda_c)$. By Cauchy–Schwarz,

$$\sum_{i \in \mathcal{I}_<} \lambda_i |w_i^\star w_i(T'_1)| \leq \sqrt{T(\lambda_c)\, s(T'_1)}.$$

The last sum is bounded by $S_\geq(\lambda_c) e^{-4s_0 \lambda_c \, (t - T'_1)}$. Combining these estimates yields (D.11). $\qquad\square$

**Corollary 1** (Time to reach accuracy $\varepsilon$). *Fix $\varepsilon \in (0,1)$ and choose $\lambda_\varepsilon > 0$ such that*

$$T(\lambda_\varepsilon) \leq \frac{\varepsilon}{4}, \qquad \sqrt{T(\lambda_\varepsilon)\, s(T'_1)} \leq \frac{\varepsilon}{4} \iff T(\lambda_\varepsilon) \leq \min\left\{\frac{\varepsilon}{4}, \frac{\varepsilon^2}{16\, s(T'_1)}\right\}. \tag{D.12}$$

*Then*

$$T_2(\varepsilon) := T'_1 + \frac{1}{4s_0\, \lambda_\varepsilon} \log\left(\frac{4\, S_\geq(\lambda_\varepsilon)}{\varepsilon}\right) \tag{D.13}$$

*satisfies $\Delta(t) \leq \varepsilon$ for all $t \geq T_2(\varepsilon)$.*

*Proof.* From (D.11) with $\lambda_c = \lambda_\varepsilon$ and (D.12),

$$\Delta(t) \leq \frac{\varepsilon}{4} + \frac{\varepsilon}{4} + S_\geq(\lambda_\varepsilon)\, e^{-4s_0 \lambda_\varepsilon \, (t - T'_1)} = \frac{\varepsilon}{2} + S_\geq(\lambda_\varepsilon)\, e^{-4s_0 \lambda_\varepsilon \, (t - T'_1)}.$$

Choosing $t$ so that the last term is $\leq \varepsilon/2$ gives (D.13). $\qquad\square$

**Corollary 2** (Power-law spectral tail). *Suppose $T(\lambda) \asymp C_{\text{tail}} \lambda^\beta$ as $\lambda \downarrow 0$ with $\beta = 1 - \frac{1}{a} \in (0,1)$. Then*

$$\lambda_\varepsilon \asymp \left(\frac{\min\{\varepsilon, \, \varepsilon^2/s(T'_1)\}}{C_{\text{tail}}}\right)^{1/\beta}, \qquad T_2(\varepsilon) \lesssim T'_1 + \frac{1}{4s_0}\left(\frac{C_{\text{tail}}}{\min\{\varepsilon, \, \varepsilon^2/s(T'_1)\}}\right)^{1/\beta} \log \frac{1}{\varepsilon}.$$

*In particular, when $\varepsilon$ is small and $s(T'_1) \leq 1$, the quadratic condition dominates:*

$$T_2(\varepsilon) \lesssim T'_1 + \frac{1}{4s_0} \varepsilon^{-2/\beta} \log \frac{1}{\varepsilon} = T'_1 + \frac{1}{4s_0} \varepsilon^{-\frac{2a}{a-1}} \log \frac{1}{\varepsilon}.$$

### D.2.1 TECHNICAL LEMMA

It will be convenient to approximate the discrete measure $\mu_d$ by a continuous reference measure with density proportional to $\lambda^{1-1/a}$ near $\lambda = 0$. This approximation is crucial in Phase II, since the long-time behavior of the kernel $K_\Theta$ depends only on the small-$\lambda$ (tail) mass of $\mu_d$, which in turn reflects the power-law spectrum assumption. The following proposition shows that, with high probability, $\mu_d$ has a similar tail behaviour to its continuous counterpart.

**Proposition 14** (Spectral/teacher tail near $\lambda = 0$ holds w.h.p.). *Let $a > 1$ and define the spectrum by $\lambda_i := i^{-a}/H_{d,a}$, where $H_{d,a} = \sum_{j=1}^d j^{-a}$. Let $w_i^\star \overset{\text{i.i.d.}}{\sim} \mathcal{N}(0,1)$, independent of $(\lambda_i)$, and set*

$$\mu_d := 8 \sum_{i=1}^d \lambda_i (w_i^\star)^2 \, \delta_{\lambda_i}.$$

*Fix constants $C_\star > 1$ and $\rho \in (0,1)$, and define the tail scale $\Lambda_d := C_\star \lambda_d = C_\star d^{-a}/H_{d,a}$ and the geometric bins*

$$B_k := \left(\rho^{k+1} \Lambda_d, \, \rho^k \Lambda_d\right], \qquad k = 0, 1, \ldots, K,$$

*where $K := \lceil \log_{1/\rho}(C_\star) \rceil - 1$. Then there exist constants $C_-, C_+, c > 0$ depending only on $(a, \rho, C_\star)$ such that, for all sufficiently large $d$, with probability at least $1 - e^{-cd}$,*

$$C_- \int_{B_k} \lambda^{-\frac{1}{a}} \, d\lambda \leq \mu_d(B_k) \leq C_+ \int_{B_k} \lambda^{-\frac{1}{a}} \, d\lambda \qquad \text{for all } k = 0, 1, \ldots, K. \tag{D.14}$$

*Consequently, with the same probability bound, for every $\lambda \in (0, \Lambda_d]$,*

$$\mu_d\big((0, \lambda]\big) \;\asymp\; \lambda^{1 - \frac{1}{a}}, \tag{D.15}$$

*and, more generally, for any nonnegative step test function $\varphi(\lambda) = \sum_{k=0}^{K} a_k \, \mathbf{1}_{B_k}(\lambda)$,*

$$C_- \int_0^{\Lambda_d} \lambda^{-\frac{1}{a}} \, \varphi(\lambda) \, d\lambda \;\leq\; \int_{(0, \Lambda_d]} \varphi(\lambda) \, \mu_d(d\lambda) \;\leq\; C_+ \int_0^{\Lambda_d} \lambda^{-\frac{1}{a}} \, \varphi(\lambda) \, d\lambda. \tag{D.16}$$

*Proof.* We split the proof into several steps.

**Step 1: Bin sizes are linear in $d$.** Let $I_k := \{ 1 \leq i \leq d : \lambda_i \in B_k \}$. Since $\lambda_i = i^{-a}/H_{d,a}$, the condition $\lambda_i \leq t$ is equivalent to $i \geq (t H_{d,a})^{-1/a}$. Thus the counting function

$$N(t) := \#\{i : \lambda_i \leq t\} = d - \left\lceil (t H_{d,a})^{-1/a} \right\rceil + 1.$$

Therefore,

$$|I_k| \;=\; N(\rho^k \Lambda_d) - N(\rho^{k+1} \Lambda_d) = \left\lceil (\rho^{k+1} \Lambda_d H_{d,a})^{-1/a} \right\rceil - \left\lceil (\rho^k \Lambda_d H_{d,a})^{-1/a} \right\rceil.$$

Using $\Lambda_d H_{d,a} = C_\star d^{-a}$ gives

$$|I_k| = \left\lceil \rho^{-(k+1)/a} C_\star^{-1/a} \, d \right\rceil - \left\lceil \rho^{-k/a} C_\star^{-1/a} \, d \right\rceil.$$

Hence, for all $d$ large enough,

$$c_1 \, d \, \rho^{-k/a} \;\leq\; |I_k| \;\leq\; c_2 \, d \, \rho^{-k/a} \qquad (k = 0, 1, \ldots, K), \tag{D.17}$$

for constants $c_1, c_2 > 0$ depending only on $(a, \rho, C_\star)$. Thus $|I_k| = \Theta(d \, \rho^{-k/a})$, and $K$ is a fixed constant (independent of $d$). Note: for the last bin $B_K$, the lower endpoint may fall below $\lambda_d$, but by definition $I_K$ only counts actual indices $i \leq d$.

**Step 2: Deterministic comparison of $\sum_{i \in I_k} \lambda_i$.** For $i \in I_k$ we have $\rho^{k+1} \Lambda_d < \lambda_i \leq \rho^k \Lambda_d$, so

$$(\rho^{k+1} \Lambda_d) \, |I_k| \;\leq\; \sum_{i \in I_k} \lambda_i \;\leq\; (\rho^k \Lambda_d) \, |I_k|.$$

On the other hand,

$$\int_{B_k} \lambda^{-\frac{1}{a}} \, d\lambda = \frac{(\rho^k \Lambda_d)^{1 - \frac{1}{a}} - (\rho^{k+1} \Lambda_d)^{1 - \frac{1}{a}}}{1 - \frac{1}{a}} = \left( \frac{1 - \rho^{1 - \frac{1}{a}}}{1 - \frac{1}{a}} \right) \Lambda_d^{1 - \frac{1}{a}} \, \rho^{k(1 - \frac{1}{a})}. \tag{D.18}$$

Since $a > 1$, the exponent $1 - 1/a > 0$. Using (D.17) and (D.18), there exist constants $D_-, D_+ > 0$ (depending only on $a, \rho, C_\star$ and the bounded factor $H_{d,a}^{-1/a} \in [\zeta(a)^{-1/a}, 1]$) such that

$$D_- \int_{B_k} \lambda^{-\frac{1}{a}} \, d\lambda \;\leq\; \sum_{i \in I_k} \lambda_i \;\leq\; D_+ \int_{B_k} \lambda^{-\frac{1}{a}} \, d\lambda \qquad (k = 0, 1, \ldots, K). \tag{D.19}$$

**Step 3: Concentration for the teacher weights.** Let $S_k := \sum_{i \in I_k} (w_i^\star)^2$. Since $S_k \sim \chi_{|I_k|}^2$ and $|I_k| \geq c_1 d$ by (D.17), the standard $\chi^2$ tail bound yields, for any $\varepsilon \in (0, 1)$,

$$\mathbb{P}\Big( \big|S_k - |I_k|\big| > \varepsilon |I_k| \Big) \;\leq\; 2 \exp\Big( -\tfrac{\varepsilon^2}{4} |I_k| \Big) \;\leq\; 2 \, e^{-c_2 d}.$$

Since $K + 1$ is fixed, a union bound gives

$$\mathbb{P}\Big( \forall k = 0, \ldots, K : \; (1 - \varepsilon) |I_k| \leq S_k \leq (1 + \varepsilon) |I_k| \Big) \;\geq\; 1 - e^{-cd} \tag{D.20}$$

for some $c > 0$ independent of $d$.

**Step 4: Comparing $\mu_d(B_k)$ to the bin integral.** On the event in (D.20),

$$8(1-\varepsilon)\sum_{i\in I_k}\lambda_i \;\le\; \mu_d(B_k) \;=\; 8\sum_{i\in I_k}\lambda_i(w_i^\star)^2 \;\le\; 8(1+\varepsilon)\sum_{i\in I_k}\lambda_i.$$

Combining with (D.19) proves (D.14) with

$$C_- := 8(1-\varepsilon)D_-, \qquad C_+ := 8(1+\varepsilon)D_+.$$

**Step 5: Tail.** Summing (D.14) over $\{j \ge m\}$ where $m$ is the unique index such that $\lambda \in (\rho^{m+1}\Lambda_d, \rho^m\Lambda_d]$, and using the geometric form (D.18), yields

$$\mu_d\big((0,\lambda]\big) \;\asymp\; \Lambda_d^{1-\frac{1}{a}}\,\rho^{m(1-\frac{1}{a})} \;\asymp\; \lambda^{1-\frac{1}{a}} \quad \text{(since } \rho^m \asymp \lambda/\Lambda_d\text{)},$$

which is (D.15). The step-function bound (D.16) follows from linearity, as $\int_{(0,\Lambda_d]}\varphi\,d\mu_d = \sum_k a_k\,\mu_d(B_k)$ and the same for the right-hand integrals. $\qquad\square$

The following lemma formally shows that after $T_1'$, $u(t)$ and $s(t)$ cannot go above 1. In particular, it justifies the positivity of $\Delta(t)$.

**Lemma 11** (Post-alignment barrier and correlation bound). *Let $Q = \mathrm{diag}(\lambda_i) \succ 0$ and let $w^\star$ satisfy $\langle w^\star, Qw^\star\rangle = 1$. Consider the population gradient flow for anisotropic phase retrieval with Gaussian inputs, and define*

$$s(t) := \langle w(t), Qw(t)\rangle, \qquad u(t) := \langle w(t), Qw^\star\rangle.$$

*Fix a time $t_0$ (e.g. $t_0 = T_1'$) such that*

$$0 < u(t_0) < 1, \qquad s(t_0) \le 1.$$

*Then:*

1. *$s(t) \le 1$ for all $t \ge t_0$ (forward invariance of $\{s \le 1\}$).*

2. *$0 < u(t) \le 1$ for all $t \ge t_0$, and in fact $u(t) < 1$ for every finite $t \ge t_0$ unless $w(t_0) = w^\star$.*

*Proof.* Work with the $Q$-inner product $\langle x, y\rangle_Q := \langle x, Qy\rangle$. Decompose $w$ into its $Q$-orthogonal parts:

$$w = u\,w^\star + z, \qquad \langle z, Qw^\star\rangle = 0.$$

Then

$$s = \langle w, Qw\rangle = u^2 + \|z\|_Q^2, \qquad \|z\|_Q^2 = s - u^2 \ge 0. \tag{D.21}$$

**Claim 0 (pre-boundary: $u < 1$).** Since $s(t_0) \le 1$, Cauchy–Schwarz in $\langle\cdot,\cdot\rangle_Q$ gives

$$u(t_0) = \langle w(t_0), Qw^\star\rangle \le \|w(t_0)\|_Q\,\|w^\star\|_Q = \sqrt{s(t_0)} \le 1,$$

and by assumption $u(t_0) > 0$; hence $0 < u(t_0) < 1$. Moreover, as long as $s(t) < 1$, the same inequality implies $u(t) \le \sqrt{s(t)} < 1$. Thus *before any potential first time* when $s$ could reach the boundary 1, we indeed have $u(t) < 1$.

**Claim 1 (uniform drift for $s \ge 1$).** Recall that

$$\dot{s} \;=\; 8(1-3s)\,s^{(2)} \;+\; 16u\,u^{(2)}.$$

Using $\lambda_{min}\langle z, Qz\rangle \le s^{(2)}$ and (D.21), when $s \ge 1$ we have $1 - 3s \le -2$, hence

$$\dot{s} \;\le\; 8(1-3s)\,s^{(2)} \;\le\; -16\,s^{(2)} \;\le\; -16\lambda_{min}\,\langle z, Qz\rangle \;=\; -16\lambda_{min}\,\|z\|_Q^2 \;=\; -16\lambda_{min}\,(s-u^2). \tag{D.22}$$

**Claim 2 (no upward crossing at $s = 1$).** Let $t_1 > t_0$ be a *first* time with $s(t_1) = 1$ and $s(t) < 1$ for $t < t_1$ (if no such $t_1$ exists, we are done). By the discussion in Claim 0, we then have $u(t_1^-) \le \sqrt{s(t_1^-)} < 1$, hence $u(t_1) \le 1$ by continuity. By definition of first hitting, there exists $\varepsilon > 0$ with

$s(t) \geq 1$ for all $t \in [t_1, t_1 + \varepsilon)$, so (D.22) applies on $(t_1, t_1 + \varepsilon)$. Taking the right Dini derivative and using continuity of $s - u^2$,

$$D^+ s(t_1) := \limsup_{h \downarrow 0} \frac{s(t_1 + h) - s(t_1)}{h} \leq -16\lambda_{min} \lim_{t \downarrow t_1} \left( s(t) - u(t)^2 \right) = -16\lambda_{min} \left( 1 - u(t_1)^2 \right) \leq 0.$$

Thus the vector field is inward-pointing (nonpositive) at the boundary, which contradicts an upward crossing from $s < 1$ to $s > 1$ at $t_1$. Hence $s(t) \leq 1$ for all $t \geq t_0$.

**Claim 3** ($u \leq 1$ **and strictness**). For all $t$, Cauchy–Schwarz gives

$$|u(t)| \leq \|w(t)\|_Q \, \|w^\star\|_Q = \sqrt{s(t)} \leq 1,$$

so $0 < u(t) \leq 1$ for $t \geq t_0$ (positivity after $T_1'$ follows from the Volterra positivity in Phase I). Finally, unless $w(t_0) = w^\star$, the real-analytic gradient flow reaches the minimizer only as $t \to \infty$, so $u(t) < 1$ for all finite $t \geq t_0$. $\square$

Next, we show $u(t)$ and $s(t) \to 1$.

**Lemma 12** (Convergence of $s(t)$ and $u(t)$). *As $t \to \infty$,*

$$s(t) \to 1, \qquad u(t) \to 1.$$

*Proof.* By Lemma 5, the trajectory $w(t)$ is bounded, hence precompact in $\mathbb{R}^d$. Moreover,

$$\dot{\mathcal{L}}(t) = -\|\nabla \mathcal{L}(w(t))\|^2 \leq 0,$$

so $\mathcal{L}(w(t))$ is nonincreasing and convergent. Phase IIa yields forward invariance of the region $\{s > \frac{1}{3}\}$: there exists $T_1' \geq 0$ such that $s(t) > \frac{1}{3}$ for all $t \geq T_1'$.

By LaSalle's invariance principle, the $\omega$-limit set

$$\Omega := \left\{ \omega : \exists t_n \to \infty \text{ with } w(t_n) \to \omega \right\}$$

is nonempty, compact, invariant, and contained in the largest invariant subset of $\{\nabla \mathcal{L} = 0\} \cap \{s \geq \frac{1}{3}\}$. By the critical-point characterization in Section 3 (Proposition 2), every critical point with $s > \frac{1}{3}$ is of the form $w = \pm w^\star$, under the normalization $\sum_i \lambda_i (w_i^\star)^2 = 1$. In particular, all such critical points satisfy $s = 1$ and $u = \pm 1$.

Hence for any $\omega \in \Omega$ we must have $(s(\omega), u(\omega)) \in \{(1, 1), (1, -1)\}$. Since $s(\cdot), u(\cdot)$ are continuous, it follows that

$$s(t_n) \to 1, \qquad u(t_n) \to \pm 1$$

for every sequence $t_n \to \infty$ with $w(t_n) \to \omega$. Phase I guarantees $u(T_1') > 0$, and positivity is forward-invariant (via the Volterra representation after $T_1'$), so $u(t) > 0$ for all $t \geq T_1'$. Therefore, every limit point must satisfy $u(\omega) = +1$. Thus $(s(t), u(t)) \to (1, 1)$ as $t \to \infty$.

Finally, if $s(t)$ or $u(t)$ did not converge, there would exist $\varepsilon > 0$ and a sequence $t_n \to \infty$ such that either $|s(t_n) - 1| \geq \varepsilon$ or $|u(t_n) - 1| \geq \varepsilon$. By precompactness, we may extract a subsequence $w(t_{n_k}) \to \omega \in \Omega$, but then $(s(t_{n_k}), u(t_{n_k})) \to (1, 1)$, a contradiction. Hence $s(t) \to 1$ and $u(t) \to 1$. $\square$

# E  PHASE III: SCALING LAWS

We first show in Section E.1 that the MSE does not change significantly from initialization. In Section E.2, we then characterize its decay for $t \geq T_2$.

## E.1  STABILITY OF THE MSE FOR $t \leq T_2$

**Proposition 4.** *Let $\sigma_\star^2 = \frac{1}{d} \sum_{i=1}^d (w_i^\star)^2$. Under the assumptions of Theorem 1 we have*

$$\left| \mathrm{MSE}(T_2) - \sigma_\star^2 \right| \lesssim \left( \frac{\varepsilon^{-a}}{d} \right)^{1/3} + \left( \frac{\log d}{d} \right)^{1/3}.$$

*Proof.* **Step 1: Variation of constants.** For each coordinate $i$,

$$\dot{w}_i(t) \;=\; 4\lambda_i\big(1 - 3s(t)\big)\,w_i(t) \;+\; 8\lambda_i\,u(t)\,w_i^\star, \qquad \Theta(t) := \int_0^t (1 - 3s(\tau))\,d\tau.$$

By Duhamel's formula, for all $t \geq 0$,

$$w_i(t) \;=\; e^{4\lambda_i\Theta(t)}w_i(0) \;+\; 8\lambda_i w_i^\star \int_0^t e^{4\lambda_i(\Theta(t)-\Theta(\tau))}\,u(\tau)\,d\tau. \tag{E.1}$$

**Step 2: Phase clocks and the key split.** Let $T_1'$ be the first time $s(t) = \tfrac{1}{3}$. Then $\Theta$ increases on $[0, T_1']$ and decreases on $[T_1', \infty)$. Define

$$\Theta_{\max} := \Theta(T_1'), \qquad \Lambda(t) := \Theta_{\max} - \Theta(t), \quad t \geq T_1'.$$

Fix $t \geq T_1'$. Splitting the integral in (E.1) at $T_1'$ and changing variables to $\Theta$ on each side, and using $0 \leq u \leq 1$, we obtain the bound

$$\int_0^t e^{4\lambda_i(\Theta(t)-\Theta(\tau))}\,d\tau \;\leq\; C\left(\frac{1 - e^{-4\lambda_i\Theta_{\max}}}{\lambda_i} + \frac{1 - e^{-4\lambda_i\Lambda(t)}}{\lambda_i}\right), \tag{E.2}$$

where $C = C(\delta, s_0)$. This refines earlier estimates by accounting for the change of sign in $\Theta(t) - \Theta(\tau)$ across phases.

**Step 3: Frontier at $T_2$ and inactive bound.** At time $T_2$, fix $\zeta \in (0, 1]$ and define the index sets

$$\mathcal{I}_\zeta(T_2) := \{\, i : \; 4\lambda_i\,\Lambda(T_2) \geq \zeta \,\}, \qquad \mathcal{I}_\zeta^c(T_2) := [d] \setminus \mathcal{I}_\zeta(T_2).$$

For $i \in \mathcal{I}_\zeta^c(T_2)$, $4\lambda_i\Lambda(T_2) \leq \zeta$, so $1 - e^{-4\lambda_i\Lambda(T_2)} \leq 4\lambda_i\Lambda(T_2)$. Combining (E.1) and (E.2), and after absorbing constants,

$$|w_i(T_2)| \;\leq\; e^{4\lambda_i\Theta(T_2)}|w_i(0)| + 8\lambda_i|w_i^\star| \cdot C\left(\tfrac{1-e^{-4\lambda_i\Theta_{\max}}}{\lambda_i} + 4\Lambda(T_2)\right) \;\leq\; e^\zeta|w_i(0)| + C'\zeta\,|w_i^\star|. \tag{E.3}$$

**Step 4: Active correlation and energy.** On $\mathcal{I}_\zeta(T_2)$, $\lambda_i \geq \zeta/(4\Lambda(T_2))$. Applying the weighted Cauchy–Schwarz inequality $\sum_i |x_iy_i| \leq (\sum x_i^2/\lambda_i)^{1/2}(\sum \lambda_i y_i^2)^{1/2}$, we get

$$\sum_{i\in\mathcal{I}_\zeta(T_2)} |w_i^\star|\,|w_i(T_2)| \;\leq\; \Big(\sum_{i\in\mathcal{I}_\zeta(T_2)} \tfrac{(w_i^\star)^2}{\lambda_i}\Big)^{1/2}\Big(\sum_{i\in\mathcal{I}_\zeta(T_2)} \lambda_i\,w_i(T_2)^2\Big)^{1/2} \;\leq\; \sqrt{\tfrac{4\Lambda(T_2)}{\zeta}}\,\sqrt{d\,\sigma_\star^2}\,\sqrt{s(T_2)}.$$

Hence the active correlation contribution satisfies

$$\frac{2}{d}\Big|\sum_{i\in\mathcal{I}_\zeta(T_2)} w_i^\star w_i(T_2)\Big| \;\lesssim\; \sqrt{\frac{\Lambda(T_2)}{\zeta\,d}}. \tag{E.4}$$

Similarly, for the active energy,

$$\frac{1}{d}\sum_{i\in\mathcal{I}_\zeta(T_2)} w_i(T_2)^2 \;\leq\; \frac{1}{d\,\lambda_{\min}(\mathcal{I}_\zeta(T_2))}\sum_{i\in\mathcal{I}_\zeta(T_2)} \lambda_i\,w_i(T_2)^2 \;\leq\; \frac{4\Lambda(T_2)}{\zeta\,d}\,s(T_2) \;\lesssim\; \frac{\Lambda(T_2)}{\zeta\,d}, \tag{E.5}$$

since $s(T_2) \leq 1$.

**Step 5: Inactive correlation and energy.** From (E.3) and Cauchy–Schwarz, and using $\|w^\star\|_2^2 = d\,\sigma_\star^2$,

$$\sum_{i\in\mathcal{I}_\zeta^c(T_2)} |w_i^\star|\,|w_i(T_2)| \;\leq\; \|w^\star\|_2 \Big(\sum_{i\in\mathcal{I}_\zeta^c(T_2)} w_i(T_2)^2\Big)^{1/2} \;\leq\; \sqrt{d\,\sigma_\star^2}\Big(2e^{2\zeta}\|w(0)\|_2^2 + 8\zeta^2\,d\,\sigma_\star^2\Big)^{1/2}.$$

Therefore,

$$\frac{2}{d}\sum_{i\in\mathcal{I}_\zeta^c(T_2)} |w_i^\star|\,|w_i(T_2)| \;\lesssim\; \zeta \;+\; d^{-1/2}. \tag{E.6}$$

Moreover,

$$\frac{1}{d} \sum_{i \in \mathcal{I}_\zeta^c(T_2)} w_i(T_2)^2 \ \leq\ \frac{2e^{2\zeta}}{d} \|w(0)\|_2^2 + 8\zeta^2 \sigma_\star^2 \ \lesssim\ \zeta^2 + d^{-1}. \tag{E.7}$$

**Step 6: Combination and optimization.** Since

$$\mathrm{MSE}(T_2) - \sigma_\star^2 = -\frac{2}{d} \sum_i w_i^\star w_i(T_2) + \frac{1}{d} \sum_i w_i(T_2)^2,$$

combining (E.4)–(E.7) yields

$$\left| \mathrm{MSE}(T_2) - \sigma_\star^2 \right| \ \lesssim\ \underbrace{\sqrt{\frac{\Lambda(T_2)}{\zeta\, d}} + \frac{\Lambda(T_2)}{\zeta\, d}}_{\text{active}} + \underbrace{\zeta + \zeta^2}_{\text{inactive}} + d^{-1/2}. \tag{E.8}$$

The right-hand side is minimized (up to constants) by taking

$$\zeta^\star \ \asymp\ \left(\frac{\Lambda(T_2)}{d}\right)^{1/3} \wedge 1,$$

which balances the active and inactive contributions. For this choice,

$$\sqrt{\frac{\Lambda(T_2)}{\zeta^\star d}} \ \asymp\ \frac{\Lambda(T_2)}{\zeta^\star d} \ \asymp\ \zeta^\star \ \asymp\ \left(\frac{\Lambda(T_2)}{d}\right)^{1/3}.$$

Therefore,

$$\left| \mathrm{MSE}(T_2) - \sigma_\star^2 \right| \ \lesssim\ \left(\frac{\Lambda(T_2)}{d}\right)^{1/3} + d^{-1/2}.$$

Finally, the Phase IIb analysis (via the Volterra–renewal argument) gives $1 - u(t) \asymp \Lambda(t)^{-1/a}$. At $t = T_2$, $1 - u(T_2) = \varepsilon$, hence $\Lambda(T_2) \asymp \varepsilon^{-a}$. Substituting into the above bound gives

$$\left| \mathrm{MSE}(T_2) - \sigma_\star^2 \right| \ \lesssim\ \left(\frac{\varepsilon^{-a}}{d}\right)^{1/3} + d^{-1/2}.$$

Adding the Phase I "plateau" contribution $(\log d / d)^{1/3}$ yields the stated result. $\qquad\square$

### E.2 Evolution of the MSE for $t \geq T_2$

We first analyze the idealized dynamics where $u \equiv s \equiv 1$ (Section E.2.1), and then control the approximation error in Section E.2.2.

#### E.2.1 Ideal case where $u \equiv s \equiv 1$

We begin by studying the evolution of the MSE under the idealization $u \equiv s \equiv 1$.

**Lemma 13.** *In the post-$T_2$ idealization ($s \equiv u \equiv 1$), the coordinate errors $e_i(t) := w_i(t) - w_i^\star$ evolve as*

$$e_i(T_2 + \tau) \ =\ e_i(T_2)\, e^{-8\lambda_i \tau} \qquad (\tau \geq 0).$$

*Consequently,*

$$\mathrm{MSE}(T_2 + \tau) \ =\ \frac{1}{d} \sum_{i=1}^d e_i(T_2)^2\, e^{-16\lambda_i \tau} \ =\ \mathrm{MSE}(T_2)\, \widehat{S}_d(\tau),$$

*where the* normalized spectral mixing curve *is*

$$\widehat{S}_d(\tau) \ :=\ \sum_{i=1}^d \pi_i\, e^{-16\lambda_i \tau}, \qquad \pi_i \ :=\ \frac{e_i(T_2)^2}{\sum_{j=1}^d e_j(T_2)^2}, \qquad \sum_{i=1}^d \pi_i = 1.$$

*Proof.* Setting $s \equiv u \equiv 1$ in the gradient flow gives

$$\dot{w}(t) = -8Q\big(w(t) - w^\star\big).$$

Defining $e(t) = w(t) - w^\star$ yields $\dot{e}(t) = -8Qe(t)$. Since $Q$ is diagonal with entries $(\lambda_i)$, each coordinate satisfies $\dot{e}_i(t) = -8\lambda_i e_i(t)$, hence

$$e_i(T_2 + \tau) = e_i(T_2)e^{-8\lambda_i \tau}.$$

Squaring and averaging proves the formula for the MSE. Factoring $\mathrm{MSE}(T_2)$ defines the weights $\pi_i$, and the claim follows. □

Since $\widehat{S}_d$ depends on the (random) weights $\pi_i$, we introduce the proxy

$$S_d(\tau) := \frac{1}{d} \sum_{i=1}^{d} e^{-\beta_d \tau\, i^{-a}}, \qquad \beta_d = 16 L_d,$$

which corresponds to uniform weights. The next result shows that this is a valid approximation.

**Proposition 15.** *Assume that the teacher coordinates $(w_i^\star)_{i=1}^d$ are i.i.d. sub-Gaussian (in the eigenbasis of $Q$), and independent of the initialization. There exists a constant $C < \infty$, independent of $d$, such that with high probability, for every $\tau \geq 0$,*

$$\big|\widehat{S}_d(\tau) - S_d(\tau)\big| \leq C\Big(\zeta^\star + \sqrt{\frac{\log d}{d}}\Big)e^{-\beta_d \tau d^{-a}}. \tag{E.9}$$

*Proof.* Fix $\tau \geq 0$ and write $f_i := e^{-16\lambda_i \tau} = e^{-\beta_d \tau\, i^{-a}}$. Since $(\lambda_i)$ is nonincreasing in $i$, the sequence $(f_i)_{i=1}^d$ is nondecreasing.

Let $P_k := \sum_{i=1}^k \pi_i$. A standard summation-by-parts identity yields

$$\widehat{S}_d(\tau) - S_d(\tau) = -\sum_{k=1}^{d-1} \Big(P_k - \frac{k}{d}\Big)\big(f_{k+1} - f_k\big).$$

Since $\sum_{k=1}^{d-1}(f_{k+1} - f_k) = f_d - f_1$, we obtain

$$\big|\widehat{S}_d(\tau) - S_d(\tau)\big| \leq (f_d - f_1) \sup_{1 \leq k \leq d} \Big|P_k - \frac{k}{d}\Big|. \tag{E.10}$$

Noting that $f_d - f_1 = e^{-\beta_d \tau d^{-a}} - e^{-\beta_d \tau}$ gives the prefactor in (E.9).

Define

$$\pi_i^\star := \frac{(w_i^\star)^2}{\|w^\star\|^2}, \qquad P_k^\star := \sum_{i=1}^k \pi_i^\star.$$

Then

$$\sup_k \Big|P_k - \frac{k}{d}\Big| \leq \sup_k |P_k - P_k^\star| + \sup_k \Big|P_k^\star - \frac{k}{d}\Big|.$$

Write $Z := \sum_{j=1}^d e_j(T_2)^2 = d\,\mathrm{MSE}(T_2)$, so $\pi_i = e_i(T_2)^2/Z$. Using $e_i(T_2)^2 - (w_i^\star)^2 = w_i(T_2)^2 - 2w_i^\star w_i(T_2)$ and Cauchy–Schwarz,

$$\sum_{i=1}^d |e_i(T_2)^2 - (w_i^\star)^2| \leq \sum_{i=1}^d w_i(T_2)^2 + 2\|w^\star\|_2 \|w(T_2)\|_2.$$

From (E.5) and (E.7) with $\zeta = \zeta^\star$ we have

$$\frac{1}{d} \sum_{i=1}^d w_i(T_2)^2 \lesssim (\zeta^\star)^2, \qquad \text{hence} \qquad \frac{1}{d}\|w^\star\|_2 \|w(T_2)\|_2 \lesssim \zeta^\star,$$

so altogether $\sum_i |e_i(T_2)^2 - (w_i^\star)^2| \lesssim d\,\zeta^\star$. Moreover, Proposition 4 gives $\mathrm{MSE}(T_2) = \sigma_\star^2 + o(1)$, hence $Z \asymp d$ with high probability. Therefore

$$\|\pi - \pi^\star\|_1 = \sum_{i=1}^{d} \left| \frac{e_i(T_2)^2}{Z} - \frac{(w_i^\star)^2}{\|w^\star\|^2} \right| \lesssim \zeta^\star,$$

and thus $\sup_k |P_k - P_k^\star| \le \|\pi - \pi^\star\|_1 \lesssim \zeta^\star$.

Since $((w_i^\star)^2)_i$ are i.i.d. sub-exponential with mean $\sigma_\star^2$, Bernstein's inequality and a union bound over $k = 1, \ldots, d$ yield

$$\sup_{1 \le k \le d} \left| \sum_{i=1}^{k} (w_i^\star)^2 - \frac{k}{d} \sum_{j=1}^{d} (w_j^\star)^2 \right| \lesssim \sqrt{d \log d}$$

with high probability. Dividing by $\sum_j (w_j^\star)^2 \asymp d$ gives

$$\sup_{1 \le k \le d} \left| P_k^\star - \frac{k}{d} \right| \lesssim \sqrt{\frac{\log d}{d}}$$

with high probability.

Combining the previous bounds with (E.10) yields (E.9). Taking $\sup_{\tau \ge 0}$ and using $f_d - f_1 \le 1$ gives the result. $\qquad\square$

We now analyze the asymptotics of $S_d(\tau)$.

**Proposition 16** (Asymptotics of the spectral average)**.** *Let $a > 1$ and set $x_d := (\beta_d \tau)^{1/a}$. Then $S_d(\tau)$ satisfies:*

1. *Early time ($\beta_d \tau \ll 1$):*

$$S_d(\tau) = 1 - \frac{\beta_d \tau}{d} \sum_{i=1}^{d} i^{-a} + \frac{(\beta_d \tau)^2}{2d} \sum_{i=1}^{d} i^{-2a} + O\left( \frac{(\beta_d \tau)^3}{d} \right).$$

   *Under trace normalization $L_d = 1/H_{d,a}$ this simplifies to*

$$S_d(\tau) = 1 - \tfrac{16}{d}\tau + O\left( \tfrac{\tau^2}{d} \right).$$

2. *Mesoscopic window ($1 \ll x_d \ll d$):*

$$S_d(\tau) = 1 - \Gamma\left(1 - \tfrac{1}{a}\right) \frac{x_d}{d} + o\left( \frac{x_d}{d} \right).$$

3. *Late time ($x_d \gtrsim d$):*

$$S_d(\tau) \le \exp\left( -\beta_d \tau\, d^{-a} \right).$$

*Proof.* (i) For $z \in [0,1]$, $e^{-z} = 1 - z + z^2/2 + R_3(z)$ with $|R_3(z)| \le z^3/6$. If $\beta_d \tau \ll 1$, all $z_i = \beta_d \tau i^{-a}$ are small, and averaging the expansion over $i = 1, \ldots, d$ yields the stated series.

(ii) Writing

$$1 - S_d(\tau) = \frac{1}{d} \sum_{i=1}^{d} \left( 1 - e^{-(x_d/i)^a} \right),$$

and defining $g_x(t) = 1 - e^{-(x/t)^a}$, we have

$$\int_0^\infty g_x(t)\, dt = \int_0^\infty \left( 1 - e^{-(x/t)^a} \right) dt = x\, \Gamma\left(1 - \tfrac{1}{a}\right),$$

by the substitution $y = (x/t)^a$. Since $g_x$ is monotone decreasing in $t$,

$$\int_1^d g_x(t)\, dt \le \sum_{i=1}^{d} g_x(i) \le g_x(1) + \int_1^d g_x(t)\, dt.$$

Thus

$$\sum_{i=1}^{d} g_x(i) = x\,\Gamma(1 - 1/a) + O(1) + O(x^a d^{1-a}).$$

Dividing by $d$ proves the expansion for $1 \ll x_d \ll d$.

(iii) For $i \le d$, $i^{-a} \ge d^{-a}$, hence $e^{-\beta_d \tau i^{-a}} \le e^{-\beta_d \tau d^{-a}}$. Averaging over $i$ gives the bound. $\qquad\square$

**Corollary 3.** *With high probability, uniformly for all $\tau \ge 0$,*

$$\mathrm{MSE}(T_2 + \tau) = \mathrm{MSE}(T_2)\,\widehat{S}_d(\tau) = \mathrm{MSE}(T_2)\,S_d(\tau) + O\Big(\mathrm{MSE}(T_2)\big(\zeta^\star + \sqrt{\tfrac{\log d}{d}}\big)\big(e^{-\beta_d \tau d^{-a}} - e^{-\beta_d \tau}\big)\Big).$$

*In particular, in any regime where*

$$S_d(\tau) \gg \big(\zeta^\star + \sqrt{\tfrac{\log d}{d}}\big)\big(e^{-\beta_d \tau d^{-a}} - e^{-\beta_d \tau}\big),$$

*we have*

$$\mathrm{MSE}(T_2 + \tau) = \mathrm{MSE}(T_2)\,S_d(\tau)\,(1 + o(1)).$$

**Remark 7.** *Recall that for $x_d := (\beta_d \tau)^{1/a}$ and $s_d := \beta_d \tau d^{-a} = (x_d/d)^a$, we have in the late-time regime $s_d \to \infty$,*

$$S_d(\tau) \sim \frac{e^{-s_d}}{a\,s_d}.$$

*On the other hand, the correction term in Corollary 3 is of order*

$$\big(\zeta^\star + \sqrt{\tfrac{\log d}{d}}\big)e^{-s_d}.$$

*Hence the multiplicative condition*

$$S_d(\tau) \gg \big(\zeta^\star + \sqrt{\tfrac{\log d}{d}}\big)\big(e^{-\beta_d \tau d^{-a}} - e^{-\beta_d \tau}\big)$$

*reduces, in the regime $s_d \gg 1$, to*

$$\frac{1}{s_d} \gg \zeta^\star + \sqrt{\tfrac{\log d}{d}}.$$

*In particular, if $\zeta^\star \asymp (\log d/d)^{1/3}$ (as in (E.8)), this condition holds as soon as*

$$s_d \ll \Big(\frac{d}{\log d}\Big)^{1/3}, \qquad \textit{i.e.} \qquad \tau \ll \frac{d^a}{\beta_d}\Big(\frac{d}{\log d}\Big)^{1/3}.$$

*When $\beta_d \asymp d^a$, this corresponds to*

$$\tau \ll \Big(\frac{d}{\log d}\Big)^{1/3}, \qquad \textit{equivalently} \qquad t \ll d^{a+1/3}(\log d)^{-1/3}.$$

### E.2.2 HANDLING THE APPROXIMATION TERM

Let $\delta_s(t) = 1 - s(t)$ and $\delta_u(t) = 1 - u(t)$. We will first show that the convergence approximation remains of order $\varepsilon$ after $T_2$.

**Lemma 14.** *Assume Phase IIb yields, at time $T_2$,*

$$|\delta_s(T_2)| + |\delta_u(T_2)| \le \varepsilon. \tag{E.11}$$

*Then there exists a constant $C > 0$ and $\varepsilon_0 > 0$ such that, if $\varepsilon \le \varepsilon_0$, then for all $\tau \ge 0$,*

$$|\delta_s(T_2 + \tau)| + |\delta_u(T_2 + \tau)| \le C\,\varepsilon. \tag{E.12}$$

*Proof.* Write $\tau = t - T_2$. After $T_2$, the full flow satisfies

$$\dot{w}(t) = -8Qw(t) + (8\delta_u(t) - 12\delta_s(t))Qw^\star - 12\,\delta_s(t)\,Qw(t).$$

This is a linear time-varying system. Its mild form (variation of constants) is

$$w(T_2 + \tau) = e^{-8Q\tau}w(T_2) + \int_0^\tau e^{-8Q(\tau - s)}\Big[(8\delta_u - 12\delta_s)(T_2 + s)\,Qw^\star - 12\,\delta_s(T_2 + s)\,Qw(T_2 + s)\Big]ds. \tag{E.13}$$

Set

$$A(\tau) := e^{-8Q\tau} w(T_2), \qquad B(\tau) := \int_0^\tau e^{-8Q(\tau-s)} \Big[ (8\delta_u - 12\delta_s)(T_2+s) \, Qw^\star - 12 \, \delta_s(T_2+s) \, Qw(T_2+s) \Big] ds,$$

so that $w(T_2 + \tau) = A(\tau) + B(\tau)$. We can interpret $A(\tau)$ as the term giving the dynamics in the ideal case, and $B(\tau)$ as a perturbation term.

*Step 1. (Uniform semigroup bounds).* Since $Q \succ 0$,

$$\int_0^\infty e^{-8Qr} Q \, dr = \frac{1}{8} I, \qquad \int_0^\infty Q^{1/2} e^{-8Qr} Q \, dr = \frac{1}{8} Q^{1/2}.$$

Hence for any bounded $h$ and any vector $v$,

$$\Big\| \int_0^\tau e^{-8Q(\tau-s)} h(s) Qv \, ds \Big\|_2 \le \tfrac{1}{8} \|h\|_\infty \|v\|_2, \tag{E.14}$$

$$\Big\| Q^{1/2} \int_0^\tau e^{-8Q(\tau-s)} h(s) Qv \, ds \Big\|_2 \le \tfrac{1}{8} \|h\|_\infty \|Q^{1/2} v\|_2. \tag{E.15}$$

*Step 2. (Bounding $B(\tau)$).* Let $g(s) := 8\delta_u(T_2+s) - 12\delta_s(T_2+s)$. On $[T_2, T_2+\tau]$, $|g(s)| \le 20 \, G(\tau)$ where

$$G(\tau) := \sup_{0 \le t \le \tau} (|\delta_s(T_2 + t)| + |\delta_u(T_2 + t)|).$$

By (E.14)–(E.15),

$$\|B(\tau)\|_2 \le c_1 \, G(\tau), \qquad \|Q^{1/2} B(\tau)\|_2 \le c_2 \, G(\tau), \tag{E.16}$$

for some absolute constants $c_1, c_2$.

*Step 3. (Comparing with the ideal trajectory).* The "ideal" evolution keeps only the base drift:

$$w^{\text{ideal}}(T_2 + \tau) := w^\star + e^{-8Q\tau}(w(T_2) - w^\star) = w^\star + e^{-8Q\tau} e(T_2),$$

so that

$$s^{\text{ideal}}(T_2 + \tau) = \|Q^{1/2} w^{\text{ideal}}(T_2 + \tau)\|_2^2, \qquad u^{\text{ideal}}(T_2 + \tau) = \langle w^{\text{ideal}}(T_2 + \tau), Qw^\star \rangle.$$

Because $e^{-8Q\tau}$ is a contraction,

$$|1 - s^{\text{ideal}}(T_2 + \tau)| \le |\delta_s(T_2)| \le \varepsilon, \qquad |1 - u^{\text{ideal}}(T_2 + \tau)| \le |\delta_u(T_2)| \le \varepsilon.$$

*Step 4. (Deviations of $s$ and $u$).* From $w(T_2 + \tau) = A(\tau) + B(\tau)$,

$$s(T_2 + \tau) = \|Q^{1/2}(A(\tau) + B(\tau))\|_2^2 = s^{\text{ideal}}(T_2 + \tau) + 2\langle Q^{1/2} w^{\text{ideal}}(T_2 + \tau), Q^{1/2} B(\tau) \rangle + \|Q^{1/2} B(\tau)\|_2^2,$$

$$u(T_2 + \tau) = \langle A(\tau) + B(\tau), Qw^\star \rangle = u^{\text{ideal}}(T_2 + \tau) + \langle B(\tau), Qw^\star \rangle.$$

Hence,

$$|\delta_s(T_2 + \tau)| \le \varepsilon + 2 \|Q^{1/2} B(\tau)\|_2 + \|Q^{1/2} B(\tau)\|_2^2, \tag{E.17}$$

$$|\delta_u(T_2 + \tau)| \le \varepsilon + \|B(\tau)\|_2 \|Qw^\star\|_2 \le \varepsilon + c_3 \, G(\tau), \tag{E.18}$$

using (E.16).

*Step 5.* Combining (E.17)–(E.18) and the definition of $G(\tau)$ gives

$$G(\tau) \le 2\varepsilon + a \, G(\tau) + b \, G(\tau)^2, \qquad a := 2c_2 + c_3, \quad b := c_2^2.$$

Fix $M \ge 3$ and set $\varepsilon_0 := \min\{(M-2)/(2aM), (M-2)/(2bM^2)\}$. If $\varepsilon \le \varepsilon_0$, define $\tau^\star := \sup\{\tau : G(\tau) \le M\varepsilon\}$. Then on $[0, \tau^\star]$,

$$G(\tau) \le 2\varepsilon + aM\varepsilon + bM^2\varepsilon^2 \le M\varepsilon.$$

By continuity, $\tau^\star = \infty$, so $G(\tau) \le M\varepsilon$ for all $\tau$. Setting $C := M$ gives

$$|\delta_s(T_2 + \tau)| + |\delta_u(T_2 + \tau)| \le C\varepsilon, \quad \forall \tau \ge 0.$$

$\square$

Let $\tau = t - T_2(\varepsilon)$ and set

$$\mathrm{MSE}^{\mathrm{full}}(T_2 + \tau) := \frac{1}{d}\|e(T_2 + \tau)\|_2^2, \qquad \mathrm{MSE}^{\mathrm{ideal}}(T_2 + \tau) := \frac{1}{d}\sum_{i=1}^{d} e_i(T_2)^2\, e^{-16\lambda_i \tau}.$$

Define $\Delta e(\tau) := e^{\mathrm{full}}(T_2 + \tau) - e^{\mathrm{ideal}}(T_2 + \tau)$. Subtracting the full and ideal flows yields, for $\tau \geq 0$,

$$\Delta e(\tau) = \int_0^\tau e^{-8Q(\tau - s)}\Big(-12\,\delta_s(T_2 + s)\, Q\big(e^{\mathrm{ideal}}(T_2 + s) + \Delta e(s)\big) + \big(8\delta_u - 12\delta_s\big)(T_2 + s)\, Qw^\star\Big)\, ds. \tag{E.19}$$

**Theorem 5.** *Assume (E.12) holds. There exist absolute constants $A, B, C < \infty$ such that, for all $\tau \geq 0$,*

$$\left| \mathrm{MSE}^{\mathrm{full}}(T_2 + \tau) - \mathrm{MSE}^{\mathrm{ideal}}(T_2 + \tau) \right| \leq A\varepsilon \Big(\mathrm{MSE}^{\mathrm{ideal}}(T_2) - \mathrm{MSE}^{\mathrm{ideal}}(T_2 + \tau)\Big)$$

$$+ B\varepsilon^2 \frac{1}{d}\sum_{i=1}^{d}\big(1 - e^{-8\lambda_i \tau}\big)^2 (w_i^\star)^2 + C\varepsilon F(\tau), \tag{E.20}$$

*where $F(\tau) := \sup_{0 \leq s \leq \tau}\|\Delta e(s)\|_2/\sqrt{d}$. In particular, if $\varepsilon \leq \varepsilon_0$ is small enough so that $C\varepsilon \leq \frac{1}{2}$, then the feedback term can be absorbed to give*

$$\left| \mathrm{MSE}^{\mathrm{full}}(T_2 + \tau) - \mathrm{MSE}^{\mathrm{ideal}}(T_2 + \tau) \right| \leq \frac{A}{1 - C\varepsilon}\, \varepsilon \Big(\mathrm{MSE}^{\mathrm{ideal}}(T_2) - \mathrm{MSE}^{\mathrm{ideal}}(T_2 + \tau)\Big) \tag{E.21}$$

$$+ \frac{B}{1 - C\varepsilon}\, \varepsilon^2 \frac{1}{d}\sum_{i=1}^{d}\big(1 - e^{-8\lambda_i \tau}\big)^2 (w_i^\star)^2, \tag{E.22}$$

*and, e.g., for $\varepsilon \leq 1/3$, $(1 - C\varepsilon)^{-1} \leq 2$ (after adjusting $C$ if needed).*

*Proof. (i) Term with $Qe^{\mathrm{ideal}}$.* Using $e^{\mathrm{ideal}}(T_2 + s) = e^{-8Qs}\, e(T_2)$ and semigroup commutation,

$$\int_0^\tau e^{-8Q(\tau - s)}Qe^{\mathrm{ideal}}(T_2 + s)\, ds = \tau\, e^{-8Q\tau}\, Qe(T_2).$$

Taking inner product with $e^{\mathrm{ideal}}(T_2 + \tau) = e^{-8Q\tau}e(T_2)$, dividing by $d$, and using $xe^{-x} \leq 1 - e^{-x}$ with $x = 16\lambda_i \tau$,

$$\frac{2}{d}\Big\langle e^{\mathrm{ideal}}(T_2 + \tau), \int_0^\tau e^{-8Q(\tau - s)}Qe^{\mathrm{ideal}}(T_2 + s)\, ds \Big\rangle \leq \frac{1}{8}\Big(\mathrm{MSE}^{\mathrm{ideal}}(T_2) - \mathrm{MSE}^{\mathrm{ideal}}(T_2 + \tau)\Big).$$

Multiplying by $|\delta_s| \leq \varepsilon$ yields the first term in (E.20) with $A = \frac{1}{8}$.

*(ii) Teacher–forcing term.* Let

$$\mathcal{T}(\tau) := \int_0^\tau e^{-8Q(\tau - s)}\big(8\delta_u - 12\delta_s\big)(T_2 + s)\, Qw^\star\, ds, \quad g(s) := 8\delta_u(T_2 + s) - 12\delta_s(T_2 + s).$$

Then $|g(s)| \leq 20\varepsilon$ and

$$\mathcal{T}_i(\tau) = \lambda_i w_i^\star \int_0^\tau g(s)\, e^{-8\lambda_i(\tau - s)}\, ds, \qquad \left| \int_0^\tau g(s)\, e^{-8\lambda_i(\tau - s)}\, ds \right| \leq 20\varepsilon\, \frac{1 - e^{-8\lambda_i \tau}}{8\lambda_i}.$$

Hence

$$\frac{1}{d}\|\mathcal{T}(\tau)\|_2^2 \leq \frac{25}{4}\,\varepsilon^2 \frac{1}{d}\sum_{i=1}^{d}\big(1 - e^{-8\lambda_i \tau}\big)^2 (w_i^\star)^2,$$

which yields the second term in (E.20) with $B = \frac{25}{4}$. The bound is *saturating*: for small $\tau$, one may use $1 - e^{-8\lambda_i \tau} \leq 8\lambda_i \tau$ to recover the quadratic behavior

$$\frac{1}{d}\|\mathcal{T}(\tau)\|_2^2 \leq 400\,\varepsilon^2\, \tau^2 \frac{1}{d}\sum_{i=1}^{d}\lambda_i^2 (w_i^\star)^2,$$

whereas for large $\tau$ it saturates at $O(\varepsilon^2) \cdot \frac{1}{d} \sum_i (w_i^\star)^2$.

*(iii) Self–interaction term.* Write

$$\mathcal{S}(\tau) := \int_0^\tau e^{-8Q(\tau-s)} \big( -12 \, \delta_s(T_2 + s) \big) Q \, \Delta e(s) \, ds.$$

For each coordinate $i$,

$$\mathcal{S}_i(\tau) = -12 \int_0^\tau \lambda_i e^{-8\lambda_i(\tau-s)} \delta_s(T_2 + s) \Delta e_i(s) \, ds.$$

Using $|\delta_s| \leq \varepsilon$ and the uniform $L^1$ bound $\int_0^\infty \lambda_i e^{-8\lambda_i u} \, du = \frac{1}{8}$, Young's inequality gives

$$\frac{\|\mathcal{S}(\tau)\|_2}{\sqrt{d}} \leq \frac{3}{2} \varepsilon \sup_{0 \leq s \leq \tau} \frac{\|\Delta e(s)\|_2}{\sqrt{d}}.$$

Thus the self–interaction contributes a *linear feedback* term $C\varepsilon \, F(\tau)$ in (E.20) with $C = \frac{3}{2}$. Absorbing this term to the left yields (E.21) whenever $C\varepsilon < 1$. $\qquad \square$

**Corollary 4.** *Under (E.12) and for $\varepsilon \leq \varepsilon_0$ small enough,*

$$\left| \mathrm{MSE}^{\mathrm{full}}(T_2 + \tau) - \mathrm{MSE}^{\mathrm{ideal}}(T_2 + \tau) \right| \leq C_1 \varepsilon \left( \mathrm{MSE}^{\mathrm{ideal}}(T_2) - \mathrm{MSE}^{\mathrm{ideal}}(T_2 + \tau) \right) + C_2 \frac{\varepsilon^2}{d} s_\star^{(2)}.$$

*In particular, for each fixed $\tau$ the difference is $O(\varepsilon)$ relative to the* drop *of the ideal MSE from $T_2$ to $T_2 + \tau$, plus a negligible $O(\varepsilon^2/d)$ additive term.*

# F ADDITIONAL NUMERICAL EXPERIMENTS

## F.1 APPROXIMATION $\dot{w}_i(t) \approx 8\lambda_i(w_i^\star - w_i(t))$.

We empirically confirm the approximation used for the Phase III analysis. In Figure 4 we compare the MSE of the population dynamic with the one predicted by the approximation $\dot{w}_i(t) \approx 8\lambda_i(w_i^\star - w_i(t))$ (i.e. we froze $u(t)$ and $s(t)$ at their convergence value) where $T_2$ is chosen such that $u(T_2) = 0.9$.

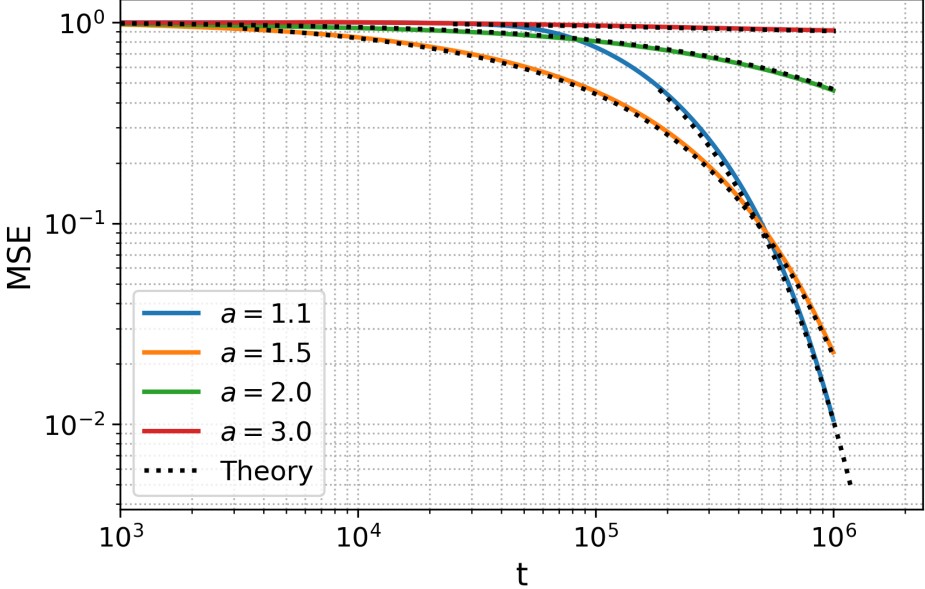

Figure 4: Comparison between MSE obtained from the population dynamic, and the approximated one used to derive scaling law (th.) (log-log scale). Parameters: $d = 300, \eta = 10^{-3}$.

### F.2 ONLINE SGD

In the main text, we focused on the gradient flow to highlight the phase decomposition. Here, we complement the analysis with simulations of online SGD, see Figure 5. We track the trajectories of the key order parameters when training with a small learning rate and without mini-batching. We used the same parameters as in the previous section, but used empirical gradient and added noise $\varepsilon_t \sim \mathcal{N}(0, 0.05)$.

Compared to gradient flow, online SGD exhibits the same qualitative phases, but with additional noise. This suggests that our theoretical scaling laws are robust to stochastic perturbations introduced by SGD. Moreover, the magnitude of fluctuations increases with the spectral decay parameter $a$. Intuitively, when $a$ is large, only a few top directions dominate the signal, so SGD updates concentrate heavily along these directions, amplifying variance and making the trajectory noisier.

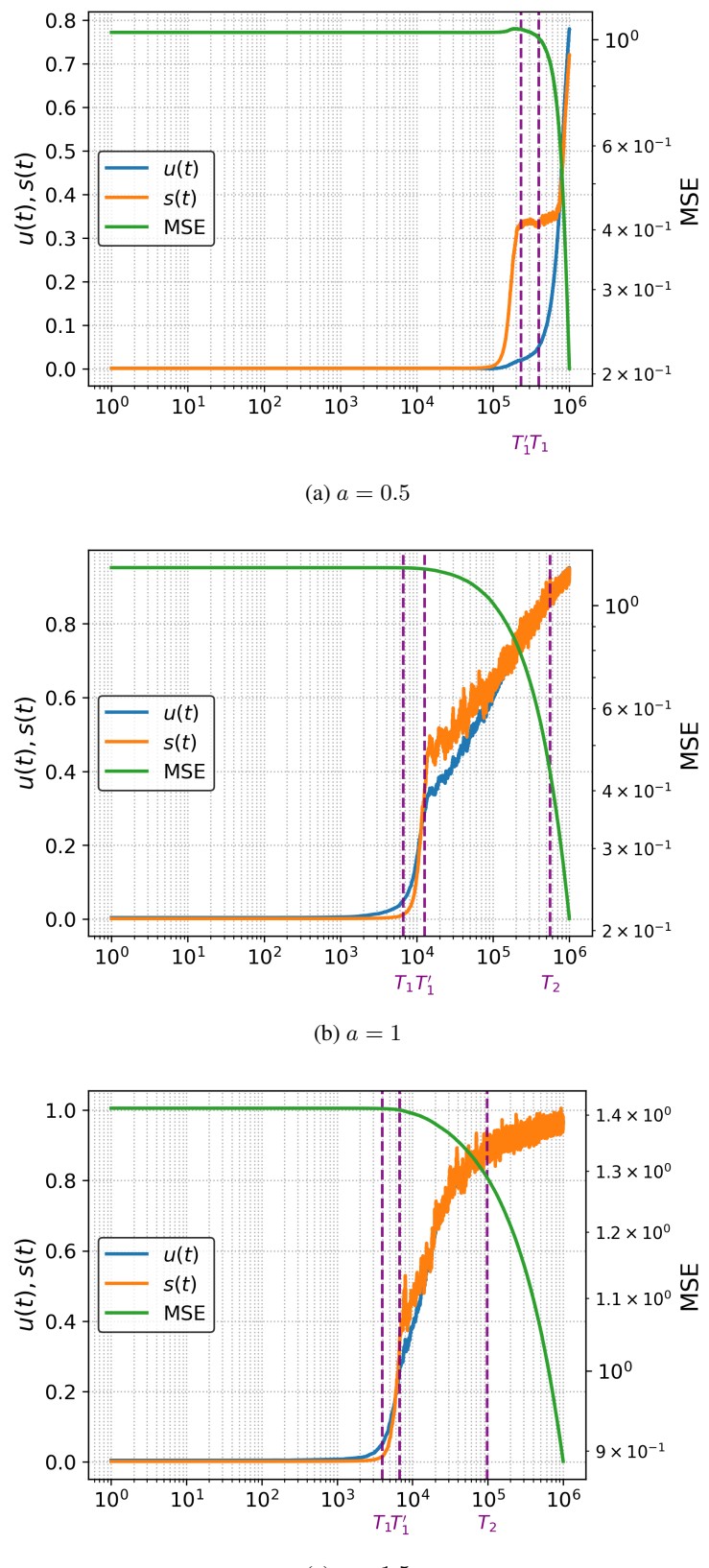

(a) $a = 0.5$

(b) $a = 1$

(c) $a = 1.5$

Figure 5: Dynamics of online SGD for different $a$, $d = 1000$, $\eta = 10^{-3}$.

