# OpenReview forum: "Fast Escape, Slow Convergence: Learning Dynamics of Phase Retrieval under Power-Law Data"
_ICLR.cc/2026/Conference — ICLR 2026 Oral_

### Official Review · Reviewer_8Pxo · 2025-10-30

**Soundness:** 4
**Presentation:** 2
**Contribution:** 3
**Rating:** 6
**Confidence:** 3

**Summary:**

The authors rigorously study and characterize the gradient flow dynamics of the population loss for phase retrieval in a setting where the data are drawn from a normal distribution whose covariance matrix has eigenvalues following a power-law distribution.
Due to the combined effects of nonlinearity and anisotropy across learning directions, they show that the dynamics are qualitatively different from cases where only one of these aspects is present.
Analytically, they find that the dynamics are governed by an infinite hierarchy of coupled ODEs for the order parameters. Using these equations, they are able to describe the entire learning trajectory and rigorously distinguish three distinct phases, in agreement with numerical experiments. In particular, they reveal that the dynamics are generally slower than in the isotropic case, and that the final phase is controlled by the smallest eigenvalues of the covariance matrix, corresponding to the hardest directions to learn. In this final regime,
they obtain an analytical scaling law describing how the mean-squared error decays with training time, showing that the decay exponent is determined by the power-law distribution of the covariance spectrum.

**Strengths:**

The setting studied in this work allows for a complete and rigorous analysis of the gradient flow of the population loss in a case where both a nonlinearity in the regression problem and a power-law distribution of the data are present.
To my best knowledge, this is the first rigorous paper to do so.
The analytical description is very thorough and matches very well the numerical simulations, allowing one to build a general intuition about the interplay between these two elements.

**Weaknesses:**

The setting is intentionally specific to enable a rigorous analysis since the authors use both the generalization risk (whose landscape is devoid of spurious minima) and the gradient flow approximation, which does not directly capture discrete-time effects of an actual gradient descent.
Moreover, while a scaling law is analytically found in the third regime, it is not directly linked to the classical scaling laws that connect generalization error with the number of data points (here assumed infinite in the population loss) and the number of parameters (here equal to the input dimension and also tending to infinity).
The technical sections are quite dense, making the paper more difficult to read in that part.

**Questions:**

While the works of Hestness, Kaplan, and Hoffmann are cited as motivation, those scaling laws relate error to number of data, number of parameters and discrete FLOPs. How do you see your dynamical scaling laws connecting conceptually or quantitatively to the empirical ones?

In ”Escaping mediocrity” (Arnaboldi, Krzakala, Loureiro, Stephan, 2023a), the authors analyze how online SGD benefits from over-parametrization (among other factors) to accelerate escape. Do you expect similar over-parametrization gains to influence the phase structure you observe here, for example, by modifying the escape from mediocrity stage or the later spectral-tail phase?

Do you have insights on the qualitative differences there would be there would be if one studied the dynamics where the student norm (the Frobenius norm) is kept fixed to the same one of the teacher, so that the dynamics only aligns the vector w to w*?

There is a minor graphical bug in Figure 2 that causes the labels in the x axis to overlap.

---

> ### Author Response · Authors · 2025-11-18
>
> 1. > While the works of Hestness, Kaplan, and Hoffmann are cited as motivation, those scaling laws relate error to number of data, number of parameters and discrete FLOPs. How do you see your dynamical scaling laws connecting conceptually or quantitatively to the empirical ones?
>
>    Gradient flow on the population risk can be seen as the small learning rate limit of one-pass SGD. Therefore, we can identify the running time $t = \eta k$, where $k$ is the number of SGD steps, provided $\eta \ll 1$. Since for one-pass SGD one sample is used per iteration, the amount of data seen by the algorithm is equivalent to the total number of discrete steps, which allows us to translate running time scaling to a scaling in the number of samples. This can be similarly done for the number of $\mathrm{flops} = k \times \mathrm{batch\ size} \times d$, with $\mathrm{batch\ size} = 1$ for one-pass SGD.
>
> 2. > In ”Escaping mediocrity” (Arnaboldi, Krzakala, Loureiro, Stephan, 2023a), the authors analyze how online SGD benefits from over-parametrization (among other factors) to accelerate escape. Do you expect similar over-parametrization gains to influence the phase structure you observe here, for example, by modifying the escape from mediocrity stage or the later spectral-tail phase?
>
>    Precisely answering this question would require extending our methodology to the two-layer setting. While we believe such an extension is feasible, it entails substantial additional work beyond the scope of the current manuscript. Nevertheless, by extrapolating the structure of our equations, we anticipate that overparameterization would play a similar role in that setting, yielding an additional $\Theta_d(1)$ improvement in the escape-of-mediocrity time on top of the gains already induced by anisotropy.
>
> 3. > Do you have insights on the qualitative differences there would be if one studied the dynamics where the student norm (the Frobenius norm) is kept fixed to the same one of the teacher, so that the dynamics only aligns the vector $w$ to $w^\*$?
>
>    Numerically, we observed that using the same scaling as $w^\*$ for $w$ didn't change much of the dynamics. However, the analysis is easier by using a small random initialization, a common practice in the feature learning regime.

---

### Official Review · Reviewer_aHGn · 2025-10-30

**Soundness:** 4
**Presentation:** 3
**Contribution:** 3
**Rating:** 6
**Confidence:** 3

**Summary:**

This paper studies the learning dynamics of gradient descent (GD) in the context of phase retrieval with anisotropic gaussian inputs. The main technical approach is based on reducing the coupled dynamics of an infinite hierarchy of moments to a Volterra integral equation, and then obtaining solution bounds and asymptotics via Laplace-type analysis.

The main conceptual result, supported by theory and numerical experiments, is that learning roughly exhibits 3 phases:
- Escape: The weights slowly escape the small gradient regime near \$w=0\$ and acquire nontrivial overlap with the ground truth weights.
- Convergence: The relevant summary statistics converge toward their correct population values while the loss converges toward 0. The authors subdivide this phase into two qualitatively distinct subphases.
- Spectral tail learning: The authors finally consider a second metric of learning, the MSE of the learned weights. At the beginning of phase 3, the MSE remains high despite low fitting loss. The MSE decreases through phase 3 at a rate that depends on the spectral decay of the data covariance; as one might expect, flatter spectra - i.e. smaller decay exponents - give rise to faster learning in this phase.

Another conceptual result that is discussed only very briefly is a tradeoff between fast initial learning and fast tail learning. Roughly speaking: low rank data drives fast initial alignment but leads to slow learning of small-eigenvalue directions.

The bulk of the paper is devoted to describing the analytical results and outlining the proof strategy, though the authors provide empirical support for their results through limited numerical experiments (2 main figures and several supplemental figure panels).

**Strengths:**

This paper studies an important issue in learning theory. It considers a nontrivial, currently poorly understood, setting combining model nonlinearity and data anisotropy - both of which are expected to affect the detailed behavior of neural scaling laws. The technical analysis of the paper is interesting and rigorous, and is likely to be applicable in future work. The main theoretical conclusions of the paper are intuitive and appealing, and are supported by limited experimental verification/exploration. Finally, the paper is well written, and the derivations are relatively easy to follow.

**Weaknesses:**

**Robustness of conclusions and applicability to more general settings**. The paper obtains satisfying conclusions about the qualitative behavior of GD dynamics in phase retrieval. However, the neat 3-phase picture appears to depend somewhat on the loss structure of the specific problem under consideration. Currently, the paper does not appear to give any reason - even hand waving - to suspect that more complex models will exhibit escape, convergence, and/or tail learning phases resembling those of phase retrieval. The extent to which these specific results generalize is therefore unclear. This issue is especially concerning given that the results, while theoretically interesting, are likely to be of limited practical relevance in the specific context of phase retrieval, given the wealth of published fitting algorithms that significantly outperform (S)GD.

Relatedly, I have a question/concern about the dependence of the results on the particular initialization scheme studied in the paper (see detailed questions).

**Impact of anisotropy**. Data anisotropy is the main novel ingredient of the analysis. While this introduces novel and interesting technical challenges for the analysis, it appears to have a relatively simple effect on the learning curves. In particular, I don’t feel convinced that anisotropy “leads to a completely different picture from the isotropic intuition” as stated in the paper. The paper would benefit from more explicit discussion of the impact of anisotropy, especially relative to the escape/convergence tradeoff referenced in the title. (Right now, discussion of this result appears to be limited to two sentences in the Contributions section.) See detailed comments below.

**Questions:**

**Major comments/questions**

1. Clarify/amplify role of anisotropy
    - The title and contributions section make reference to a tradeoff in escape and tail learning speeds. This tradeoff is not discussed further in the main text, and is not illustrated in any of the main figures. Given the prominence of this result, I feel it deserves more attention.
    - It took a careful reading to understand what was being traded off - ie. “escape from mediocrity” and “convergence to low risk” (line 091). These terms ought to be defined clearly up front. This confusion was related to the fact that the paper studies two distinct measures of performance. See comment 2.
    - In light of the results, I feel that the description of the anisotropic setting as a “completely different picture from the isotropic intuition” (line 087) is too strong. It seems more consistent with the results to say that anisotropy leaves the dynamics qualitatively unchanged while determining the precise scaling exponents and phase transition times.
2. Up until Section 4.2, it is not obvious on first reading that the paper studies two distinct measures of performance - population loss and MSE. Several points of confusion could be avoided by making this clear early on. I think the MSE should also be defined alongside the population loss in Section 2 - Problem Setup, and that the text should state explicitly that the analysis studies both quantities, which become relevant in different phases of learning.
3. Escape phase
    - The concept of an escape phase in phase retrieval is apparently well known (eg. cited paper Ben Arous et al., 2025). I feel this is worth mentioning when describing the escape phase on line 256.
    - I am somewhat confused by the initialization scheme. In particular, I don’t understand why the authors have $|w^\star| = O(\sqrt{d})$ but initialize the weights with $|w|=1$. This leads to issues later on: Remark 1 seems to suggest that the escape phase occurs because the initial gradient is small, in turn because $u(0)\sim d^{-1/2}$. However, the third term of 3.1 has both $u$ and $w^\star$, which has norm $d^{1/2}$. It’s not clear that the alignment term of the gradient is actually small relative to others. A very different explanation is given on line 256, which points out that since $w(0)\approx 0$, the weights start out near the local maximum at $0$, and thus require long times to escape. Please clarify. (To my mind, the simplest initialization has both $w,w^\star$ with norm $\sqrt{d}$. Then $u(0)\approx d^{-1/2}$ implies the alignment term in the gradient is small relative to the first two “norm terms”. Gradient descent quickly sends $s\to 1/3$, and then one has a slow “alignment phase” wherein $u$ increases toward 1.)
4. The authors should briefly describe the approximation scheme used in the phase analysis, perhaps around the statement of the ODEs in eqs 3.3-3.5. In particular, the analysis proceeds by assuming simple forms for $s(t),u(t)$, obtaining expressions for the simplified dynamics, and then bounding the approximation error. The initial approximation $\Theta(t)\approx t$ corresponds to $s(t)=0$ - ie. initially small weights, while the phase 3 approximation corresponds to $s(t)=u(t)=1$ - ie. perfect convergence of macroscopic overlaps.
5. The authors may want to comment on the meaning, relevance, and broader applicability of the division of Phase II into two subphases. Is there any broader conclusion to be drawn, or is this merely a peculiarity of phase retrieval?


**Minor**
1. Line 083-084 uses uppercase $T$ for time while later equations use $t$.
2. Line 323 typo (growth -> grows)
3. Line 413 / eq. 5.4 - I don’t see a definition for $\alpha$ anywhere in the main text.

---

> ### Author Response · Authors · 2025-11-18
>
> Thank you for your helpful suggestions to improve the presentation. We incorporated them in the revised version.
>
> 1. > Clarify/amplify role of anisotropy.
>    > 1. The title and contributions section make reference to a tradeoff in escape and tail learning speeds. This tradeoff is not discussed further in the main text, and is not illustrated in any of the main figures. Given the prominence of this result, I feel it deserves more attention.
>    > 2. It took a careful reading to understand what was being traded off - ie. “escape from mediocrity” and “convergence to low risk” (line 091). These terms ought to be defined clearly up front. This confusion was related to the fact that the paper studies two distinct measures of performance. See comment 2.
>    > 3. In light of the results, I feel that the description of the anisotropic setting as a “completely different picture from the isotropic intuition” (line 087) is too strong. It seems more consistent with the results to say that anisotropy leaves the dynamics qualitatively unchanged while determining the precise scaling exponents and phase transition times.
>
>    We clarified the definitions of "escaping mediocrity" and "low-risk" in Section 1.1 of the introduction. We also added a picture (Figure 3) showing the evolution of the correlation $u(t)$ for different values of $a$, illustrating the "fast escape, slow convergence" phenomenon. We also revised the expression “completely different picture” in line 88 to adopt a more moderate phrasing.
>
> 2. > Until Section 4.2, it is not obvious on first reading that the paper studies two distinct measures of performance - population loss and MSE. Several points of confusion could be avoided by making this clear early on. I think the MSE should also be defined alongside the population loss in Section 2 - Problem Setup, and that the text should state explicitly that the analysis studies both quantities, which become relevant in different phases of learning.
>
>    We emphasized this distinction between the population loss and the MSE in Section 2 in the revised version.
>
> 3. > Initialization scheme.
>
>    We use a small random initialization, which is standard in the feature-learning regime. Changing the initialization scale does not affect the normalized alignment with $w^\star$, but starting from a small norm simplifies the analysis of the early dynamics.
>
> 4. > The authors should briefly describe the approximation scheme used in the phase analysis, perhaps around the statement of the ODEs in eqs 3.3–3.5.
>
>    In the revised version, we specified which approximations are used in each phase directly in the titles of the corresponding paragraphs.
>
> 5. > The authors may want to comment on the meaning, relevance, and broader applicability of the division of Phase II into two subphases. Is there any broader conclusion to be drawn, or is this merely a peculiarity of phase retrieval?
>
>    We believe this is specific to the model considered in this work: when $s(t)$ crosses above $1/3$, the prefactor $1 - 3 s(t)$ appearing in the ODE giving $u$ changes sign, which has a qualitative effect on the dynamics.

---

### Official Review · Reviewer_KHcf · 2025-10-31

**Soundness:** 4
**Presentation:** 4
**Contribution:** 2
**Rating:** 6
**Confidence:** 3

**Summary:**

The paper uses phase retrieval as a simple setting to study non- linear dynamics, i.e., learning a single-neuron with quadratic activation in the presence of noise. This is a non-convex problem with two global minimizers, a local max at 0 and a saddle manifold. As a result, the landscape is nice in the sense of no spurious minima, but anisotropy (power-law decay eigenvalues of the covariance Q) makes convergence slow: in particular, the parameter trajectory escapes quickly from low alignment but then converges slowly because of small eigenvalues.

**Strengths:**

I think the key originality this paper brings is the analysis under anisotropy. I also think on the whole that the paper is well written and clearly explained. In terms of significance I think the identification of slow convergence with the tail of the distribution of the data is nice, particularly around the emergence of this third phase not present when $a=0$. I was also think the fact that a larger $a$ implies faster decay of the loss in the first phase but slower convergence in the third phase is interesting and a nice characterization. Finally the fact that for isotropic data you can just track the magnitude and inner product of the weights versus the target weights, but for anisotropic data this no longer holds and you must track an infinite tower of weighted overlaps instead (proposition 3), I also liked. The analysis appears to involve a nice combination of techniques.

**Weaknesses:**

My only critique perhaps concerns the idealized setting, namely a quadratically activated neuron trained under full batch gradient flow on Gaussian data with power law decay. It is not clear to me how much of the narrative here applies elsewhere to other non-linear optimization problem settings of interest. As a result, I think the scope and general applicability of the results are not clear.

**Questions:**

- On the necessity of Gaussian data: would the same dynamics hold if the inputs were drawn from a non-Gaussian but rotationally symmetric distribution say? Which parts of the proof genuinely require Gaussianity?

- Can one compute the scaling of the loss explicitly in the tail phase as a function of  $a$?

- How does this multiphase trajectory analysis connect to other problems in machine learning. Do you think you can gain any insight say into training of e.g., deep linear networks from this work and the observed phases of training there?

- Your work focuses on gradient flow under anisotropy which is what preconditioning approaches like natural gradient or Adam try to fix. Have you considered or thought about performing a similar analysis with methods like this to see if they can ameliorate the issues of slow escape from a saddle and slow convergence to a global minimum? In short, this setup seems like a nice testbed for studying these two problems, escaping saddles and slow convergence along flat directions?

**Details Of Ethics Concerns:**

Not applicable.

---

> ### Author Response · Authors · 2025-11-18
>
> 1. > My only critique perhaps concerns the idealized setting, namely a quadratically activated neuron trained under full batch gradient flow on Gaussian data with power law decay. It is not clear to me how much of the narrative here applies elsewhere to other non-linear optimization problem settings of interest. As a result, I think the scope and general applicability of the results are not clear.
>
>    As mentioned in our global answer, in contrast to the isotropic setting, when data is strongly anisotropic, the dynamics is no longer reducible to a low-rank system of ODEs, but involves an infinite system of coupled ODEs. Understanding how such a system can be solved in the particular case of Phase Retrieval is a necessary and non-trivial first step toward studying more general models.
>
> 2. > On the necessity of Gaussian data: would the same dynamics hold if the inputs were drawn from a non-Gaussian but rotationally symmetric distribution, say? Which parts of the proof genuinely require Gaussianity?
>
>    The Gaussianity assumption is used critically to obtain a closed-form expression of the loss via Wick's formula. The case of inputs following a symmetric distribution has been studied by Bruna et al. (2023). The proof strategy is similar to that of the isotropic single-index model; instead of using the Hermite basis, they employ spherical harmonics to derive the equation governing the dynamics. This type of distribution is close to the isotropic case, and its tail doesn't follow a power-law decay.
>
> 3. > Can one compute the scaling of the loss explicitly in the tail phase as a function of $a$?
>
>    The loss is a function of $u$ and $s$, which are functions of $a$. It seems, however, difficult to provide a simple closed-form formula for these quantities.
>
> 4. > How does this multiphase trajectory analysis connect to other problems in machine learning. Do you think you can gain any insight say into training of e.g., deep linear networks from this work and the observed phases of training there?
>
>    We do believe that the key technical idea in our analysis can be generalized to more complex scenarios, such as two-layer neural networks. In the isotropic case, the SGD dynamics in the two-layer case can be significantly richer, for instance, displaying a saddle-to-saddle structure associated with the specialization of different hidden units to the structure of the target (Saad & Solla, 1995b;a; Goldt et al., 2019; Bietti et al., 2025). Understanding the interplay between anisotropy and the hierarchical timescales for learning in this richer setting is a very interesting question for future work!
>
> 5. > Your work focuses on gradient flow under anisotropy, which is what preconditioning approaches like natural gradient or Adam try to fix. Have you considered or thought about performing a similar analysis with methods like this to see if they can ameliorate the issues of slow escape from a saddle and slow convergence to a global minimum? In short, this setup seems like a nice testbed for studying these two problems, escaping saddles, and slow convergence along flat directions?
>
>    We thank the reviewer for this interesting suggestion. While we have not looked into it, very recent work in the literature have studied the impact of momentum in the scaling laws in the linear case (Ferbach et al. 2025). We believe our technical result could be adapted to this setting.
>
>    - D Ferbach, K Everett, G Gidel, E Paquette, C Paquette. *Dimension-adapted Momentum Outscales SGD*. https://arxiv.org/abs/2505.16098

---

> > ### Comment · Reviewer_KHcf · 2025-11-25
> >
> > Thanks for answering my questions. Despite the limitations of the setup I will vote to accept.

---

### Official Review · Reviewer_gLFC · 2025-11-01

**Soundness:** 3
**Presentation:** 3
**Contribution:** 3
**Rating:** 4
**Confidence:** 3

**Summary:**

This paper considers the theory of scaling laws of phase retrieval, a classical nonlinear regression problem. The authors start by asking "How does the input spectrum govern finite-time convergence in nonlinear regression, and can we predict the learning curve from the spectral decay?".

To answer the question, they consider the gradient flow of optimizing the MSE loss, which can be formulated as infinite-dimensional, continuous dynamical systems. The dynamics is further analyzed through different phases in Section 4, including escape, approximate convergence, and spectral-tail learning. In Section 5 the authors provide a sketch of their proof, and then they wrap up the work with some discussions on limitations and future directions.

**Strengths:**

1. This paper gives a solid theoretical study of scaling laws of phase retrieval problems.

2. The authors reveal that the problem can be viewed as infinite-dimensional dynamical systems, and can be analyzed by some math tools such as ODE, dynamical systems, etc.

**Weaknesses:**

1. It is unclear how the dynamics of gradient flow of the phase retrieval problems can inspire the design of new algorithms for solving real-world, large-scale problems. Although this paper mainly focuses on theory, it might be beneficial to include some discussions on the connections between this toy problem and real-world ones.

2. Most of the results include the big-O notations, meaning that the conclusions can only hold when certain numbers are sufficiently large/small. This might indicate that the theorems and propositions are a bit limited.

3. The mathematical tools and conclusions are standard in theory lliterature.

**Questions:**

1. Could the authors provide some discussions on the connections between the problem studied and real-world problems, such as how the reuslts would inspire new algorithms design for large-scale real-data problems.

2. Are there any challenges of removing the big-O notations and asymptotics in theorems? For example theorem 1 requires d to be sufficiently large, how about the cases when d does not satisfy this?

3. The analysis is mainly on the gradient flow. Would it be possible to analyze the discrete cases, i.e., gradient descent or even stochastic gradient descent algorithms, which might align better with real-world scenarios.

---

> ### Author Response · Authors · 2025-11-18
>
> 1. > “Could the authors provide some discussions on the connections between the problem studied and real-world problems, such as how the reuslts would inspire new algorithms design for large-scale real-data problems."
>
>    Our analysis reveals why anisotropy induces heterogeneous learning speeds across eigendirections. This motivates the use of preconditioned or geometry-aware optimizers (e.g., natural gradient, K-FAC, Shampoo), which explicitly correct for covariance structure and could speed up convergence in anisotropic settings.
>
> 2. > “Are there any challenges of removing the big-O notations and asymptotics in theorems? For example theorem 1 requires d to be sufficiently large, how about the cases when d does not satisfy this?"
>
>
>    When $d$ is small, the correlation at initialization is already of constant order, so Phase I is very short and does not play a meaningful role. In the isotropic setting, however, Phase I is the principal obstruction to learning the model, and to keep the parallel with that setting, it is natural to consider $d$ large. Furthermore, when $d$ is small, the eigenvalues $\lambda_i$ are all of comparable magnitude, so there is no significant spectral separation and no particular direction is harder to learn.  Note that we are not in an asymptotic regime: our bounds hold for any finite $d$ provided that $d$ is larger than some numerical constant.
>
> 3. > “The analysis is mainly on the gradient flow. Would it be possible to analyze the discrete cases, i.e., gradient descent or even stochastic gradient descent algorithms, which might align better with real-world scenarios."
>
>
>    Many theoretical works (e.g., Bietti et al., 2025) focus on idealized settings, such as Gradient Flow (GF) on the population loss, as a first attempt to understand complex dynamics. Since our main focus was to understand the effect of anisotropy, we left as future work questions related to the discretization and handling the noise. We expect that in the online setting, the noise can be handled by a martingale argument similar to that in Ben Arous et al. (2022). However, contrary to our setting, we anticipate that the noise will become dominant after a sufficient number of iterations.
>
> 4. > “The mathematical tools and conclusions are standard in theory lliterature."
>
>    Although we employ classical tools from probability theory and ODEs, the analysis of the anisotropic non-linear setting requires new technical ideas (formulation as an infinite-dimensional ODE, reducing the system to an analytically tractable Volterra equation through approximations), which are not a simple extension of the isotropic case widely studied in the literature. To the best of our knowledge, our work is the first to rigorously address this setting.

---

### Author Response · Authors · 2025-11-18
**Global Response**

We thank the reviewers for their thoughtful feedback. Several recurring themes emerged, which we address below and have incorporated into the revised manuscript. Individual comments are answered in detail in the point-by-point rebuttal.

More than one reviewer noted the simplicity of the model studied. While phase retrieval on anisotropic, power-law data is a specific setting, it captures the essential theoretical challenges that arise in broader problems of interest:

- **Non-convexity:** Classical optimization primarily focuses on convex objectives or provides only local guarantees. Phase retrieval is a fundamentally non-convex problem that arises naturally in signal processing (Dong et al., 2023) and has become a standard testbed in theoretical ML for understanding non-convex optimization (Ma et al., 2021; Candès et al., 2015; Tan & Vershynin, 2019; 2023; Davis et al., 2020; Sarao Mannelli et al., 2020; Arnaboldi et al., 2023a; Martin et al., 2024; Erba et al., 2025; Ben Arous et al., 2025). Our work provides a global convergence rate for the convergence of GF.

- **Anisotropy:** A central simplifying assumption in much of the literature above is isotropy of the data distribution. Rotational invariance reduces the SGD dynamics to a small set of concentrated variables, dramatically simplifying high-dimensional analysis (Ben Arous et al., 2022). Anisotropy breaks this structure and introduces substantial technical difficulties. Our work develops a new analytical approach to anisotropic, non-convex dynamics, which we believe can be extended to richer models such as two-layer neural networks and more general link functions.

- **Scaling laws:** Understanding SGD convergence under anisotropic power-law data has recently gained traction as a model for neural scaling laws (Hestness et al., 2017; Kaplan et al., 2020; Hoffmann et al., 2022). Existing results, however, focus primarily on linear or isotropic settings. To our knowledge, our work is the first to analyze this question in a non-convex and anisotropic covariate model, uncovering a non-trivial trade-off between weak and strong learning that is specific to the nonlinear setting.

In summary, the setting we study is both relevant to the ICLR theoretical community and exhibits rich, nontrivial phenomena. The technical challenges addressed here are not straightforward extensions of prior work, and we believe the methods introduced are of independent value for analyzing more complex problems of interest.

---

### Meta-Review · Area_Chair_ULq4 · 2026-01-03

**Summary:**

This paper provides a rigorous theoretical analysis of the learning dynamics of gradient flow for phase retrieval with anisotropic Gaussian inputs exhibiting power-law covariance spectra.
In contrast to the isotropic case, where the dynamics are relatively simple, the authors show that the setting under study exhibits a three-phase learning behavior: escape, convergence, and spectral tail learning.
Based on the theoretical analysis, the paper also offers several insightful discussions, including a potential tradeoff between fast initial alignment and slow tail learning.

All four reviewers acknowledge the novelty and significance of the contribution, while also noting the simplicity of the model considered.
In my view, despite this simplicity, the results provide sufficiently new and valuable insights to the community.

**Reviewer Concerns:**

- Reviewer **gLFC** asked for: (1) further insights into practical models/algorithms, and (2) extensions to discrete gradient descent and other settings. I believe that these concerns are still outstanding, but the explanations provided by the authors are reasonable.
- Reviewer **KHcf** asked for: (1) extension beyond the current Gaussian setting, (2) more explicit scaling characterization as a function of the key parameter $a$, (3) extensions and connections to preconditioning approaches, as well as (4) practical insights from the theoretical analysis. I think that most of these points were addressed, at least partially, during the rebuttal.
- Reviewer **aHGn** asked for: (1) extensions to more general settings, and (2) clarification on the role of anisotropy. I think that some of these were addressed during the rebuttal, while others were not, but the authors' replies make sense.
- Reviewer **8Pxo** asked for: (1) extensions to more general and practical settings, and (2) further connections to previously established (explicit) scaling laws. I think that some of these were addressed during the rebuttal, while others were not, but the authors' replies are convincing in general.

**Reviewer Scores:**

- I believe that reviewer **gLFC** would have slightly increased or kept the score unchanged (4), as some of the concerns remain outstanding (although I think that the authors' explanations are reasonable).
- I believe that reviewer **KHcf** would have slightly increased the score or kept it unchanged (6, which is already positive), since most concerns were at least partially addressed during the rebuttal.
- I believe that reviewer **aHGn** would have kept the score unchanged (6, which is already positive), as some concerns were addressed while others were not, but the authors' replies are reasonable overall.
- I believe that reviewer **8Pxo** would have kept the score unchanged (6, which is already positive), as some concerns were addressed and the authors' responses are generally convincing.

---

### Decision · Program_Chairs · 2026-01-26

Accept (Oral)